# Robust Online Conformal Prediction under Uniform Label Noise

## Abstract

Conformal prediction is an emerging technique for uncertainty quantification that constructs prediction sets guaranteed to contain the true label with a predefined probability. Recent work develops online conformal prediction methods that adaptively construct prediction sets to accommodate distribution shifts. However, existing algorithms typically assume *perfect label accuracy* which rarely holds in practice. In this work, we investigate the robustness of online conformal prediction under uniform label noise with a known noise rate, in both constant and dynamic learning rate schedules. We show that label noise causes a persistent gap between the actual mis-coverage rate and the desired rate $\alpha$, leading to either overestimated or underestimated coverage guarantees. To address this issue, we propose *Noise Robust Online Conformal Prediction* (dubbed NR-OCP) by updating the threshold with a novel *robust pinball loss*, which provides an unbiased estimate of clean pinball loss without requiring ground-truth labels. Our theoretical analysis shows that NR-OCP eliminates the coverage gap in both constant and dynamic learning rate schedules, achieving a convergence rate of $\mathcal{O}(T^{-1/2})$ for both empirical and expected coverage errors under uniform label noise. Extensive experiments demonstrate the effectiveness of our method by achieving both precise coverage and improved efficiency.

## 1. Introduction

Machine learning techniques are revolutionizing decision-making in high-stakes domains, such as autonomous driving (Bojarski et al., 2016) and medical diagnostics (Caruana et al., 2015). It is crucial to ensure the reliability of model predictions in these contexts, as wrong predictions can result in serious consequences. While various techniques have been developed for uncertainty estimation, including confidence calibration (Guo et al., 2017) and Bayesian neural networks (Smith, 2013), they typically lack rigorous theoretical guarantees. *Conformal prediction* addresses this limitation by establishing a systematic framework to construct prediction sets with provable coverage guarantee (Vovk et al., 2005; Shafer & Vovk, 2008; Balasubramanian et al., 2014; Angelopoulos et al., 2023). Notably, this framework requires no parametric assumptions about the data distribution and can be applied to any black-box predictor, which makes it a powerful technique for uncertainty quantification.

Recent research extends conformal prediction to *online* setting where the data arrives in a sequential order (Gibbs & Candes, 2021; Feldman et al., 2023; Bhatnagar et al., 2023; Angelopoulos et al., 2024; Gibbs & Candès, 2024; Haji-hashemi & Shen, 2024). These methods provably achieve the desired coverage property under arbitrary distributional changes. However, previous studies typically assume *perfect label accuracy*, an assumption that seldom holds true in practice due to the common occurrence of noisy labels in online learning (Ben-David et al., 2009; Natarajan et al., 2013; Wu et al., 2024). Recent work (Einbinder et al., 2024) proves that online conformal prediction can achieve a conservative coverage guarantee under uniform label noise, leading to unnecessarily large prediction sets. However, their analysis relies on a strong distributional assumption of non-conformity score and cannot quantify the specific deviation of coverage guarantees. These limitations motivate us to establish a general theoretical framework for this problem and develop a noise-robust algorithm that maintains precise coverage guarantees while producing small prediction sets.

In this work, we analyze online conformal prediction under uniform label noise with a known noise rate, in both constant (Gibbs & Candes, 2021) and dynamic (Angelopoulos et al., 2024) learning rate schedule. Notably, our theoretical results are independent of the distributional assumption made in the previous work (Einbinder et al., 2024). In particular, we demonstrate that label noise causes a persistent gap between the actual mis-coverage rate and the desired rate $\alpha$, with higher noise rates resulting in larger gaps. This gap can lead to either overestimated or underestimated coverage guarantees, which depend on the size of the prediction sets.

To address this challenge, we propose *Noise-Robust Online Conformal Prediction* (dubbed NR-OCP) by updating the

[1]Anonymous Institution, Anonymous City, Anonymous Region, Anonymous Country. Correspondence to: Anonymous Author <anon.email@domain.com>.

Preliminary work. Under review by the International Conference on Machine Learning (ICML). Do not distribute.

threshold with a *robust pinball loss*, which provides an unbiased estimate of clean pinball loss value without requiring ground-truth labels. Specifically, we construct the robust pinball loss as a weighted combination of the pinball loss with noisy scores and the pinball loss with scores of all classes. We prove that this loss is equivalent to the pinball loss under clean labels in expectation. Our theoretical analysis shows that NR-OCP eliminates the coverage gap, caused by the label noise, in both constant and dynamic learning rate schedules, achieving a convergence rate of $\mathcal{O}(T^{-1/2})$ for both empirical and expected coverage errors (i.e., absolute deviation of the empirical and expected mis-coverage rate from the target level $\alpha$) under the uniform label noise.

To verify the effectiveness of our method, we conduct extensive experiments on CIFAR-100 (Krizhevsky, 2009) and ImageNet (Deng et al., 2009) with synthetic uniform label noise. Empirical results show that NR-OCP consistently achieves the desired $1 - \alpha$ long-run coverage rate, while the standard online conformal prediction updated with noisy labels (baseline) shows a significant coverage gap. For example, using DenseNet121 on CIFAR-100 dataset with noise rate $\epsilon = 0.05$, error rate $\alpha = 0.1$, and constant learning rate, the baseline method (Gibbs & Candes, 2021) significantly deviates from the target coverage level of 0.9 using the LAC score, exhibiting a coverage gap of 3.178% and an average prediction set size of 25.88. In contrast, NR-OCP achieves a negligible coverage gap of 0.156% and a smaller average prediction set size of 15.20. Notably, our method demonstrates superior performance over the baseline across different settings, including various model architectures, error rates, noise rates, and non-conformity scores.

We summarize our contributions as follows:

- We present a general theoretical framework for analyzing the impact of uniform label noise on online conformal prediction under constant (Proposition 3.1) and dynamic (Proposition 4.1) learning rates, respectively. Our theoretical results are free from the distributional assumption in prior work (Einbinder et al., 2024).

- We introduce NR-OCP, a novel method of online conformal prediction that is robust to uniform label noise. Our method includes a *robust pinball loss* that provides an unbiased estimate of clean pinball loss without requiring access to clean labels.

- We prove that NR-OCP eliminates the coverage gap in online conformal prediction with both constant (Propositions 3.3 and 3.4) and dynamic learning rate (Propositions 4.3 and 4.4), achieving convergence rates of $\mathcal{O}(T^{-1/2})$ for both empirical and expected coverage errors under uniform label noise.

## 2. Preliminary

**Online conformal prediction.** We study the problem of generating prediction sets in *online* classification where the data arrives in a sequential order (Gibbs & Candes, 2021; Angelopoulos et al., 2024). Formally, we consider a sequence of data points $(X_t, Y_t)$, $t \in \mathbb{N}^+$, which are sampled from a joint distribution $\mathcal{P}_{\mathcal{X}\mathcal{Y}}$ over the input space $\mathcal{X} \subset \mathbb{R}^d$, and the label space $\mathcal{Y} = \{1, \ldots, K\}$. In online conformal prediction, the goal is to construct prediction sets $\mathcal{C}_t(X_t)$, $t \in \mathbb{N}^+$, that provides *precise* coverage guarantee: $\lim_{T \to +\infty} \frac{1}{T} \sum_{t=1}^{T} \mathbb{1}\{Y_t \notin \mathcal{C}_t(X_t)\} = \alpha$, where $\alpha \in (0, 1)$ denotes a user-specified error rate.

At each time step $t$, we construct a prediction set $\mathcal{C}_t(X_t)$ by

$$\mathcal{C}_t(X_t) = \{y \in \mathcal{Y} : \mathcal{S}(X_t, y) \leq \hat{\tau}_t\}.$$

where $\hat{\tau}_t$ is a data-driven threshold, and $\mathcal{S} : \mathcal{X} \times \mathcal{Y} \to \mathbb{R}$ denotes a *non-conformity score* function that measures the deviation between a data sample and the training data. For example, given a pre-trained classifier $f : \mathcal{X} \to \mathbb{R}^K$, the LAC score (Sadinle et al., 2019) is defined as $\mathcal{S}(X, Y) = 1 - \hat{\pi}_Y(X)$, where $\hat{\pi}_Y(X) = \sigma_Y(f(X))$ denotes the softmax probability of instance $X$ for class $Y$, and $\sigma$ is the softmax function. For notation shorthand, we use $S_t$ to denote the random variable $\mathcal{S}(X_t, Y_t)$ and use $S_{t,y}$ to denote $\mathcal{S}(X_t, y)$ for a given class $y \in \mathcal{Y}$. Following previous work (Kiyani et al., 2024; Angelopoulos et al., 2024), we will assume that the non-conformity score function is bounded, and the threshold is specifically initialized:

**Assumption 2.1.** The score is bounded by $\mathcal{S}(\cdot, \cdot) \in [0, 1]$.

**Assumption 2.2.** The threshold is initialized by $\hat{\tau}_1 \in [0, 1]$.

After the prediction, we can observe the corresponding label $Y_t$ for $X_t$. Given a desired mis-coverage rate $\alpha$, online conformal prediction (Gibbs & Candes, 2021; Angelopoulos et al., 2024) updates the threshold $\hat{\tau}_t$ with *pinball loss*:

$$l_{1-\alpha}(\tau, s) = \alpha(\tau - s)\mathbb{1}\{\tau \geq s\} + (1-\alpha)(s-\tau)\mathbb{1}\{\tau \leq s\},$$

where $\tau$ denotes a threshold and $s$ is a non-conformity score. Then, the threshold is updated via *online gradient descent*:

$$\begin{aligned}
\hat{\tau}_{t+1} &= \hat{\tau}_t - \eta_t \cdot \nabla_{\hat{\tau}_t} l_{1-\alpha}(\hat{\tau}_t, S_t) \\
&= \hat{\tau}_t + \eta_t \cdot (\mathbb{1}\{Y_t \notin \mathcal{C}_t(X_t)\} - \alpha)
\end{aligned} \tag{1}$$

where $\nabla_\tau l_{1-\alpha}(\tau, s)$ denotes the gradient of pinball loss w.r.t the threshold $\tau$, and $\eta_t > 0$ is the learning rate. The optimization will increase the threshold if the prediction set $\mathcal{C}_t(X_t)$ fails to encompass the label $Y_t$, resulting in more conservative predictions in future instances (and vice versa).

We use the *empirical coverage error* and the *expected coverage error* to evaluate the coverage performance. The

empirical coverage error measures the absolute deviation of the mis-coverage rate from the target level $\alpha$:

$$\text{EmErr(T)} = \left| \frac{1}{T} \sum_{t=1}^{T} \mathbb{1} \left\{ Y_t \notin \mathcal{C}_t(X_t) \right\} - \alpha \right|,$$

while the expected coverage error quantifies the absolute deviation in expectation:

$$\text{ExErr(T)} = \left| \frac{1}{T} \sum_{t=1}^{T} \mathbb{P} \left\{ Y_t \notin \mathcal{C}_t(X_t) \right\} - \alpha \right|.$$

In addition, online learning theory (see Theorem 2.13 of Orabona (2019)) established that standard online conformal prediction achieves a *regret* bound:

$$\begin{aligned}
\text{Reg}(T) &= \sum_{t=1}^{T} l_{1-\alpha}(\hat{\tau}_t, S_t) - \min_{\tau} \sum_{t=1}^{T} l_{1-\alpha}(\tau, S_t) \\
&= \mathcal{O}(T^{-\frac{1}{2}}),
\end{aligned}$$

with optimally chosen $\eta_t$, which serves as a helpful measure alongside coverage (Bhatnagar et al., 2023). We provide a regret analysis for our method in Appendix B.1.

**Label noise.** In this paper, we focus on the issue of noisy labels in online learning, a common occurrence in the real world. This is primarily due to the dynamic nature of real-time data streams and the potential for human error or sensor malfunctions during label collection. Let $(X_t, \tilde{Y}_t)$ be the data sequence with label noise, and $\tilde{S}_t = \mathcal{S}(X_t, \tilde{Y}_t)$ be the noisy non-conformity score. In this work, we focus on the setting of uniform label noise (Einbinder et al., 2024; Penso & Goldberger, 2024), i.e., the correct label is replaced by a label that is randomly sampled from the $K$ classes with a fixed probability $\epsilon \in (0, 1)$:

$$\tilde{Y}_t = Y_t \cdot \mathbb{1} \left\{ U \geq \epsilon \right\} + \bar{Y} \cdot \mathbb{1} \left\{ U \leq \epsilon \right\},$$

where $U$ is uniformly distributed over $[0, 1]$, and $\bar{Y}$ is uniformly sampled from the set of classes $\mathcal{Y}$. We assume the probability $\epsilon$ (i.e., the noise rate) is known, in alignment with prior works (Einbinder et al., 2024; Penso & Goldberger, 2024; Sesia et al., 2024). This assumption is practical as the noise rate can be estimated from historical data (Liu & Tao, 2015; Yu et al., 2018; Wei et al., 2020).

Recent work (Einbinder et al., 2024) investigates the noise robustness of online conformal prediction under uniform label noise, with a strong distributional assumption. Their analysis demonstrates that noisy labels will lead to a conservative long-run coverage guarantee, with the assumption that noisy score distribution stochastically dominates the clean score distribution, i.e., $\mathbb{P} \left\{ \tilde{S} \leq s \right\} \leq \mathbb{P} \left\{ S \leq s \right\}, \forall s \in \mathbb{R}$.

The distributional assumption is strong so that it only ensures valid coverage under limited cases. Moreover, their analysis fails to quantify the specific deviation of coverage guarantees. These limitations motivate us to establish a general theoretical framework for this problem and develop a noise-robust algorithm for online conformal prediction. We proceed by presenting the theoretical results on both constant (Gibbs & Candes, 2021) and dynamic (Angelopoulos et al., 2024) learning rate schedules, respectively.

## 3. Results for constant learning rate

### 3.1. The impact of label noise on prediction sets

In this section, we theoretically analyze the impacts of uniform label noise on online conformal prediction with a *constant* learning rate, i.e., $\eta_t \equiv \eta$, for all $t \in \mathbb{N}^+$. In this case, the threshold $\hat{\tau}_t$ is updated by

$$\hat{\tau}_{t+1} = \hat{\tau}_t - \eta \cdot \nabla_{\hat{\tau}_t} l_{1-\alpha}(\hat{\tau}_t, \tilde{S}_t), \tag{2}$$

where $\tilde{S}_t$ is the noisy non-conformity score. As shown in Eq. (2), the essence of online conformal prediction is to update the threshold $\hat{\tau}_t$ with the gradient of pinball loss. However, the gradient estimates can be biased if the observed labels are corrupted. Formally, with high probability:

$$\nabla_{\hat{\tau}_t} l_{1-\alpha}(\hat{\tau}_t, S_t) \neq \nabla_{\hat{\tau}_t} l_{1-\alpha}(\hat{\tau}_t, \tilde{S}_t).$$

This bias in gradient estimation can result in two potential consequences: the conformal predictor may either fail to maintain the desired coverage or suffer from reduced efficiency (i.e., generating large prediction sets). We formalize these consequences in the following proposition:

**Proposition 3.1.** *Consider online conformal prediction under uniform label noise with noise rate $\epsilon \in (0, 1)$. Given Assumptions 2.1 and 2.2, when updating the threshold according to Eq. (2), for any $\delta \in (0, 1)$ and $T \in \mathbb{N}^+$, the following bound holds with probability at least $1 - \delta$:*

$$\alpha - \frac{1}{T} \sum_{t=1}^{T} \mathbb{1} \left\{ Y_t \notin \mathcal{C}_t(X_t) \right\} \leq \frac{A}{\sqrt{T}} + \frac{B}{T}$$
$$+ \frac{\epsilon}{1-\epsilon} \cdot \frac{1}{T} \sum_{t=1}^{T} \left( (1-\alpha) - \frac{1}{K} \mathbb{E} \left[ |\mathcal{C}_t(X_t)| \right] \right),$$

*where*

$$A = \frac{2-\epsilon}{1-\epsilon} \sqrt{2 \log \left( \frac{4}{\delta} \right)} \quad and \quad B = \frac{1}{1-\epsilon} \left( \frac{1}{\eta} + 1 - \alpha \right).$$

**Interpretation.** The proof is provided in Appendix C.1. In Proposition 3.1, we evaluate the long-run mis-coverage rate using $\frac{1}{T} \sum_{t=1}^{T} \mathbb{1} \left\{ Y_t \notin \mathcal{C}_t(X_t) \right\}$. Our analysis shows

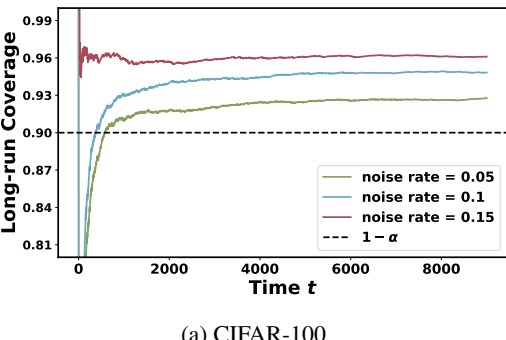

(a) CIFAR-100

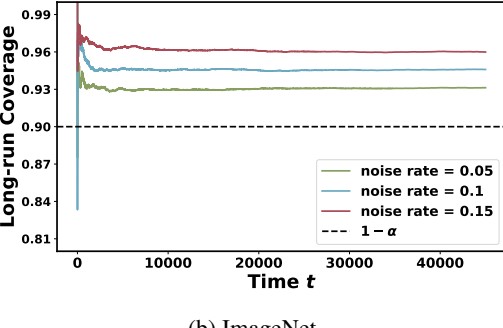

(b) ImageNet

Figure 1: Performance of standard online conformal prediction under different noise rates, with ResNet18 on CIFAR-100 and ImageNet. We use noisy labels to update the threshold with a constant learning rate $\eta = 0.05$.

that label noise introduces a coverage gap of $\frac{\epsilon}{1-\epsilon}$ · $\frac{1}{T} \sum_{t=1}^{T} \left( (1-\alpha) - \frac{1}{K} \mathbb{E}\left[ |\mathcal{C}_t(X_t)| \right] \right)$, between the actual mis-coverage rate and desired mis-coverage rate $\alpha$ (the term $\frac{A}{\sqrt{T}} + \frac{B}{T}$ will diminish and eventually approach zero, as $T$ increases). In particular, the effect of label noise on the coverage guarantee can manifest in two distinct scenarios, depending on the prediction set size:

- When the prediction sets are small such that $\mathbb{E}\left[ |\mathcal{C}_t(X_t)| \right] \leq K(1-\alpha)$, label noise can result in *over-coverage* of prediction sets: $\frac{1}{T} \sum_{t=1}^{T} \mathbb{1}\left\{ Y_t \notin \mathcal{C}_t(X_t) \right\} \leq \alpha$. In this scenario, a higher noise rate $\epsilon$ results in a larger coverage gap.

- When the prediction sets are large such that $\mathbb{E}\left[ |\mathcal{C}_t(X_t)| \right] \geq K(1-\alpha)$, label noise causes *under-coverage* of prediction sets. This situation generally occurs only when the model significantly underperforms on the task, which is uncommon in practical applications (See details in Appendix B.3)).

**Empirical verification.** We compare the performance of online conformal prediction under different noise rates $\epsilon = 0.05, 0.1, 0.15$, with a ResNet18 model on CIFAR-100 and ImageNet datasets. We employ LAC score (Sadinle et al., 2019) to generate prediction sets, and use noisy labels to update the threshold with a constant learning rate $\eta = 0.05$. The experimental results in Figure 1 validate our theoretical analysis: label noise introduces discrepancies between the actual and target coverage rates $1 - \alpha$, with higher noise rates resulting in a more pronounced coverage gap.

Overall, our results show that label noise significantly impacts the coverage guarantee of online conformal prediction. The size of the prediction set determines whether coverage is inflated or deflated, and the noise rate controls how much coverage is changed. In the following, we introduce a robust pinball loss, which addresses the issue of label noise.

### 3.2. Method

Our theoretical analysis establishes that biased gradient estimates arising from label noise can significantly impact the coverage properties of online conformal prediction. Therefore, the key challenge of the noisy setting lies in how to obtain unbiased gradient estimates without requiring ground-truth labels. In this work, we propose a novel algorithm - *Noise Robust Online Conformal Prediction* (dubbed **NR-OCP**), which updates the threshold with *robust pinball loss*. We begin by developing the intuition behind approximating clean pinball loss under uniform label noise.

Consider a data sample $(X, Y)$ with a noisy label $\tilde{Y}$, we denote the clean non-conformity score as $S = \mathcal{S}(X, Y)$, the noisy score as $\tilde{S} = \mathcal{S}(X, \tilde{Y})$, and the score for an arbitrary class $y \in \mathcal{Y}$ as $S_y = \mathcal{S}(X, y)$. Under a uniform label noise with noise rate $\epsilon \in (0, 1)$, the distributions of these scores have the following relationship: $\mathbb{P}\{S \leq s\} = \frac{1}{1-\epsilon}\mathbb{P}\left\{ \tilde{S} \leq s \right\} - \frac{\epsilon}{K(1-\epsilon)} \sum_{y=1}^{K} \mathbb{P}\{S_y \leq s\}$, for an arbitrary number $s \in \mathbb{R}$. We formally establish this equation in Lemma G.1 with a rigorous proof. This correlation motivates the following approximation:

$$\mathbb{1}\{S \leq s\} \approx \frac{1}{1-\epsilon}\mathbb{1}\left\{ \tilde{S} \leq s \right\} - \frac{\epsilon}{K(1-\epsilon)} \sum_{y=1}^{K} \mathbb{1}\{S_y \leq s\}.$$

**Robust pinball loss.** The above decomposition suggests that the clean pinball loss can be approximated by replacing its indicator function with the above expression. Inspired by this, we propose the *robust pinball loss* as:

$$\tilde{l}_{1-\alpha}(\tau, \tilde{S}, \{S_y\}_{y=1}^{K}) = l_1(\tau, \tilde{S}) - l_2(\tau, \{S_y\}_{y=1}^{K}), \quad (3)$$

where

$$l_1(\tau, \tilde{S}) = \frac{1}{1-\epsilon}l_{1-\alpha}(\tau, \tilde{S}),$$

$$l_2(\tau, \{S_y\}_{y=1}^{K}) = \frac{\epsilon}{K(1-\epsilon)} \sum_{y=1}^{K} l_{1-\alpha}(\tau, S_y).$$

The following theoretical properties demonstrate how this loss function mitigates label noise bias:

**Proposition 3.2.** *The robust pinball loss defined in Eq.* (3) *satisfies the following two properties:*

$$(1)\mathbb{E}_S\left[l_{1-\alpha}(\tau, S)\right] = \mathbb{E}_{\tilde{S}, S_y}\left[\tilde{l}_{1-\alpha}(\tau, \tilde{S}, \{S_y\}_{y=1}^K)\right],$$

$$(2)\mathbb{E}_S\left[\nabla_\tau l_{1-\alpha}(\tau, S)\right] = \mathbb{E}_{\tilde{S}, S_y}\left[\nabla_\tau \tilde{l}_{1-\alpha}(\tau, \tilde{S}, \{S_y\}_{y=1}^K)\right].$$

**Interpretation.** The proof can be found in Appendix C.2. In Proposition 3.2, the first property ensures that our robust pinball loss matches the expected value of the true pinball loss, while the second guarantees that the gradients of both losses have the same expectation. These properties establish that updating the threshold with robust pinball loss is equivalent to updating with clean pinball loss in expectation.

With the robust pinball loss, the proposed NR-OCP updates the threshold as follows:

$$\begin{aligned}
\hat{\tau}_{t+1} &= \hat{\tau}_t - \eta \cdot \nabla_{\hat{\tau}_t} \tilde{l}_{1-\alpha}(\hat{\tau}_t, \tilde{S}_t, \{S_{t,y}\}_{y=1}^K) \\
&= \hat{\tau}_t - \eta \cdot \nabla_{\hat{\tau}_t} l_1(\hat{\tau}_t, \tilde{S}_t) + \eta \cdot \nabla_{\hat{\tau}_t} l_2(\hat{\tau}_t, \{S_{t,y}\}_{y=1}^K),
\end{aligned}$$
(4)

where

$$\nabla_{\hat{\tau}_t} l_1(\hat{\tau}_t, \tilde{S}_t) = \frac{1}{1-\epsilon}\left[\mathbb{1}\left\{\tilde{S}_t \leq \hat{\tau}_t\right\} - (1-\alpha)\right],$$

$$\nabla_{\hat{\tau}_t} l_2(\hat{\tau}_t, S_{t,y}) = \frac{\epsilon}{K(1-\epsilon)}\sum_{y=1}^K \left[\mathbb{1}\left\{S_{t,y} \leq \hat{\tau}_t\right\} - (1-\alpha)\right].$$

In summary, Proposition 3.2 establishes the validity of our gradient estimates *in expectation*, laying the foundation for our subsequent convergence analysis. In the following sections, we present a detailed theoretical analysis that demonstrates how these results translate into finite-sample coverage guarantees, in the presence of label noise.

### 3.3. Convergence of coverage rate

We now analyze the convergence of the coverage rate of NR-OCP under uniform label noise with a constant learning rate schedule. For notation shorthand, we denote:

$$\mathbb{E}\left[\nabla_{\hat{\tau}_t} l_{1-\alpha}\right] = \mathbb{E}_{S_t}\left[\nabla_{\hat{\tau}_t} l_{1-\alpha}(\hat{\tau}_t, S_t)\right],$$

$$\mathbb{E}\left[\nabla_{\hat{\tau}_t} \tilde{l}_{1-\alpha}\right] = \mathbb{E}_{\tilde{S}_t, S_{t,y}}\left[\nabla_{\hat{\tau}_t} \tilde{l}_{1-\alpha}(\hat{\tau}_t, \tilde{S}_t, \{S_{t,y}\}_{y=1}^K)\right].$$

We first present the results for expected coverage error:

**Proposition 3.3.** *Consider online conformal prediction under uniform label noise with noise rate $\epsilon \in (0,1)$. Given Assumptions 2.1 and 2.2, when updating the threshold according to Eq.* (4), *for any $\delta \in (0,1)$ and $T \in \mathbb{N}^+$, the following bound holds with probability at least $1-\delta$:*

$$\text{ExErr(T)} \leq \sqrt{\frac{\log(2/\delta)}{1-\epsilon}} \cdot \frac{1}{\sqrt{T}} + \left(\frac{1}{\eta} - \alpha + \frac{\epsilon}{1-\epsilon}\right) \cdot \frac{1}{T}.$$

**Proof Sketch.** The proof of Proposition 3.3 (in Appendix C.3) relies on the following decomposition:

$$\begin{aligned}
\text{ExErr(T)} &= \left|\sum_{t=1}^T \mathbb{E}\left[\nabla_{\hat{\tau}_t} l_{1-\alpha}\right]\right| = \left|\sum_{t=1}^T \mathbb{E}\left[\nabla_{\hat{\tau}_t} \tilde{l}_{1-\alpha}\right]\right| \\
&\leq \underbrace{\left|\sum_{t=1}^T \mathbb{E}\left[\nabla_{\hat{\tau}_t} \tilde{l}_{1-\alpha}\right] - \sum_{t=1}^T \nabla_{\hat{\tau}_t} \tilde{l}_{1-\alpha}\right|}_{(a)} + \underbrace{\left|\sum_{t=1}^T \nabla_{\hat{\tau}_t} \tilde{l}_{1-\alpha}\right|}_{(b)}.
\end{aligned}$$

Part (a) converges to zero in probability at rate $\mathcal{O}(T^{-1/2})$ by the Azuma–Hoeffding inequality, and part (b) achieves a convergence rate of $\mathcal{O}(T^{-1})$ following standard online conformal prediction theory. Combining the two parts establishes the desired upper bound.

Building on Proposition 3.3, we now provide an upper bound for empirical coverage error for NR-OCP:

**Proposition 3.4.** *Consider online conformal prediction under uniform label noise with noise rate $\epsilon \in (0,1)$. Given Assumptions 2.1 and 2.2, when updating the threshold according to Eq.* (4), *for any $\delta \in (0,1)$ and $T \in \mathbb{N}^+$, the following bound holds with probability at least $1-\delta$:*

$$\begin{aligned}
\text{EmErr(T)} \leq \; &\frac{2-\epsilon}{1-\epsilon}\sqrt{2\log\left(\frac{4}{\delta}\right)} \cdot \frac{1}{\sqrt{T}} + \\
&\left(\frac{1}{\eta} - \alpha + \frac{\epsilon}{1-\epsilon}\right) \cdot \frac{1}{T}.
\end{aligned}$$

**Proof Sketch.** The proof of Proposition 3.4 (detailed in Appendix C.4) employs a similar decomposition:

$$\begin{aligned}
\text{EmErr(T)} &= \left|\sum_{t=1}^T \nabla_{\hat{\tau}_t} l_{1-\alpha}\right| \\
&\leq \underbrace{\left|\sum_{t=1}^T \nabla_{\hat{\tau}_t} l_{1-\alpha} - \sum_{t=1}^T \mathbb{E}\left[\nabla_{\hat{\tau}_t} l_{1-\alpha}\right]\right|}_{(a)} + \underbrace{\left|\sum_{t=1}^T \mathbb{E}\left[\nabla_{\hat{\tau}_t} l_{1-\alpha}\right]\right|}_{(b)}.
\end{aligned}$$

The analysis shows that both terms achieve $\mathcal{O}(T^{-1/2})$ convergence: part (a) through the Azuma–Hoeffding inequality, and part (b) via Proposition 3.3. Combining the two parts yields the desired upper bound.

**Remark 3.5.** It is worth noting that our method achieves a $\mathcal{O}(T^{-1/2})$ convergence rate for empirical coverage error even in the absence of noise (i.e., $\epsilon = 0$), which is slightly slower than the $\mathcal{O}(T^{-1})$ rate achieved by standard online conformal prediction theory (Gibbs & Candes, 2021; Angelopoulos et al., 2024). This is because our analysis relies on martingale-based concentration to handle label noise, leading to the $\mathcal{O}(T^{-1/2})$ rate.

# 4. Results for dynamic learning rate

## 4.1. Theoretical results

Recent work (Angelopoulos et al., 2024) highlights a limitation of constant learning rates: while coverage holds on average over time, the *instantaneous* coverage rate $\text{Cov}(\hat{\tau}_t) = \mathbb{P}\{S \leq \hat{\tau}_t\}$ would exhibit substantial temporal variability (see Proposition 1 in Angelopoulos et al. (2024)). Thus, they propose using *dynamic* learning rates where $\eta_t$ can change over time for updating the threshold.

We analyze this setting in the presence of label noise, specifically examining the case where noisy labels update the conformal threshold with a time-varying learning rate:

$$\hat{\tau}_{t+1} = \hat{\tau}_t - \eta_t \cdot \nabla_{\hat{\tau}_t} l_{1-\alpha}(\hat{\tau}_t, \tilde{S}_t). \tag{5}$$

For notational simplicity, we denote $\eta_0^{-1} = 0$. Then, we have the following proposition analogous to Proposition 3.1:

**Proposition 4.1.** *Consider online conformal prediction under uniform label noise with noise rate $\epsilon \in (0,1)$. Given Assumptions 2.1 and 2.2, when updating the threshold according to Eq. (5), for any $\delta \in (0,1)$ and $T \in \mathbb{N}^+$, the following bound holds with probability at least $1-\delta$:*

$$\alpha - \frac{1}{T}\sum_{t=1}^{T}\mathbb{1}\{Y_t \notin \mathcal{C}_t(X_t)\} \leq \frac{A}{\sqrt{T}} + \frac{B}{T}$$

$$+ \frac{\epsilon}{1-\epsilon} \cdot \frac{1}{T}\sum_{t=1}^{T}\left((1-\alpha) - \frac{1}{K}\mathbb{E}\left[|\mathcal{C}_t(X_t)|\right]\right)$$

*where*

$$A = \frac{2-\epsilon}{1-\epsilon}\sqrt{2\log\left(\frac{4}{\delta}\right)};$$

$$B = \frac{1 + \max\limits_{1\leq t \leq T-1}\eta_t}{1-\epsilon}\sum_{t=1}^{T}\left|\eta_t^{-1} - \eta_{t-1}^{-1}\right|.$$

**Interpretation.** The proof is detailed in Appendix D.1. Similar to the findings in Proposition 3.1, this result demonstrates that label noise induces a coverage gap given by: $\frac{\epsilon}{1-\epsilon} \cdot \frac{1}{T}\sum_{t=1}^{T}\left((1-\alpha) - \frac{1}{K}\mathbb{E}\left[|\mathcal{C}_t(X_t)|\right]\right)$, which quantifies the discrepancy between the actual mis-coverage rate and the target rate $\alpha$. The primary distinction from Proposition 3.1 lies in the constant $B$.

**Remark 4.2.** Proposition 4.1 generalizes our previous analysis: the result of constant learning rate in Proposition 3.1 emerges as a special case when $\eta_t \equiv \eta$ for all $t \in \mathbb{N}^+$. More broadly, this result holds for any sequence of learning rates that satisfies the condition $\sum_{t=1}^{T}\left|\eta_t^{-1} - \eta_{t-1}^{-1}\right|/T \to 0$ as $T \to +\infty$. Additionally, empirical evidence corroborating our theory can be found in Appendix B.5.

## 4.2. Method

Our analysis establishes that label noise significantly impacts coverage rate regardless of the learning rate schedule. Building on these insights, we extend our method - NR-OCP to the case of dynamic learning rates. Specifically, we update the threshold by:

$$\begin{aligned}\hat{\tau}_{t+1} &= \hat{\tau}_t - \eta_t \cdot \nabla_{\hat{\tau}_t}\tilde{l}_{1-\alpha}(\hat{\tau}_t, \tilde{S}_t, \{S_{t,y}\}_{y=1}^{K})\\ &= \hat{\tau}_t - \eta_t \cdot \nabla_{\hat{\tau}_t}l_1(\hat{\tau}_t, \tilde{S}_t) + \eta_t \cdot \nabla_{\hat{\tau}_t}l_2(\hat{\tau}_t, \tilde{S}_t),\end{aligned} \tag{6}$$

where

$$\nabla_{\hat{\tau}_t}l_1(\hat{\tau}_t, \tilde{S}_t) = \frac{1}{1-\epsilon}\left[\mathbb{1}\left\{\tilde{S}_t \leq \hat{\tau}_t\right\} - (1-\alpha)\right];$$

$$\nabla_{\hat{\tau}_t}l_2(\hat{\tau}_t, S_{t,y}) = \frac{\epsilon}{K(1-\epsilon)}\sum_{y=1}^{K}[\mathbb{1}\{S_{t,y} \leq \hat{\tau}_t\} - (1-\alpha)].$$

For the convergence of coverage rate, our theoretical results show that, under dynamic learning rate, NR-OCP achieves convergence rates of $\text{EmCovErr}(T) = \mathcal{O}(T^{-1/2})$ and $\text{ExCovErr}(T) = \mathcal{O}(T^{-1/2})$. The proofs are presented in Appendix D.2 and D.3.

**Proposition 4.3.** *Consider online conformal prediction under uniform label noise with noise rate $\epsilon \in (0,1)$. Given Assumptions 2.1 and 2.2, when updating the threshold according to Eq. (6), for any $\delta \in (0,1)$ and $T \in \mathbb{N}^+$, the following bound holds with probability at least $1-\delta$:*

$$\text{ExErr}(T) \leq \sqrt{\frac{\log(2/\delta)}{1-\epsilon}} \cdot \frac{1}{\sqrt{T}} + \frac{C}{T}$$

*where*

$$C = \left[\left(1 + \max\limits_{1\leq t \leq T-1}\eta_t \cdot \frac{1+\epsilon}{1-\epsilon}\right)\sum_{t=1}^{T}\left|\eta_t^{-1} - \eta_{t-1}^{-1}\right|\right].$$

**Proposition 4.4.** *Consider online conformal prediction under uniform label noise with noise rate $\epsilon \in (0,1)$. Given Assumptions 2.1 and 2.2, when updating the threshold according to Eq. (6), for any $\delta \in (0,1)$ and $T \in \mathbb{N}^+$, the following bound holds with probability at least $1-\delta$:*

$$\text{EmErr}(T) \leq \frac{2-\epsilon}{1-\epsilon}\sqrt{2\log\left(\frac{4}{\delta}\right)} \cdot \frac{1}{\sqrt{T}} + \frac{\gamma}{T}$$

*where*

$$\gamma = \left[\left(1 + \max\limits_{1\leq t \leq T-1}\eta_t \cdot \frac{1+\epsilon}{1-\epsilon}\right)\sum_{t=1}^{T}\left|\eta_t^{-1} - \eta_{t-1}^{-1}\right|\right]$$

*with at least $1-\delta$ probability.*

In Proposition 4.3 and 4.4, we establish that for any sequence of learning rates that satisfies the condition

Table 1: Average performance of different methods under uniform noisy labels across 4 score functions, using ResNet18. The performance on each score function is provided in Appendix B.6. "Baseline" denotes the standard online conformal prediction methods. We include two learning rate schedules: constant learning rate $\eta = 0.05$ and dynamic learning rates $\eta_t = 1/t^{1/2+\varepsilon}$ where $\varepsilon = 0.1$. "↓" indicates smaller values are better and **Bold** numbers are superior results.

| LR Schedule | Error rate | Method | CIFAR100 | | | | ImageNet | | | |
| | | | CovGap(%) ↓ | | Size ↓ | | CovGap(%) ↓ | | Size ↓ | |
| | | | $\alpha = 0.1$ | $\alpha = 0.05$ | $\alpha = 0.1$ | $\alpha = 0.05$ | $\alpha = 0.1$ | $\alpha = 0.05$ | $\alpha = 0.1$ | $\alpha = 0.05$ |
| Constant | $\epsilon = 0.05$ | Baseline | 3.942 | 2.867 | 11.93 | 30.66 | 4.163 | 3.289 | 101.0 | 231.7 |
| | | Ours | **0.386** | **0.183** | **7.786** | **16.54** | **0.084** | **0.272** | **67.37** | **143.3** |
| | $\epsilon = 0.1$ | Baseline | 7.139 | 3.753 | 21.81 | 46.93 | 7.509 | 4.068 | 178.4 | 389.7 |
| | | Ours | **0.270** | **0.428** | **8.815** | **17.96** | **0.095** | **0.194** | **81.87** | **157.3** |
| | $\epsilon = 0.15$ | Baseline | 8.032 | 4.147 | 36.95 | 59.92 | 8.566 | 4.348 | 318.0 | 513.2 |
| | | Ours | **0.520** | **0.395** | **9.396** | **19.41** | **0.198** | **0.144** | **90.52** | **163.4** |
| Dynamic | $\epsilon = 0.05$ | Baseline | 4.106 | 2.780 | 6.527 | 21.97 | 4.498 | 2.584 | 31.15 | 128.1 |
| | | Ours | **0.170** | **0.658** | **2.958** | **10.95** | **0.120** | **0.242** | **7.502** | **47.41** |
| | $\epsilon = 0.1$ | Baseline | 7.414 | 3.733 | 18.18 | 37.18 | 7.249 | 3.595 | 99.54 | 214.9 |
| | | Ours | **0.414** | **0.217** | **3.361** | **12.01** | **0.079** | **0.321** | **10.34** | **67.03** |
| | $\epsilon = 0.15$ | Baseline | 8.403 | 4.211 | 29.41 | 48.24 | 8.372 | 4.127 | 171.2 | 274.9 |
| | | Ours | **0.214** | **0.195** | **4.065** | **12.98** | **0.183** | **0.256** | **13.10** | **78.64** |

$\sum_{t=1}^{T} \left| \eta_t^{-1} - \eta_{t-1}^{-1} \right| / T \to 0$ as $T \to +\infty$, both expected and empirical coverage errors asymptotically vanish with a convergence rate of $\mathcal{O}(T^{-1/2})$. Therefore, by applying our method, the long-term coverage rate would approach the desired target level of $1 - \alpha$. In addition, we formally establish the convergence of $\hat{\tau}_t$ towards the global minima in Appendix B.2 by extending the standard convergence analysis of stochastic gradient descent.

## 5. Experiments

**Datasets and setup.** We evaluate the performance of NR-OCP under uniform label noise. The experiments include both constant $\eta = 0.05$ and dynamic learning rates $\eta_t = 1/t^{1/2+\varepsilon}$ with $\varepsilon = 0.1$, following prior work (Angelopoulos et al., 2024)). We use CIFAR-100 (Krizhevsky, 2009) and ImageNet (Deng et al., 2009) dataset with synthetic label noise. On ImageNet, we use four pre-trained classifiers from TorchVision (Paszke et al., 2019) - ResNet18, ResNet50 (He et al., 2016), DenseNet121 (Huang et al., 2017) and VGG16 (Simonyan & Zisserman, 2015). On CIFAR-100, we train these models for 200 epochs using SGD with a momentum of 0.9, a weight decay of 0.0005, and a batch size of 128. We set the initial learning rate as 0.1, and reduce it by a factor of 5 at 60, 120 and 160 epochs.

**Conformal prediction algorithms.** We apply LAC (Sadinle et al., 2019), APS (Romano et al., 2020b), RAPS (Angelopoulos et al., 2021) and SAPS (Huang et al., 2023)

to generate prediction sets. A detailed description of these non-conformity scores is provided in Appendix B.4. For the evaluation metrics, we employ the *long-run coverage gap (CovGap)* and *long-run size (Size)*. Formally, given a test dataset $\mathcal{I}_{test}$, these two metrics are defined as:

$$\text{CovGap} = \left| \frac{1}{|\mathcal{I}_{test}|} \sum_{t \in \mathcal{I}_{test}} \mathbb{1}\left\{Y_t \in \mathcal{C}_t(X_t)\right\} - (1 - \alpha) \right|,$$

$$\text{Size} = \frac{1}{|\mathcal{I}_{test}|} \sum_{t \in \mathcal{I}_{test}} |\mathcal{C}_t(X_t)|.$$

The long-run size measures the efficiency of prediction sets. Small prediction sets are preferred as they can provide precise predictions, thereby enabling accurate human decision-making in real-world scenarios (Cresswell et al., 2024).

**Main results.** In Table 1, we evaluate NR-OCP (Ours) against the standard online conformal prediction (Baseline) that updates the threshold with noisy labels (see Eq. (2) and Eq. (5)), employing a ResNet18 model. Due to space constraints, we report the average performance across four non-conformity score functions. The performance on each score function is provided in Appendix B.6. A salient observation is that NR-OCP eliminates the long-run coverage gap present in the baseline, achieving precise coverage guarantees while significantly improving the long-run efficiency of prediction sets. For example, on ImageNet with error rate $\alpha = 0.1$, noise rate $\epsilon = 0.15$ and dynamic learning rate, the baseline deviates from the target coverage level of

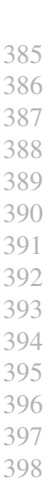
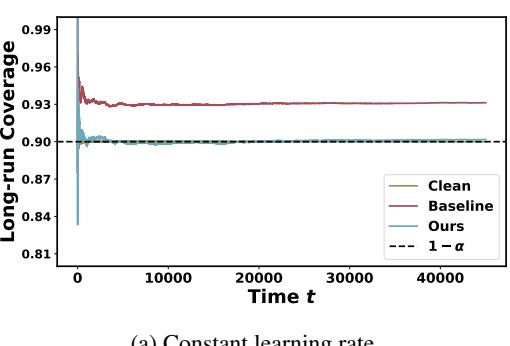
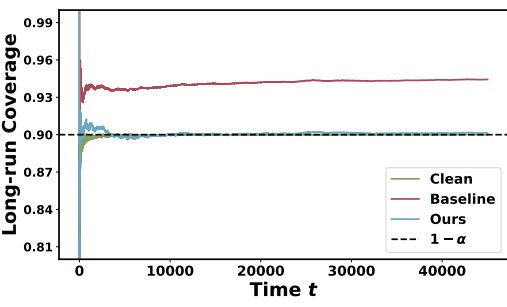

(a) Constant learning rate
(b) Dynamic learning rate

Figure 2: Performance comparison of various methods under uniform noisy labels with noise rate $\epsilon = 0.05$, employing (a) constant learning rate $\eta = 0.05$ and (b) dynamic learning rates $\eta_t = 1/t^{1/2+\varepsilon}$ where $\varepsilon = 0.1$. "Baseline" and "Clean" denote the standard online conformal prediction with noisy and clean labels, respectively. We employ LAC scores to generate prediction sets with error rate $\alpha = 0.1$, using ResNet18 on ImageNet.

Table 2: Performance of different methods using different models under uniform label noise with noise rate $\epsilon = 0.05$ on CIFAR-100. "Baseline" denotes the standard online conformal prediction method. We employ LAC score to generate prediction sets with error rate $\alpha = 0.1$ and use a constant learning rate $\eta = 0.05$. "↓" indicates smaller values are better and **Bold** numbers are superior results.

| Models | CovGap(%) ↓ | | Size ↓ | |
| | $\alpha = 0.1$ | $\alpha = 0.05$ | $\alpha = 0.1$ | $\alpha = 0.05$ |
| --- | --- | --- | --- | --- |
| | Baseline / Ours | | | |
| ResNet18 | 2.744 / **0.289** | 1.900 / **0.122** | 31.61 / **22.78** | 56.84 / **47.31** |
| ResNet50 | 2.700 / **0.378** | 1.389 / **0.289** | 33.23 / **19.21** | 59.60 / **50.24** |
| DenseNet121 | 3.178 / **0.156** | 1.933 / **0.211** | 25.88 / **15.20** | 52.93 / **43.70** |
| VGG16 | 1.833 / **0.411** | 1.011 / **0.067** | 52.33 / **42.11** | 72.69 / **68.03** |

0.9, exhibiting a coverage gap of 8.372% and an average set size of 171.2. In contrast, NR-OCP achieves a negligible coverage gap of 0.183% and a prediction set size of 13.10.

Moreover, following prior work (Angelopoulos et al., 2024), we demonstrate the dynamics of coverage rate in Figure 2. We use the LAC score and a ResNet18 model on ImageNet, with error rate $\alpha = 0.1$ and noise rate $\epsilon = 0.05$. The results show that NR-OCP successfully converges to the desired coverage rate of $1-\alpha$ on both constant and dynamic learning rates. Overall, NR-OCP demonstrates superior performance over the baseline by achieving both precise coverage guarantee and improved prediction set efficiency.

**NR-OCP is effective on different model architectures.** In Table 2, we show that NR-OCP is effective on a diverse range of model architectures. The experiments are conducted on CIFAR-100, using LAC score to generate prediction sets with error rate $\alpha = 0.1$ and noise rate $\epsilon = 0.05$, under constant learning rate schedule. In particular, our method consistently outperforms the baseline across various models. For example, on DenseNet121 with error rate

$\alpha = 0.1$, the baseline exhibits a coverage gap of 3.178% and size of 25.88, while NR-OCP achieves a smaller coverage gap of 0.156% and a reduced size of 15.20. Additional results on these models is available in Appendix B.7.

## 6. Conclusion

In this work, we investigate the robustness of online conformal prediction under uniform label noise with a known noise rate, in both constant and dynamic learning rate schedule. Our theoretical analysis shows that the presence of label noise causes a deviation between the actual and desired mis-coverage rate $\alpha$, with higher noise rates resulting in larger gaps. This gap can lead to either overestimated or underestimated coverage guarantees, which depend on the size of the prediction sets. To address this issue, we propose *Noise Robust Online Conformal Prediction* (dubbed NR-OCP) which updates the threshold using the proposed *robust pinball loss*. We establish that this loss is equivalent to the pinball loss under clean labels in expectation. We prove that NR-OCP eliminates the coverage gap caused by the label noise, in both constant and dynamic learning rate schedules, achieving a convergence rate of $\mathcal{O}(T^{-1/2})$ for both empirical and expected coverage errors under the uniform label noise. Extensive experiments verify that our method achieves superior performance over the baseline by achieving both precise coverage rates and improved set efficiency across various settings, including various model architectures, error rates, noise rates, and non-conformity scores. We hope the theoretical framework in this work can inspire more specifically designed methods for online conformal prediction under label noise.

**Limitation.** In this work, our analysis and method are limited to the setting of uniform noisy labels with a known noise rate. We believe it will be interesting to develop online conformal prediction algorithms that are robust to various types of label noise, in the future.

## Impact Statements

This paper presents work whose goal is to advance the field of Machine Learning. There are many potential societal consequences of our work, none of which we feel must be specifically highlighted here.

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

## A. Related work

Conformal prediction (Papadopoulos et al., 2002; Vovk et al., 2005) is a statistical framework for uncertainty qualification. In the literature, many conformal prediction methods have been proposed across various domains, such as regression (Lei & Wasserman, 2014; Romano et al., 2020a), image classification (Angelopoulos et al., 2021; Huang et al., 2024; Liu et al., 2024), outlier detection (Guan & Tibshirani, 2022; Bates et al., 2023; Liang et al., 2024), and large language models (Gui et al., 2024; Cherian et al., 2024). Conformal prediction is also deployed in other real-world applications, such as human-in-the-loop decision-making (Cresswell et al., 2024), automated vehicles (Bang et al., 2024), and scientific computation (Moya et al., 2024). In what follows, we introduce the most related works in two settings: online learning and noise robustness.

**Online conformal prediction.** Conventional conformal prediction algorithms provide coverage guarantees under the assumption of data exchangeability. However, in real-world online scenarios, the data distribution may evolve over time, violating the exchangeability assumption (Zhang et al., 2018; Zhao et al., 2022). To address this challenge, recent research develops online conformal prediction methods that adaptively construct prediction sets to accommodate distribution shift (Gibbs & Candes, 2021; Feldman et al., 2023; Bhatnagar et al., 2023; Angelopoulos et al., 2024; Gibbs & Candès, 2024; Hajihashemi & Shen, 2024). Building on online convex optimization techniques (Anderson, 2008; Moore et al., 2011; Hazan et al., 2016; Singh & Thurman, 2019; Hoi et al., 2021; Orabona, 2019), these methods employ online gradient descent with pinball loss to provably achieve desired coverage under arbitrary distributional changes (Gibbs & Candes, 2021). Still, these algorithms typically assume perfect label accuracy, an assumption that rarely holds in practice, given the prevalence of noisy labels in online learning (Ben-David et al., 2009; Natarajan et al., 2013; Wu et al., 2024). In this work, we theoretically show that label noise can significantly affect the long-run mis-coverage rate through biased pinball loss gradients, leading to either inflated or deflated coverage guarantee.

**Noise-robust conformal prediction.** The issue of label noise has been a common challenge in machine learning with extensive studies (Xia et al., 2019; Wei et al., 2020; Chen et al., 2021; Wei et al., 2021; Wu et al., 2021; Li et al., 2021; Zhu et al., 2022; Wei et al., 2023; Chen et al., 2024; Gao et al., 2024). In the context of conformal prediction, recent works develop noise-robust conformal prediction algorithms for both uniform noise (Penso & Goldberger, 2024) and noise transition matrix (Sesia et al., 2024). The most relevant work (Einbinder et al., 2024) shows that online conformal prediction maintains valid coverage when noisy scores stochastically dominate clean scores. Our analysis extends this work by removing the assumption on noisy and clean scores, offering a more general theoretical framework for understanding the impact of uniform label noise.

## B. Additional results

### B.1. Regret analysis

As Bhatnagar et al. (2023) demonstrates, regret serves as a helpful performance measure alongside coverage. In particular, it can identify algorithms that achieve valid coverage guarantees through impractical means. For example, prediction sets that alternate between empty and full sets with frequencies $\{\alpha, 1 - \alpha\}$ satisfy coverage bounds on any distribution but have linear regret on simple distributions (see detailed proof in Appendix A.2 of Bhatnagar et al. (2023)). Drawing from standard online learning theory (see Theorem 2.13. of Orabona (2019)), we analyze the regret bound for our method. Let $\tau^* := \arg\min_{\hat{\tau}} \sum_{t=1}^{T} \tilde{l}_{1-\alpha}(\tau^*, \tilde{S}_t, \{S_{t,y}\}_{y=1}^K)$, and define the regret as

$$\text{Reg}(T) = \sum_{t=1}^{T} \left( \tilde{l}_{1-\alpha}(\hat{\tau}_t, \tilde{S}_t, \{S_{t,y}\}_{y=1}^K) - \tilde{l}_{1-\alpha}(\tau^*, \tilde{S}_t, \{S_{t,y}\}_{y=1}^K) \right).$$

This leads to the following regret bound (the proof is provided in Appendix E):

**Proposition B.1.** *Consider online conformal prediction under uniform label noise with noise rate $\epsilon \in (0, 1)$. Given Assumptions 2.1 and 2.2, when updating the threshold according to Eq. (6), for any $T \in \mathbb{N}^+$, we have:*

$$\text{Reg}(T) \leq \frac{1}{2\eta_T} \left( 1 + \max_{1 \leq t \leq T-1} \eta_t \cdot \frac{1+\epsilon}{1-\epsilon} \right)^2 + \left( \frac{1+\epsilon}{1-\epsilon} \right)^2 \cdot \sum_{t=1}^{T} \frac{\eta_t}{2}.$$

**Example 1: constant learning rate.** With a constant learning rate $\eta_t \equiv \eta$, our method yields the following regret bound:

$$\text{Reg}(T) \leq \frac{1}{\eta}\left(1 + \eta \cdot \frac{1+\epsilon}{1-\epsilon}\right)^2 + \left(\frac{1+\epsilon}{1-\epsilon}\right)^2 \cdot \frac{T\eta}{2}.$$

This linear regret aligns with the observation in Angelopoulos et al. (2024): the constant learning rate schedule, while providing valid coverage on average, lead to significant temporal variability in Coverage($\hat{\tau}_t$) (see Proposition 1 in Angelopoulos et al. (2024)). The linear regret bound provides additional theoretical justification for this instability.

**Example 2: decaying learning rate.** Suppose the proposed method updates the threshold with a decaying learning rate schedule: $\eta_t = (1 - \epsilon/1 + \epsilon + \eta_0) \cdot \sqrt{t}$, where $\eta_0 \in \mathbb{R}$. The inequality $\sum_{t=1}^{T} 1/\sqrt{t} \leq 2\sqrt{T}$ follows that

$$\text{Reg}(T) \leq 2\left[\frac{1+\epsilon}{1-\epsilon} + \eta_0 \cdot \left(\frac{1+\epsilon}{1-\epsilon}\right)^2\right] \cdot \sqrt{T}$$

This sublinear regret bound $\mathcal{O}(\sqrt{T})$ implies that the decaying learning rate schedule achieves superior convergence compared to the linear regret of constant learning rates.

### B.2. Convergence analysis of $\hat{\tau}_t$

We analyze the convergence of our method toward global minima by combining the convergence analysis of SGD (see Theorem 5.3. in Garrigos & Gower (2023)) and self-calibration inequality of pinball loss (see Theorem 2.7. in Steinwart & Christmann (2011)). Following Steinwart & Christmann (2011), we first establish assumptions on the distribution of $S$. Let $R := 2S - 1 \in [0, 1]$, with $\gamma$ denoting the $1 - \alpha$ quantile of $R$. This indicates that $\gamma = 2\tau - 1$, where $\tau$ is the $1 - \alpha$ quantile of $S$. We make the following assumption:

**Assumption B.2.** There exists constants $b > 0$, $q \geq 2$, and $\varepsilon_0 > 0$ such that $\mathbb{P}\{R = \hat{\gamma}\} \geq b|\hat{\gamma} - \gamma|^{q-2}$ holds for all $\hat{\tau} \in [\tau - \varepsilon_0, \tau + \varepsilon_0]$.

Furthermore, let $\beta = b/(q - 1)$ and $\delta = \beta(2\varepsilon_0)^{q-1}$. This leads to the following proposition (the proof is provided in Appendix F):

**Proposition B.3.** *Consider online conformal prediction under uniform label noise with noise rate $\epsilon \in (0, 1)$. Given Assumptions 2.1, 2.2 and B.2, when updating the threshold according to Eq. (6), for any $T \in \mathbb{N}^+$, we have:*

$$|\bar{\tau} - \tau|^q \leq \frac{q(1-q)}{b}\left(\frac{1}{2\varepsilon_0}\right)^q \cdot \left[\frac{(\hat{\tau}_1 - \tau^*)^2}{2\sum_{t=1}^{T}\eta_t} + \frac{\sum_{t=1}^{T}\eta_t^2}{\sum_{t=1}^{T}\eta_t} \cdot \left(\frac{1+\epsilon}{1-\epsilon}\right)^2\right]$$

*where $\bar{\tau} = \sum_{t=1}^{T}\eta_t\hat{\tau}_t/\sum_{t=1}^{T}\eta_t$.*

### B.3. Why does standard online conformal prediction rarely exhibit under-coverage under label noise?

As established in Proposition 3.1 and 4.1, the label noise introduces a gap of $\frac{\epsilon}{1-\epsilon} \cdot \frac{1}{T}\sum_{t=1}^{T}\left((1-\alpha) - \frac{1}{K}\mathbb{E}\left[\mathcal{C}_t(X_t)\right]\right)$ between the actual mis-coverage rate and desired mis-coverage rate $\alpha$. This result indicates that when the prediction sets are sufficiently large such that $(1-\alpha) \leq \frac{1}{K}\mathbb{E}\left[\mathcal{C}_t(X_t)\right]$, a high noise rate $\epsilon$ decreases this upper bound, resulting in under-coverage prediction sets, i.e., $\frac{1}{T}\sum_{t=1}^{T}\mathbb{1}\{Y_t \notin \mathcal{C}_t(X_t)\} \geq \alpha$. However, this scenario only arises when $\mathbb{E}\left[\mathcal{C}_t(X_t)\right] \geq K(1-\alpha)$. To illustrate, considering CIFAR-100 with $K = 100$ classes and error rate $\alpha = 0.1$, under-coverage would require $\mathbb{E}\left[\mathcal{C}_t(X_t)\right] \geq 90$, a condition that remains improbable even with random predictions. Moreover, in such extreme cases, as $(1-\alpha) - \frac{1}{K}\mathbb{E}\left[\mathcal{C}_t(X_t)\right]$ approaches 0, the coverage gap becomes negligible, resulting in coverage rates that approximate the desired $1 - \alpha$. We verify this empirically in Figure 3, where we implement online conformal prediction using an untrained, randomly initialized ResNet18 model on CIFAR-10 and CIFAR-100 with noise rates $\epsilon = 0.05, 0.1, 0.15$. The results confirm that even in this extreme scenario, the coverage gap remains negligible, with coverage rates approaching the desired 0.9.

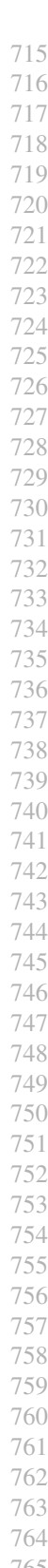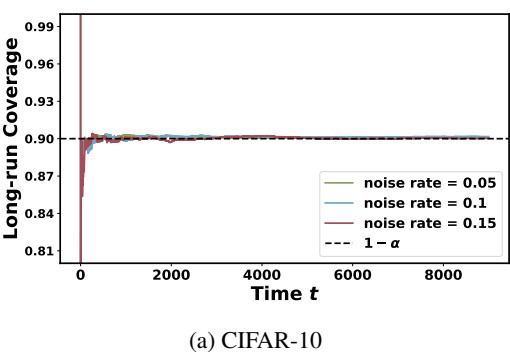 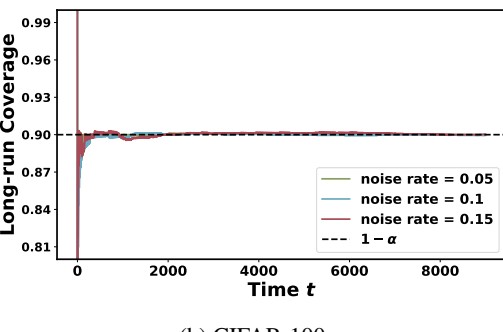

(a) CIFAR-10                    (b) CIFAR-100

Figure 3: Performance of standard online conformal prediction under different noise rates, with a ResNet18 model on CIFAR-10 and CIFAR-100 datasets. We use noisy labels to update the threshold with decaying learning rates $\eta_t = 1/t^{1/2+\varepsilon}$ where $\varepsilon = 0.1$.

### B.4. Common non-conformity scores

**Adaptive Prediction Set (APS). (Romano et al., 2020b)** In the APS method, the non-conformity score of a data pair $(\boldsymbol{x}, y)$ is calculated by accumulating the sorted softmax probability, defined as:

$$\mathcal{S}_{APS}(\boldsymbol{x}, y) = \pi_{(1)}(\boldsymbol{x}) + \cdots + u \cdot \pi_{o(y,\pi(\boldsymbol{x}))}(\boldsymbol{x}),$$

where $\pi_{(1)}(\boldsymbol{x}), \pi_{(2)}(\boldsymbol{x}), \cdots, \pi_{(K)}(\boldsymbol{x})$ are the sorted softmax probabilities in descending order, and $o(y, \pi(\boldsymbol{x}))$ denotes the order of $\pi_y(\boldsymbol{x})$, i.e., the softmax probability for the ground-truth label $y$. In addition, the term $u$ is an independent random variable that follows a uniform distribution on $[0, 1]$.

**Regularized Adaptive Prediction Set (RAPS). (Angelopoulos et al., 2021)** The non-conformity score function of RAPS encourages a small set size by adding a penalty, as formally defined below:

$$\mathcal{S}_{RAPS}(\boldsymbol{x}, y) = \pi_{(1)}(\boldsymbol{x}) + \cdots + u \cdot \pi_{o(y,\pi(\boldsymbol{x}))}(\boldsymbol{x}) + \lambda \cdot \left(o(y, \pi(\boldsymbol{x})) - k_{reg}\right)^+,$$

where $(z)^+ = \max\{0, z\}$, $k_{reg}$ controls the number of penalized classes, and $\lambda$ is the penalty term.

**Sorted Adaptive Prediction Set (SAPS). (Huang et al., 2023)** Recall that APS calculates the non-conformity score by accumulating the sorted softmax values in descending order. However, the softmax probabilities typically exhibit a long-tailed distribution, allowing for easy inclusion of those tail classes in the prediction sets. To alleviate this issue, SAPS discards all the probability values except for the maximum softmax probability when computing the non-conformity score. Formally, the non-conformity score of SAPS for a data pair $(\boldsymbol{x}, y)$ can be calculated as

$$S_{saps}(\boldsymbol{x}, y, u; \hat{\pi}) := \begin{cases} u \cdot \hat{\pi}_{max}(\boldsymbol{x}), & \text{if } o(y, \hat{\pi}(\boldsymbol{x})) = 1, \\ \hat{\pi}_{max}(\boldsymbol{x}) + (o(y, \hat{\pi}(\boldsymbol{x})) - 2 + u) \cdot \lambda, & \text{else}, \end{cases}$$

where $\lambda$ is a hyperparameter representing the weight of ranking information, $\hat{\pi}_{max}(\boldsymbol{x})$ denotes the maximum softmax probability and $u$ is a uniform random variable.

### B.5. Additional experiments for Section 4

To empirically validate Proposition 4.1, we compare the performance of standard online conformal prediction under different noise rates, with a ResNet18 model on CIFAR-100 and ImageNet datasets. We employ LAC score (Sadinle et al., 2019) to generate prediction sets, and use noisy labels to update the threshold with decaying learning rates $\eta_t = 1/t^{1/2+\varepsilon}$ where $\varepsilon = 0.1$. Figure 4 shows that label noise introduces a deviation between the achieved and target coverage rate $1 - \alpha$, and larger noise rates would lead to more substantial coverage distortion.

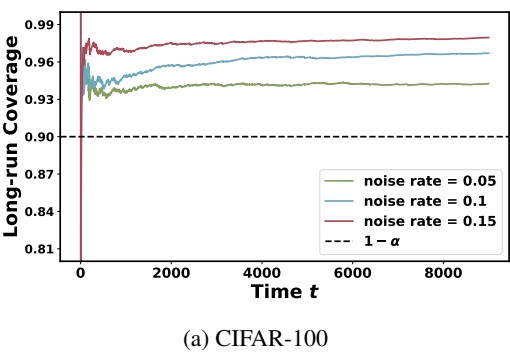

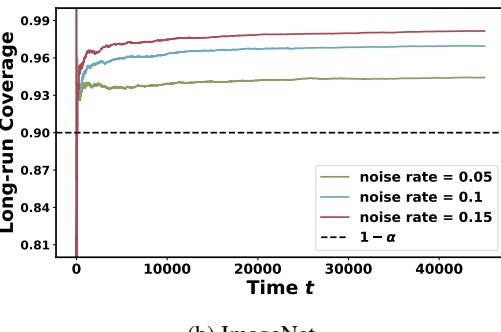

(a) CIFAR-100

(b) ImageNet

Figure 4: Performance comparison of standard online conformal prediction under different noise rates, with a ResNet18 model on CIFAR-100 and ImageNet datasets.

### B.6. Additional experiments on different non-conformity score functions

We evaluate NR-OCP (Ours) against the standard online conformal prediction (Baseline) that updates the threshold with noisy labels for both constant and dynamic learning rate schedules (see Eq. (2) and Eq. (5)). We use LAC (Table 3), APS (Table 4), RAPS (Table 5) and SAPS (Table 6) scores to generate prediction sets with error rates $\alpha \in \{0.1, 0.05\}$, and employ noise rates $\epsilon \in \{0.05, 0.1, 0.15\}$. A detailed description of these non-conformity scores is provided in Appendix B.4.

Table 3: Performance of different methods under uniform noisy labels with LAC score, using ResNet18. 'Baseline' denotes the standard online conformal prediction methods. We include two learning rate schedules: constant learning rate $\eta = 0.05$ and dynamic learning rates $\eta_t = 1/t^{1/2+\varepsilon}$ where $\varepsilon = 0.1$. "↓" indicates smaller values are better and **Bold** numbers are superior results.

| LR Schedule | Error rate | Method | CIFAR100 | | | | ImageNet | | | |
| | | | CovGap(%) ↓ | | Size ↓ | | CovGap(%) ↓ | | Size ↓ | |
| | | | $\alpha = 0.1$ | $\alpha = 0.05$ | $\alpha = 0.1$ | $\alpha = 0.05$ | $\alpha = 0.1$ | $\alpha = 0.05$ | $\alpha = 0.1$ | $\alpha = 0.05$ |
| Constant | $\epsilon = 0.05$ | Baseline | 2.744 | 1.900 | 31.61 | 56.84 | 2.747 | 1.601 | 364.5 | 599.1 |
| | | Ours | **0.289** | **0.122** | **22.78** | **47.31** | **0.018** | **0.138** | **254.7** | **530.6** |
| | $\epsilon = 0.1$ | Baseline | 4.911 | 2.967 | 43.03 | 67.58 | 4.578 | 2.656 | 488.1 | 707.6 |
| | | Ours | **0.056** | **0.189** | **26.57** | **54.18** | **0.027** | **0.156** | **312.4** | **584.9** |
| | $\epsilon = 0.15$ | Baseline | 6.056 | 3.533 | 54.11 | 75.06 | 5.791 | 3.200 | 571.5 | 759.3 |
| | | Ours | **0.378** | **0.344** | **28.39** | **57.53** | **0.251** | **0.076** | **346.7** | **603.4** |
| Dynamic | $\epsilon = 0.05$ | Baseline | 3.978 | 2.833 | 11.54 | 41.75 | 4.333 | 2.889 | 87.27 | 391.8 |
| | | Ours | **0.089** | **0.067** | **4.290** | **26.56** | **0.031** | **0.020** | **16.46** | **150.5** |
| | $\epsilon = 0.1$ | Baseline | 6.756 | 3.844 | 30.07 | 60.17 | 6.960 | 3.933 | 284.6 | 600.2 |
| | | Ours | **0.233** | **0.222** | **5.394** | **30.38** | **0.091** | **0.313** | **27.76** | **227.9** |
| | $\epsilon = 0.15$ | Baseline | 7.844 | 4.289 | 42.69 | 70.67 | 8.098 | 4.276 | 447.9 | 704.4 |
| | | Ours | **0.456** | **0.067** | **8.359** | **33.60** | **0.131** | **0.158** | **38.89** | **272.5** |

Table 4: Performance of different methods under uniform noisy labels with APS score, using ResNet18. 'Baseline' denotes the standard online conformal prediction methods. We include two learning rate schedules: constant learning rate $\eta = 0.05$ and dynamic learning rates $\eta_t = 1/t^{1/2+\varepsilon}$ where $\varepsilon = 0.1$. "↓" indicates smaller values are better and **Bold** numbers are superior results.

| LR Schedule | Error rate | Method | CIFAR100 | | | | ImageNet | | | |
| --- | --- | --- | --- | --- | --- | --- | --- | --- | --- | --- |
| | | | CovGap(%) ↓ | | Size ↓ | | CovGap(%) ↓ | | Size ↓ | |
| | | | $\alpha = 0.1$ | $\alpha = 0.05$ | $\alpha = 0.1$ | $\alpha = 0.05$ | $\alpha = 0.1$ | $\alpha = 0.05$ | $\alpha = 0.1$ | $\alpha = 0.05$ |
| Constant | $\epsilon = 0.05$ | Baseline | 4.556 | 2.933 | 5.862 | 21.17 | 4.728 | 3.508 | 13.35 | 79.88 |
| | | Ours | **2.667** | **0.311** | **2.788** | **6.763** | **0.144** | **0.211** | **4.879** | **14.09** |
| | $\epsilon = 0.1$ | Baseline | 7.511 | 3.822 | 15.71 | 37.19 | 8.402 | 4.388 | 67.25 | 224.2 |
| | | Ours | **1.333** | **0.367** | **2.861** | **6.302** | **0.119** | **0.246** | **4.797** | **14.21** |
| | $\epsilon = 0.15$ | Baseline | 8.533 | 4.188 | 29.76 | 51.63 | 9.380 | 4.644 | 192.4 | 355.3 |
| | | Ours | **0.500** | **0.233** | **2.915** | **6.813** | **0.288** | **0.139** | **4.787** | **14.71** |
| Dynamic | $\epsilon = 0.05$ | Baseline | 4.000 | 2.433 | 4.693 | 13.03 | 4.157 | 2.122 | 11.31 | 31.38 |
| | | Ours | **0.278** | **0.466** | **2.427** | **5.475** | **0.108** | **0.255** | **4.431** | **13.36** |
| | $\epsilon = 0.1$ | Baseline | 7.211 | 3.366 | 11.14 | 21.91 | 6.791 | 3.137 | 27.78 | 54.14 |
| | | Ours | **0.400** | **0.456** | **2.716** | **5.515** | **0.028** | **0.277** | **4.513** | **13.30** |
| | $\epsilon = 0.15$ | Baseline | 8.288 | 3.911 | 19.62 | 31.19 | 7.928 | 3.675 | 47.88 | 80.33 |
| | | Ours | **0.188** | **0.211** | **2.479** | **5.790** | **0.264** | **0.088** | **4.321** | **14.03** |

Table 5: Performance of different methods under uniform noisy labels with RAPS score, using ResNet18. 'Baseline' denotes the standard online conformal prediction methods. We include two learning rate schedules: constant learning rate $\eta = 0.05$ and dynamic learning rates $\eta_t = 1/t^{1/2+\varepsilon}$ where $\varepsilon = 0.1$. "↓" indicates smaller values are better and **Bold** numbers are superior results.

| LR Schedule | Error rate | Method | CIFAR100 | | | | ImageNet | | | |
| --- | --- | --- | --- | --- | --- | --- | --- | --- | --- | --- |
| | | | CovGap(%) ↓ | | Size ↓ | | CovGap(%) ↓ | | Size ↓ | |
| | | | $\alpha = 0.1$ | $\alpha = 0.05$ | $\alpha = 0.1$ | $\alpha = 0.05$ | $\alpha = 0.1$ | $\alpha = 0.05$ | $\alpha = 0.1$ | $\alpha = 0.05$ |
| Constant | $\epsilon = 0.05$ | Baseline | 3.978 | 3.533 | 5.275 | 26.71 | 4.716 | 4.509 | 13.73 | 164.7 |
| | | Ours | **0.533** | **0.089** | **2.971** | **6.265** | **0.051** | **0.673** | **5.217** | **14.21** |
| | $\epsilon = 0.1$ | Baseline | 7.656 | 4.278 | 15.25 | 50.25 | 8.689 | 4.84 | 85.19 | 395.5 |
| | | Ours | **0.356** | **0.200** | **3.126** | **6.370** | **0.184** | **0.271** | **5.264** | **14.76** |
| | $\epsilon = 0.15$ | Baseline | 8.896 | 4.467 | 36.03 | 62.27 | 9.733 | 4.933 | 305.6 | 576.3 |
| | | Ours | **0.489** | **0.560** | **3.500** | **7.693** | **0.224** | **0.227** | **5.616** | **19.81** |
| Dynamic | $\epsilon = 0.05$ | Baseline | 4.322 | 3.254 | 4.883 | 19.42 | 4.642 | 3.457 | 12.57 | 60.94 |
| | | Ours | **0.233** | **0.211** | **2.586** | **5.344** | **0.067** | **0.078** | **4.305** | **13.73** |
| | $\epsilon = 0.1$ | Baseline | 8.378 | 4.400 | 19.68 | 44.43 | 8.084 | 4.311 | 54.42 | 153.7 |
| | | Ours | **0.756** | **0.122** | **2.891** | **5.942** | **0.009** | **0.140** | **4.381** | **14.67** |
| | $\epsilon = 0.15$ | Baseline | 9.022 | 4.600 | 33.87 | 57.87 | 9.256 | 4.978 | 133.3 | 239.8 |
| | | Ours | **0.011** | **0.440** | **2.821** | **5.290** | **0.231** | **0.264** | **4.578** | **15.59** |

Table 6: Performance of different methods under uniform noisy labels with SAPS score, using ResNet18. 'Baseline' denotes the standard online conformal prediction methods. We include two learning rate schedules: constant learning rate $\eta = 0.05$ and dynamic learning rates $\eta_t = 1/t^{1/2+\varepsilon}$ where $\varepsilon = 0.1$. "↓" indicates smaller values are better and **Bold** numbers are superior results.

| LR Schedule | Error rate | Method | CIFAR100 | | | | ImageNet | | | |
| | | | CovGap(%) ↓ | | Size ↓ | | CovGap(%) ↓ | | Size ↓ | |
| | | | $\alpha = 0.1$ | $\alpha = 0.05$ | $\alpha = 0.1$ | $\alpha = 0.05$ | $\alpha = 0.1$ | $\alpha = 0.05$ | $\alpha = 0.1$ | $\alpha = 0.05$ |
|---|---|---|---|---|---|---|---|---|---|---|
| Constant | $\epsilon = 0.05$ | Baseline | 4.489 | 3.100 | 4.982 | 17.92 | 4.462 | 3.537 | 12.53 | 83.04 |
| | | Ours | **0.455** | **0.211** | **2.604** | **5.810** | **0.124** | **0.067** | **4.702** | **14.32** |
| | $\epsilon = 0.1$ | Baseline | 8.478 | 3.944 | 13.25 | 32.71 | 8.368 | 4.389 | 73.14 | 231.5 |
| | | Ours | **0.533** | **0.955** | **2.701** | **4.987** | **0.051** | **0.102** | **5.013** | **15.37** |
| | $\epsilon = 0.15$ | Baseline | 8.644 | 4.400 | 27.89 | 50.73 | 9.360 | 4.613 | 202.3 | 361.7 |
| | | Ours | **0.711** | **0.444** | **2.778** | **5.609** | **0.029** | **0.135** | **4.971** | **15.68** |
| Dynamic | $\epsilon = 0.05$ | Baseline | 4.122 | 2.600 | 4.992 | 13.68 | 4.859 | 1.866 | 13.46 | 28.28 |
| | | Ours | **0.078** | **1.889** | **2.528** | **6.417** | **0.275** | **0.615** | **4.813** | **12.06** |
| | $\epsilon = 0.1$ | Baseline | 7.311 | 3.322 | 11.84 | 22.19 | 7.162 | 3.000 | 31.37 | 51.64 |
| | | Ours | **0.267** | **0.067** | **2.444** | **6.218** | **0.186** | **0.553** | **4.715** | **12.23** |
| | $\epsilon = 0.15$ | Baseline | 8.456 | 4.044 | 21.44 | 33.21 | 8.204 | 3.577 | 55.78 | 75.26 |
| | | Ours | **0.200** | **0.063** | **2.602** | **7.230** | **0.104** | **0.515** | **4.594** | **12.43** |

## B.7. Additional experiments on different model architectures

We evaluate NR-OCP (Ours) against the standard online conformal prediction (Baseline) that updates the threshold with noisy labels for both constant and dynamic learning rate schedules (see Eq. (2) and Eq. (5)), employing ResNet50 (Table 7), DenseNet121 (Table 8), and VGG16 (Table 9). We use LAC scores to generate prediction sets with error rates $\alpha \in \{0.1, 0.05\}$, and employ noise rates $\epsilon \in \{0.05, 0.1, 0.15\}$.

Table 7: Performance of different methods under uniform noisy labels with LAC score, using ResNet50. 'Baseline' denotes the standard online conformal prediction methods. We include two learning rate schedules: constant learning rate $\eta = 0.05$ and dynamic learning rates $\eta_t = 1/t^{1/2+\varepsilon}$ where $\varepsilon = 0.1$. "↓" indicates smaller values are better and **Bold** numbers are superior results.

| LR Schedule | Error rate | Method | CIFAR100 | | | | ImageNet | | | |
| | | | CovGap(%) ↓ | | Size ↓ | | CovGap(%) ↓ | | Size ↓ | |
| | | | $\alpha = 0.1$ | $\alpha = 0.05$ | $\alpha = 0.1$ | $\alpha = 0.05$ | $\alpha = 0.1$ | $\alpha = 0.05$ | $\alpha = 0.1$ | $\alpha = 0.05$ |
|---|---|---|---|---|---|---|---|---|---|---|
| Constant | $\epsilon = 0.05$ | Baseline | 3.244 | 1.783 | 28.58 | 56.08 | 3.226 | 1.982 | 294.4 | 575.7 |
| | | Ours | **0.033** | **0.089** | **16.26** | **46.04** | **0.073** | **0.035** | **188.5** | **466.8** |
| | $\epsilon = 0.1$ | Baseline | 5.267 | 2.879 | 42.09 | 67.25 | 5.326 | 2.935 | 425.9 | 674.2 |
| | | Ours | **0.125** | **0.122** | **17.88** | **49.06** | **0.231** | **0.067** | **241.1** | **506.4** |
| | $\epsilon = 0.15$ | Baseline | 6.369 | 3.445 | 53.34 | 74.03 | 6.584 | 3.453 | 531.4 | 758.5 |
| | | Ours | **0.223** | **0.216** | **21.62** | **50.14** | **0.202** | **0.056** | **263.1** | **549.4** |
| Dynamic | $\epsilon = 0.05$ | Baseline | 4.166 | 2.724 | 10.91 | 41.95 | 4.397 | 3.104 | 44.86 | 361.8 |
| | | Ours | **0.115** | **0.189** | **3.120** | **18.35** | **0.082** | **0.048** | **8.377** | **96.68** |
| | $\epsilon = 0.1$ | Baseline | 6.984 | 3.921 | 30.09 | 61.96 | 7.279 | 4.073 | 232.8 | 582.4 |
| | | Ours | **0.205** | **0.310** | **4.143** | **27.87** | **0.113** | **0.178** | **10.83** | **170.2** |
| | $\epsilon = 0.15$ | Baseline | 8.093 | 4.223 | 43.90 | 70.67 | 8.406 | 4.453 | 417.9 | 705.9 |
| | | Ours | **0.244** | **0.043** | **4.368** | **27.81** | **0.133** | **0.059** | **21.98** | **178.0** |

Table 8: Performance of different methods under uniform noisy labels with LAC score, using DenseNet121. 'Baseline' denotes the standard online conformal prediction methods. We include two learning rate schedules: constant learning rate $\eta = 0.05$ and dynamic learning rates $\eta_t = 1/t^{1/2+\varepsilon}$ where $\varepsilon = 0.1$. "↓" indicates smaller values are better and **Bold** numbers are superior results.

| LR Schedule | Error rate | Method | CIFAR100 | | | | ImageNet | | | |
| | | | CovGap(%) ↓ | | Size ↓ | | CovGap(%) ↓ | | Size ↓ | |
| | | | $\alpha = 0.1$ | $\alpha = 0.05$ | $\alpha = 0.1$ | $\alpha = 0.05$ | $\alpha = 0.1$ | $\alpha = 0.05$ | $\alpha = 0.1$ | $\alpha = 0.05$ |
|---|---|---|---|---|---|---|---|---|---|---|
| Constant | $\epsilon = 0.05$ | Baseline | 3.978 | 3.533 | 5.275 | 26.71 | 4.716 | 4.509 | 13.73 | 164.7 |
| | | Ours | **0.533** | **0.089** | **2.971** | **6.265** | **0.051** | **0.673** | **5.217** | **14.21** |
| | $\epsilon = 0.1$ | Baseline | 7.656 | 4.278 | 15.25 | 50.25 | 8.689 | 4.84 | 85.19 | 395.5 |
| | | Ours | **0.356** | **0.200** | **3.126** | **6.370** | **0.184** | **0.271** | **5.264** | **14.76** |
| | $\epsilon = 0.15$ | Baseline | 8.896 | 4.467 | 36.03 | 62.27 | 9.733 | 4.933 | 305.6 | 576.3 |
| | | Ours | **0.489** | **0.560** | **3.500** | **7.693** | **0.224** | **0.227** | **5.616** | **19.81** |
| Dynamic | $\epsilon = 0.05$ | Baseline | 4.322 | 3.254 | 4.883 | 19.42 | 4.642 | 3.457 | 12.57 | 60.94 |
| | | Ours | **0.233** | **0.211** | **2.586** | **5.344** | **0.067** | **0.078** | **4.305** | **13.73** |
| | $\epsilon = 0.1$ | Baseline | 8.378 | 4.400 | 19.68 | 44.43 | 8.084 | 4.311 | 54.42 | 153.7 |
| | | Ours | **0.756** | **0.122** | **2.891** | **5.942** | **0.009** | **0.140** | **4.381** | **14.67** |
| | $\epsilon = 0.15$ | Baseline | 9.022 | 4.600 | 33.87 | 57.87 | 9.256 | 4.978 | 133.3 | 239.8 |
| | | Ours | **0.011** | **0.440** | **2.821** | **5.290** | **0.231** | **0.264** | **4.578** | **15.59** |

Table 9: Performance of different methods under uniform noisy labels with LAC score, using VGG16. 'Baseline' denotes the standard online conformal prediction methods. We include two learning rate schedules: constant learning rate $\eta = 0.05$ and dynamic learning rates $\eta_t = 1/t^{1/2+\varepsilon}$ where $\varepsilon = 0.1$. "↓" indicates smaller values are better and **Bold** numbers are superior results.

| LR Schedule | Error rate | Method | CIFAR100 | | | | ImageNet | | | |
| | | | CovGap(%) ↓ | | Size ↓ | | CovGap(%) ↓ | | Size ↓ | |
| | | | $\alpha = 0.1$ | $\alpha = 0.05$ | $\alpha = 0.1$ | $\alpha = 0.05$ | $\alpha = 0.1$ | $\alpha = 0.05$ | $\alpha = 0.1$ | $\alpha = 0.05$ |
|---|---|---|---|---|---|---|---|---|---|---|
| Constant | $\epsilon = 0.05$ | Baseline | 1.911 | 1.033 | 51.13 | 73.03 | 3.164 | 1.995 | 297.8 | 565.3 |
| | | Ours | **0.078** | **0.144** | **43.50** | **67.97** | **0.002** | **0.031** | **206.6** | **488.1** |
| | $\epsilon = 0.1$ | Baseline | 3.474 | 1.700 | 58.08 | 76.44 | 4.878 | 2.975 | 439.8 | 669.4 |
| | | Ours | **0.224** | **0.267** | **45.48** | **68.28** | **0.217** | **0.119** | **229.4** | **511.2** |
| | $\epsilon = 0.15$ | Baseline | 4.489 | 2.411 | 64.16 | 80.16 | 6.173 | 3.626 | 544.2 | 746.8 |
| | | Ours | **0.300** | **0.189** | **44.43** | **71.81** | **0.168** | **0.004** | **281.9** | **551.3** |
| Dynamic | $\epsilon = 0.05$ | Baseline | 2.678 | 1.388 | 40.96 | 66.81 | 4.608 | 3.044 | 51.64 | 374.1 |
| | | Ours | **0.244** | **0.004** | **28.92** | **59.26** | **0.062** | **0.013** | **5.316** | **112.7** |
| | $\epsilon = 0.1$ | Baseline | 4.189 | 2.567 | 50.40 | 74.38 | 7.142 | 4.067 | 244.4 | 580.6 |
| | | Ours | **0.289** | **0.133** | **30.81** | **61.08** | **0.195** | **0.071** | **10.65** | **152.1** |
| | $\epsilon = 0.15$ | Baseline | 5.733 | 3.044 | 60.46 | 78.65 | 8.413 | 4.451 | 432.2 | 708.9 |
| | | Ours | **0.440** | **0.144** | **35.67** | **63.38** | **0.119** | **0.282** | **18.39** | **224.6** |

## C. Proof for Section 3

### C.1. Proof for Proposition 3.1

**Lemma C.1.** *Given Assumptions 2.1 and 2.2, when updating the threshold according to Eq. (2), for any $T \in \mathbb{N}^+$, we have*

$$-\alpha\eta \leq \hat{\tau}_t \leq 1 + (1 - \alpha)\eta. \tag{7}$$

*Proof.* We prove this by induction. First, we know $\hat{\tau}_1 \in [0, 1]$ by assumption, which indicates that Eq. (7) is satisfied at $t = 1$. Then, we assume that Eq. (7) holds for $t = T$, and we will show that $\hat{\tau}_{T+1}$ lies in this range. Consider three cases:

**Case 1.** If $\hat{\tau}_T \in [0, 1]$, we have

$$\hat{\tau}_{T+1} = \hat{\tau}_T - \eta \cdot \nabla_{\hat{\tau}_t} l_{1-\alpha}(\hat{\tau}_t, \tilde{S}_t) \overset{(a)}{\in} [-\alpha\eta, 1 + (1 - \alpha)\eta]$$

where (a) is because $\nabla_{\hat{\tau}_t} l_{1-\alpha}(\hat{\tau}_t, \tilde{S}_t) \in [\alpha - 1, \alpha]$.

**Case 2.** Consider the case where $\hat{\tau}_T \in [1, 1 + (1 - \alpha)\eta]$. The assumption that $\tilde{S} \in [0, 1]$ implies $\mathbb{1}\left\{\tilde{S} > \hat{\tau}_t\right\} = 0$. Thus, we have

$$\nabla_{\hat{\tau}_t} l_{1-\alpha}(\hat{\tau}_t, \tilde{S}_t) = -\mathbb{1}\left\{\tilde{S} > \hat{\tau}_t\right\} + \alpha = \alpha$$

which follows that

$$\hat{\tau}_{T+1} = \hat{\tau}_T - \eta \cdot \nabla_{\hat{\tau}_t} l_{1-\alpha}(\hat{\tau}_t, \tilde{S}_t) = \hat{\tau}_T - \eta\alpha \in [1 - \eta\alpha, 1 + (1 - \alpha)\eta] \subset [-\alpha\eta, 1 + (1 - \alpha)\eta]$$

**Case 3.** Consider the case where $\hat{\tau}_T \in [-\alpha\eta, 0]$. The assumption that $\tilde{S} \in [0, 1]$ implies $\mathbb{1}\left\{\tilde{S} > \hat{\tau}_t\right\} = 1$. Thus, we have

$$\nabla_{\hat{\tau}_t} l_{1-\alpha}(\hat{\tau}_t, \tilde{S}_t) = -\mathbb{1}\left\{\tilde{S} > \hat{\tau}_t\right\} + \alpha = -1 + \alpha$$

which follows that

$$\hat{\tau}_{T+1} = \hat{\tau}_T - \eta \cdot \nabla_{\hat{\tau}_t} l_{1-\alpha}(\hat{\tau}_t, \tilde{S}_t) = \hat{\tau}_T - \eta(-1 + \alpha) \in [-\alpha\eta, (1 - \alpha)\eta] \subset [-\alpha\eta, 1 + (1 - \alpha)\eta]$$

Combining three cases, we can conclude that

$$-\alpha\eta \leq \hat{\tau}_t \leq 1 + (1 - \alpha)\eta$$

$\square$

**Proposition C.2** (Restatement of Proposition 3.1). *Consider online conformal prediction under uniform label noise with noise rate $\epsilon \in (0, 1)$. Given Assumptions 2.1 and 2.2, when updating the threshold according to Eq. (2), for any $\delta \in (0, 1)$ and $T \in \mathbb{N}^+$, the following bound holds with probability at least $1 - \delta$:*

$$\alpha - \frac{1}{T}\sum_{t=1}^{T}\mathbb{1}\left\{Y_t \notin \mathcal{C}_t(X_t)\right\} \leq \frac{2 - \epsilon}{1 - \epsilon}\sqrt{2\log\left(\frac{4}{\delta}\right)} \cdot \frac{1}{\sqrt{T}} + \frac{1}{1 - \epsilon}\left(\frac{1}{\eta} + 1 - \alpha\right) \cdot \frac{1}{T} + \frac{\epsilon}{1 - \epsilon} \cdot \frac{1}{T}\sum_{t=1}^{T}\left((1 - \alpha) - \frac{1}{K}\mathbb{E}\left[\mathcal{C}_t(X_t)\right]\right).$$

*Proof.* Consider the gradient of clean pinball loss:

$$\sum_{t=1}^{T}\nabla_{\tau_t} l_{1-\alpha}(\tau_t, S_t) = \sum_{t=1}^{T}\nabla_{\tau_t} l_{1-\alpha}(\tau_t, S_t) - \sum_{t=1}^{T}\mathbb{E}_S\left[\nabla_{\tau_t} l_{1-\alpha}(\tau_t, S_t)\right] + \sum_{t=1}^{T}\mathbb{E}_S\left[\nabla_{\tau_t} l_{1-\alpha}(\tau_t, S_t)\right]$$

$$\leq \underbrace{\left|\sum_{t=1}^{T}\nabla_{\tau_t} l_{1-\alpha}(\tau_t, S_t) - \sum_{t=1}^{T}\mathbb{E}_S\left[\nabla_{\tau_t} l_{1-\alpha}(\tau_t, S_t)\right]\right|}_{(a)} + \underbrace{\sum_{t=1}^{T}\mathbb{E}_S\left[\nabla_{\tau_t} l_{1-\alpha}(\tau_t, S_t)\right]}_{(b)}$$

Part (a): Lemma G.3 gives us that

$$\mathbb{P}\left\{\left|\sum_{t=1}^{T}\mathbb{E}_{S_t}\left[\nabla_{\hat{\tau}_t}l_{1-\alpha}(\hat{\tau}_t, S_t)\right] - \sum_{t=1}^{T}\nabla_{\hat{\tau}_t}l_{1-\alpha}(\hat{\tau}_t, S_t)\right| \le \sqrt{2T\log\left(\frac{4}{\delta}\right)}\right\} \ge 1 - \frac{\delta}{2}.$$

Part (b): Recall that the expected gradient of the clean pinball loss satisfies

$$\mathbb{E}_S\left[\nabla_{\hat{\tau}_t}l_{1-\alpha}(\hat{\tau}_t, S_t)\right] = \mathbb{P}\left\{S_t \le \hat{\tau}_t\right\} - (1-\alpha)$$

$$\stackrel{(a)}{=} \left(\frac{1}{1-\epsilon}\mathbb{P}\left\{\tilde{S}_t \le \hat{\tau}_t\right\} - \frac{\epsilon}{K(1-\epsilon)}\sum_{y=1}^{K}\mathbb{P}\left\{S_{t,y} \le \hat{\tau}_t\right\}\right) - (1-\alpha)$$

$$= \frac{1}{1-\epsilon}\left(\mathbb{P}\left\{\tilde{S}_t \le \hat{\tau}_t\right\} - (1-\alpha)\right) - \frac{\epsilon}{1-\epsilon}\left(\frac{1}{K}\sum_{y=1}^{K}\mathbb{P}\left\{S_{t,y} \le \hat{\tau}_t\right\} - (1-\alpha)\right)$$

$$= \frac{1}{1-\epsilon}\mathbb{E}_{\tilde{S}_t}\left[\nabla_{\hat{\tau}_t}l_{1-\alpha}(\hat{\tau}_t, \tilde{S}_t)\right] - \frac{\epsilon}{1-\epsilon}\left(\frac{1}{K}\mathbb{E}\left[\mathcal{C}_t(X_t)\right] - (1-\alpha)\right)$$

where (a) comes from Lemma G.1. In addition, Lemma G.3 implies that

$$\mathbb{P}\left\{\left|\sum_{t=1}^{T}\mathbb{E}_{\tilde{S}_t}\left[\nabla_{\hat{\tau}_t}l_{1-\alpha}(\hat{\tau}_t, \tilde{S}_t)\right] - \nabla_{\hat{\tau}_t}l_{1-\alpha}(\hat{\tau}_t, \tilde{S}_t)\right| \le \sqrt{2T\log\left(\frac{4}{\delta}\right)}\right\} \ge 1 - \frac{\delta}{2}$$

Thus, we can derive an upper bound for (b) as follows

$$\sum_{t=1}^{T}\mathbb{E}_S\left[\nabla_{\hat{\tau}_t}l_{1-\alpha}(\hat{\tau}_t, S)\right]$$

$$= \frac{1}{1-\epsilon}\sum_{t=1}^{T}\mathbb{E}_{\tilde{S}}\left[\nabla_{\hat{\tau}_t}l_{1-\alpha}(\hat{\tau}_t, \tilde{S})\right] + \frac{\epsilon}{1-\epsilon}\sum_{t=1}^{T}\left((1-\alpha) - \frac{1}{K}\mathbb{E}\left[\mathcal{C}_t(X_t)\right]\right)$$

$$= \frac{1}{1-\epsilon}\left(\sum_{t=1}^{T}\mathbb{E}_{\tilde{S}}\left[\nabla_{\hat{\tau}_t}l_{1-\alpha}(\hat{\tau}_t, \tilde{S})\right] - \sum_{t=1}^{T}\nabla_{\hat{\tau}_t}l_{1-\alpha}(\hat{\tau}_t, \tilde{S}) + \sum_{t=1}^{T}\nabla_{\hat{\tau}_t}l_{1-\alpha}(\hat{\tau}_t, \tilde{S})\right) + \frac{\epsilon}{1-\epsilon}\sum_{t=1}^{T}\left((1-\alpha) - \frac{1}{K}\mathbb{E}\left[\mathcal{C}_t(X_t)\right]\right)$$

$$\le \frac{1}{1-\epsilon}\left|\sum_{t=1}^{T}\mathbb{E}_{\tilde{S}}\left[\nabla_{\hat{\tau}_t}l_{1-\alpha}(\hat{\tau}_t, \tilde{S})\right] - \sum_{t=1}^{T}\nabla_{\hat{\tau}_t}l_{1-\alpha}(\hat{\tau}_t, \tilde{S})\right| + \frac{1}{1-\epsilon}\left|\sum_{t=1}^{T}\nabla_{\hat{\tau}_t}l_{1-\alpha}(\hat{\tau}_t, \tilde{S})\right| + \frac{\epsilon}{1-\epsilon}\sum_{t=1}^{T}\left((1-\alpha) - \frac{1}{K}\mathbb{E}\left[\mathcal{C}_t(X_t)\right]\right)$$

$$\le \frac{1}{1-\epsilon}\sqrt{2T\log\left(\frac{4}{\delta}\right)} + \frac{1}{1-\epsilon}\underbrace{\left|\sum_{t=1}^{T}\nabla_{\hat{\tau}_t}l_{1-\alpha}(\hat{\tau}_t, \tilde{S})\right|}_{(c)} + \frac{\epsilon}{1-\epsilon}\sum_{t=1}^{T}\left((1-\alpha) - \frac{1}{K}\mathbb{E}\left[\mathcal{C}_t(X_t)\right]\right)$$

Then, we derive an upper bound for (c). The update rule (Eq. (2)) gives us that

$$\hat{\tau}_{t+1} = \hat{\tau}_t - \eta\nabla_{\hat{\tau}_t}l_{1-\alpha}(\hat{\tau}_t, \tilde{S}_t) \quad \implies \quad \nabla_{\hat{\tau}_t}l_{1-\alpha}(\hat{\tau}_t, \tilde{S}_t) = \frac{1}{\eta}(\hat{\tau}_t - \hat{\tau}_{t+1}).$$

Accumulating from $t=1$ to $t=T$ and taking absolute value gives

$$\left|\sum_{t=1}^{T}\nabla_{\hat{\tau}_t}l_{1-\alpha}(\hat{\tau}_t, S_t)\right| = \left|\sum_{t=1}^{T}\frac{1}{\eta}(\hat{\tau}_t - \hat{\tau}_{t+1})\right| = \frac{1}{\eta}|\hat{\tau}_1 - \hat{\tau}_{T+1}| \stackrel{(a)}{\le} \frac{1}{\eta}(1 + (1-\alpha)\eta) = \frac{1}{\eta} + 1 - \alpha$$

where (a) follows from the assumption that $\hat{\tau}_1 \in [0, 1]$ and Lemma C.1. Thus, we have

$$\sum_{t=1}^{T}\mathbb{E}_S\left[\nabla_{\hat{\tau}_t}l_{1-\alpha}(\hat{\tau}_t, S)\right] \le \frac{1}{1-\epsilon}\sqrt{2T\log\left(\frac{4}{\delta}\right)} + \frac{1}{1-\epsilon}\left(\frac{1}{\eta} + 1 - \alpha\right) + \frac{\epsilon}{1-\epsilon}\sum_{t=1}^{T}\left((1-\alpha) - \frac{1}{K}\mathbb{E}\left[\mathcal{C}_t(X_t)\right]\right)$$

By taking union bound and combining two parts, we can obtain that

$$\sum_{t=1}^{T} \nabla_{\tau_t} l_{1-\alpha}(\tau_t, S_t) \leq \sqrt{2T \log\left(\frac{4}{\delta}\right)} + \frac{1}{1-\epsilon}\sqrt{2T \log\left(\frac{4}{\delta}\right)} + \frac{1}{1-\epsilon}\left(\frac{1}{\eta} + 1 - \alpha\right) + \frac{\epsilon}{1-\epsilon}\sum_{t=1}^{T}\left(\frac{1}{K}\mathbb{E}\left[\mathcal{C}_t(X_t)\right] - (1-\alpha)\right)$$

$$= \frac{2-\epsilon}{1-\epsilon}\sqrt{2T \log\left(\frac{4}{\delta}\right)} + \frac{1}{1-\epsilon}\left(\frac{1}{\eta} + 1 - \alpha\right) + \frac{\epsilon}{1-\epsilon}\sum_{t=1}^{T}\left((1-\alpha) - \frac{1}{K}\mathbb{E}\left[\mathcal{C}_t(X_t)\right]\right)$$

holds with at least $1 - \delta$ probability. Recall that

$$\sum_{t=1}^{T} \nabla_{\tau_t} l_{1-\alpha}(\tau_t, S_t) = \sum_{t=1}^{T}\left(\mathbb{1}\left\{S_t \leq \tau_t\right\} - (1-\alpha)\right) = \sum_{t=1}^{T}\left(-\mathbb{1}\left\{S_t \geq \tau_t\right\} + \alpha\right) = \sum_{t=1}^{T}\left(-\mathbb{1}\left\{Y_t \notin \mathcal{C}_t(X_t)\right\} + \alpha\right)$$

We can conclude that

$$\alpha - \frac{1}{T}\sum_{t=1}^{T}\mathbb{1}\left\{Y_t \notin \mathcal{C}_t(X_t)\right\} \leq \frac{2-\epsilon}{1-\epsilon}\sqrt{2\log\left(\frac{4}{\delta}\right)} \cdot \frac{1}{\sqrt{T}} + \frac{1}{1-\epsilon}\left(\frac{1}{\eta} + 1 - \alpha\right) \cdot \frac{1}{T} + \frac{\epsilon}{1-\epsilon} \cdot \frac{1}{T}\sum_{t=1}^{T}\left((1-\alpha) - \frac{1}{K}\mathbb{E}\left[\mathcal{C}_t(X_t)\right]\right)$$

$\square$

## C.2. Proof for Proposition 3.2

**Proposition C.3** (Restatement of Proposition 3.2). *The robust pinball loss defined in Eq. (3) satisfies the following two properties:*

$$(1)\, \mathbb{E}_S\left[l_{1-\alpha}(\tau, S)\right] = \mathbb{E}_{\tilde{S}, S_y}\left[\tilde{l}_{1-\alpha}(\tau, \tilde{S}, \{S_y\}_{y=1}^{K})\right];$$

$$(2)\, \mathbb{E}_S\left[\nabla_\tau l_{1-\alpha}(\tau, S)\right] = \mathbb{E}_{\tilde{S}, S_y}\left[\nabla_\tau \tilde{l}_{1-\alpha}(\tau, \tilde{S}, \{S_y\}_{y=1}^{K})\right].$$

*Proof.* **The property (1):** We begin by proving the first property. Taking expectation on pinball loss gives

$$\mathbb{E}_S\left[l_{1-\alpha}(\tau, S)\right] = \mathbb{E}_S\left[\alpha(\tau - S)\mathbb{1}\{\tau \geq S\} + (1-\alpha)(S - \tau)\mathbb{1}\{\tau \leq S\}\right]$$

$$= \alpha\tau\mathbb{P}\{\tau \geq S\} - \alpha\int_0^\tau s\mathbb{P}\{S = s\}\, ds - (1-\alpha)\tau\mathbb{P}\{\tau < S\} + (1-\alpha)\int_\tau^1 s\mathbb{P}\{S = s\}\, ds$$

$$= \underbrace{\alpha\tau\left(\frac{1}{1-\epsilon}\mathbb{P}\left\{\tilde{S} \leq \tau\right\} + \frac{\epsilon}{1-\epsilon}\mathbb{P}\left\{\bar{S} \leq \tau\right\}\right)}_{(a)} - \underbrace{\alpha\int_0^\tau s\left(\frac{1}{1-\epsilon}\mathbb{P}\left\{\tilde{S} \leq \tau\right\} + \frac{\epsilon}{1-\epsilon}\mathbb{P}\left\{\bar{S} \leq \tau\right\}\right)ds}_{(b)} -$$

$$\underbrace{(1-\alpha)\tau\left(\frac{1}{1-\epsilon}\mathbb{P}\left\{\tilde{S} > \tau\right\} + \frac{\epsilon}{1-\epsilon}\mathbb{P}\left\{\bar{S} > \tau\right\}\right)}_{(c)} + \underbrace{(1-\alpha)\int_\tau^1 s\left(\frac{1}{1-\epsilon}\mathbb{P}\left\{\tilde{S} > \tau\right\} + \frac{\epsilon}{1-\epsilon}\mathbb{P}\left\{\bar{S} > \tau\right\}\right)ds}_{(d)}$$

Part (a):

$$\alpha\tau\left(\frac{1}{1-\epsilon}\mathbb{P}\left\{\tilde{S} \leq \tau\right\} + \frac{\epsilon}{1-\epsilon}\mathbb{P}\left\{\bar{S} \leq \tau\right\}\right) = \alpha\tau\left(\frac{1}{1-\epsilon}\mathbb{P}\left\{\tilde{S} \leq \tau\right\} + \frac{\epsilon}{K(1-\epsilon)}\sum_{y=1}^{K}\mathbb{P}\left\{S_y \leq \tau\right\}\right)$$

$$= \mathbb{E}_{\tilde{S}, S_y}\left[\alpha\tau\left(\frac{1}{1-\epsilon}\mathbb{1}\left\{\tilde{S} \leq \tau\right\} + \frac{\epsilon}{K(1-\epsilon)}\sum_{y=1}^{K}\mathbb{1}\left\{S_y \leq \tau\right\}\right)\right]$$

Part (b):

$$\alpha\int_0^\tau s\left(\frac{1}{1-\epsilon}\mathbb{P}\left\{\tilde{S} \leq \tau\right\} + \frac{\epsilon}{1-\epsilon}\mathbb{P}\left\{\bar{S} \leq \tau\right\}\right)ds = \alpha\int_0^\tau s\left(\frac{1}{1-\epsilon}\mathbb{P}\left\{\tilde{S} \leq \tau\right\} + \frac{\epsilon}{K(1-\epsilon)}\sum_{y=1}^{K}\mathbb{P}\left\{S_y \leq \tau\right\}\right)ds$$

$$= \mathbb{E}_{\tilde{S}, S_y}\left[\alpha\left(\frac{\tilde{S}}{1-\epsilon}\mathbb{1}\left\{\tilde{S} \leq \tau\right\} + \frac{\epsilon S_y}{K(1-\epsilon)}\sum_{y=1}^{K}\right)\mathbb{1}\left\{S_y \leq \tau\right\}\right]$$

Part (c):

$$(1-\alpha)\tau\left(\frac{1}{1-\epsilon}\mathbb{P}\left\{\tilde{S}>\tau\right\}+\frac{\epsilon}{1-\epsilon}\mathbb{P}\left\{\bar{S}>\tau\right\}\right)=(1-\alpha)\tau\left(\frac{1}{1-\epsilon}\mathbb{P}\left\{\tilde{S}>\tau\right\}+\frac{\epsilon}{K(1-\epsilon)}\sum_{y=1}^{K}\mathbb{P}\left\{S_y>\tau\right\}\right)$$

$$=\mathbb{E}_{\tilde{S},S_y}\left[(1-\alpha)\tau\left(\frac{1}{1-\epsilon}\mathbb{1}\left\{\tilde{S}>\tau\right\}+\frac{\epsilon}{K(1-\epsilon)}\sum_{y=1}^{K}\mathbb{1}\left\{S_y>\tau\right\}\right)\right]$$

Part (d):

$$(1-\alpha)\int_{\tau}^{1}s\left(\frac{1}{1-\epsilon}\mathbb{P}\left\{\tilde{S}>\tau\right\}+\frac{\epsilon}{1-\epsilon}\mathbb{P}\left\{\bar{S}>\tau\right\}\right)ds=(1-\alpha)\int_{0}^{\tau}s\left(\frac{1}{1-\epsilon}\mathbb{P}\left\{\tilde{S}>\tau\right\}+\frac{\epsilon}{K(1-\epsilon)}\sum_{y=1}^{K}\mathbb{P}\left\{S_y>\tau\right\}\right)ds$$

$$=\mathbb{E}_{\tilde{S},S_y}\left[(1-\alpha)\left(\frac{\tilde{S}}{1-\epsilon}\mathbb{1}\left\{\tilde{S}>\tau\right\}+\frac{\epsilon S_y}{K(1-\epsilon)}\sum_{y=1}^{K}\right)\mathbb{1}\left\{S_y>\tau\right\}\right]$$

Combining (a), (b), (c), and (d), we can conclude that

$$\mathbb{E}_S\left[l_{1-\alpha}(\tau,S)\right]$$

$$=\mathbb{E}_{\tilde{S},S_y}\left[\alpha\tau\left(\frac{1}{1-\epsilon}\mathbb{1}\left\{\tilde{S}\leq\tau\right\}+\frac{\epsilon}{K(1-\epsilon)}\sum_{y=1}^{K}\mathbb{1}\left\{S_y\leq\tau\right\}\right)-\alpha\left(\frac{\tilde{S}}{1-\epsilon}\mathbb{1}\left\{\tilde{S}\leq\tau\right\}-\frac{\epsilon S_y}{K(1-\epsilon)}\sum_{y=1}^{K}\right)\mathbb{1}\left\{S_y\leq\tau\right\}\right]-$$

$$\mathbb{E}_{\tilde{S},S_y}\left[(1-\alpha)\tau\left(\frac{1}{1-\epsilon}\mathbb{1}\left\{\tilde{S}>\tau\right\}+\frac{\epsilon}{K(1-\epsilon)}\sum_{y=1}^{K}\mathbb{1}\left\{S_y>\tau\right\}\right)+(1-\alpha)\left(\frac{\tilde{S}}{1-\epsilon}\mathbb{1}\left\{\tilde{S}>\tau\right\}+\frac{\epsilon S_y}{K(1-\epsilon)}\sum_{y=1}^{K}\mathbb{1}\left\{S_y>\tau\right\}\right)\right]$$

$$=\mathbb{E}\left[\frac{1}{1-\epsilon}\left(\alpha(\tau-\tilde{S})\mathbb{1}\left\{\tilde{S}\leq\tau\right\}+(1-\alpha)(\tilde{S}-\tau)\mathbb{1}\left\{\tilde{S}>\tau\right\}\right)\right]+$$

$$\mathbb{E}\left[\frac{\epsilon}{K(1-\epsilon)}\sum_{y=1}^{K}\left(\alpha(\tau-\tilde{S})\mathbb{1}\left\{S_y\leq\tau\right\}+(1-\alpha)(\tilde{S}-\tau)\mathbb{1}\left\{S_y>\tau\right\}\right)\right]$$

$$=\mathbb{E}\left[\frac{1}{1-\epsilon}l_{1-\alpha}(\tau,\tilde{S})+\frac{\epsilon}{K(1-\epsilon)}\sum_{y=1}^{K}l_{1-\alpha}(\tau,S_y)\right]$$

$$=\mathbb{E}_{\tilde{S},S_y}\left[\tilde{l}_{1-\alpha}(\tau,\tilde{S},\{S_y\}_{y=1}^{K})\right]$$

**The property (2):** We proceed by proving the second property, which demonstrates that the expected gradient of the robust pinball loss (computed using noise and random scores) equals the expected gradient of the true pinball loss. Consider the gradient of robust pinball loss:

$$\mathbb{E}_{\tilde{S},S_y}\left[\nabla_\tau\tilde{l}_{1-\alpha}(\tau,\tilde{S},\{S_y\}_{y=1}^{K})\right]$$

$$=\mathbb{E}_{\tilde{S},S_y}\left[\frac{\alpha}{1-\epsilon}\mathbb{1}\left\{\tau\geq\tilde{S}\right\}-\sum_{y=1}^{K}\frac{\alpha\epsilon}{K(1-\epsilon)}\mathbb{1}\left\{\tau\geq S_y\right\}-\frac{1-\alpha}{1-\epsilon}\mathbb{1}\left\{\tau<\tilde{S}\right\}+\sum_{y=1}^{K}\frac{(1-\alpha)\epsilon}{K(1-\epsilon)}\mathbb{1}\left\{\tau<S_y\right\}\right]$$

$$=\frac{\alpha}{1-\epsilon}\mathbb{P}\left\{\tau\geq\tilde{S}\right\}-\sum_{y=1}^{K}\frac{\alpha\epsilon}{K(1-\epsilon)}\mathbb{P}\left\{\tau\geq S_y\right\}-\frac{1-\alpha}{1-\epsilon}\mathbb{P}\left\{\tau<\tilde{S}\right\}+\sum_{y=1}^{K}\frac{(1-\alpha)\epsilon}{K(1-\epsilon)}\mathbb{P}\left\{\tau<S_y\right\}$$

$$=\alpha\mathbb{P}\left\{\tau\geq S\right\}-(1-\alpha)\mathbb{P}\left\{\tau<S\right\}$$

$$=\mathbb{E}_S\left[\nabla_\tau l_{1-\alpha}(\tau,S)\right]$$

$\square$

## C.3. Proof for Proposition 3.3

**Lemma C.4.** *Consider online conformal prediction under uniform label noise with noise rate $\epsilon \in (0, 1)$. Given Assumptions 2.1 and 2.2, when updating the threshold according to Eq. (4), for any $\delta \in (0, 1)$ and $T \in \mathbb{N}^+$, we have*

$$-\alpha\eta - \frac{\epsilon\eta}{1-\epsilon} \leq \hat{\tau}_T \leq 1 - \alpha\eta + \frac{\eta}{1-\epsilon}. \tag{8}$$

*Proof.* We prove this by induction. First, we know $\hat{\tau}_1 \in [0, 1]$ by assumption, which indicates that Eq. (8) is satisfied at $t = 1$. Then, we assume that Eq. (8) holds for $t = T$, and we will show that $\hat{\tau}_{T+1}$ lies in this range. Consider three cases:

**Case 1.** If $\hat{\tau}_T \in [0, 1]$, we have

$$\hat{\tau}_{T+1} = \hat{\tau}_T - \eta \cdot \nabla_{\hat{\tau}_T}\tilde{l}_{1-\alpha}(\hat{\tau}_T, \tilde{S}_T, \{S_{t,y}\}_{y=1}^K) \overset{(a)}{\in} \left[-\alpha\eta - \frac{\epsilon\eta}{1-\epsilon}, 1 - \alpha\eta + \frac{\eta}{1-\epsilon}\right]$$

where (a) follows from Lemma G.2.

**Case 2.** Consider the case where $\hat{\tau}_T \in [1, 1+\alpha\eta-\eta/(1-\epsilon)]$. The assumption that $\tilde{S}_T, S_{t,y} \in [0, 1]$ implies $\mathbb{1}\left\{\tilde{S}_T \leq \hat{\tau}_T\right\} = \mathbb{1}\left\{S_{t,y} \leq \hat{\tau}_T\right\} = 1$. Thus, we have

$$\nabla_{\hat{\tau}_T}\tilde{l}_{1-\alpha}(\hat{\tau}_T, \tilde{S}_T, \{S_{t,y}\}_{y=1}^K) = \frac{\alpha}{1-\epsilon}\mathbb{1}\left\{\hat{\tau}_T \geq \tilde{S}_T\right\} - \sum_{y=1}^K \frac{\alpha\epsilon}{K(1-\epsilon)}\mathbb{1}\left\{\hat{\tau}_T \geq S_{t,y}\right\} - \frac{1-\alpha}{1-\epsilon}\mathbb{1}\left\{\hat{\tau}_T \geq \tilde{S}_T\right\}$$

$$+ \sum_{y=1}^K \frac{(1-\alpha)\epsilon}{K(1-\epsilon)}\mathbb{1}\left\{\hat{\tau}_T \geq S_{t,y}\right\}$$

$$= \frac{\alpha}{1-\epsilon} - \sum_{y=1}^K \frac{\alpha\epsilon}{K(1-\epsilon)} = \alpha,$$

which follows that

$$\hat{\tau}_{T+1} = \hat{\tau}_T - \eta \cdot \nabla_{\hat{\tau}_T}\tilde{l}_{1-\alpha}(\hat{\tau}_T, \tilde{S}_T, \{S_{t,y}\}_{y=1}^K) = \hat{\tau}_T - \eta\alpha \in \left[1 - \eta\alpha, 1 - \frac{\eta}{1-\epsilon}\right] \subset \left[-\alpha\eta - \frac{\epsilon\eta}{1-\epsilon}, 1 - \alpha\eta + \frac{\eta}{1-\epsilon}\right]$$

**Case 3.** Consider the case where $\hat{\tau}_T \in [-\alpha\eta - \epsilon\eta/(1 - \epsilon), 0]$. The assumption that $\tilde{s}, s_y \in [0, 1]$ implies $\mathbb{1}\left\{\tilde{s} \leq \hat{\tau}_T\right\} = \mathbb{1}\left\{s_y \leq \hat{\tau}_T\right\} = 0$. Thus, we have

$$\nabla_{\hat{\tau}_T}\tilde{l}_{1-\alpha}(\hat{\tau}_T, \tilde{S}_T, \{S_{t,y}\}_{y=1}^K) = \frac{\alpha}{1-\epsilon}\mathbb{1}\left\{\hat{\tau}_T \geq \tilde{S}_T\right\} - \sum_{y=1}^K \frac{\alpha\epsilon}{K(1-\epsilon)}\mathbb{1}\left\{\hat{\tau}_T \geq S_{t,y}\right\} - \frac{1-\alpha}{1-\epsilon}\mathbb{1}\left\{\hat{\tau}_T \geq \tilde{S}_T\right\}$$

$$+ \sum_{y=1}^K \frac{(1-\alpha)\epsilon}{K(1-\epsilon)}\mathbb{1}\left\{\hat{\tau}_T \geq S_{t,y}\right\}$$

$$= -\frac{1-\alpha}{1-\epsilon}\mathbb{1}\left\{\tau < \tilde{s}\right\} + \sum_{y=1}^K \frac{(1-\alpha)\epsilon}{K(1-\epsilon)} = \alpha - 1,$$

which follows that

$$\hat{\tau}_{T+1} = \hat{\tau}_T - \eta \cdot \nabla_{\hat{\tau}_T}\tilde{l}_{1-\alpha}(\hat{\tau}_T, \tilde{S}_T, \{S_{t,y}\}_{y=1}^K) = \hat{\tau}_T + \eta(1-\alpha) \in \left[-\alpha\eta - \frac{\epsilon\eta}{1-\epsilon} + \eta(1-\alpha), \eta(1-\alpha)\right]$$

$$\subset \left[-\alpha\eta - \frac{\epsilon\eta}{1-\epsilon}, 1 - \alpha\eta + \frac{\eta}{1-\epsilon}\right]$$

Combining three cases, we can conclude that

$$-\alpha\eta - \frac{\epsilon\eta}{1-\epsilon} \leq \hat{\tau}_t \leq 1 - \alpha\eta + \frac{\eta}{1-\epsilon}$$

$\square$

**Proposition C.5** (Restatement of Proposition 3.3). *Consider online conformal prediction under uniform label noise with noise rate $\epsilon \in (0, 1)$. Given Assumptions 2.1 and 2.2, when updating the threshold according to Eq. (4), for any $\delta \in (0, 1)$ and $T \in \mathbb{N}^+$, the following bound holds with probability at least $1 - \delta$:*

$$\text{ExErr}(T) \leq \sqrt{\frac{\log(2/\delta)}{1 - \epsilon}} \cdot \frac{1}{\sqrt{T}} + \left(\frac{1}{\eta} - \alpha + \frac{\epsilon}{1 - \epsilon}\right) \cdot \frac{1}{T}.$$

*Proof.* The update rule of the threshold $\hat{\tau}_t$ gives us that

$$\hat{\tau}_{t+1} = \hat{\tau}_t - \eta_t \cdot \nabla_{\hat{\tau}_t} \tilde{l}_{1-\alpha}(\hat{\tau}_t, \tilde{S}_t, \{S_{t,y}\}_{y=1}^K) \quad \implies \quad \nabla_{\hat{\tau}_t} \tilde{l}_{1-\alpha}(\hat{\tau}_t, \tilde{S}_t, \{S_{t,y}\}_{y=1}^K) = \frac{1}{\eta}(\hat{\tau}_t - \hat{\tau}_{t+1}).$$

Accumulating from $t = 1$ to $t = T$ and taking absolute value gives

$$\left|\sum_{t=1}^T \nabla_{\hat{\tau}_t} \tilde{l}_{1-\alpha}(\hat{\tau}_t, \tilde{S}_t, \{S_{t,y}\}_{y=1}^K)\right| = \left|\sum_{t=1}^T \frac{1}{\eta}(\hat{\tau}_t - \hat{\tau}_{t+1})\right| = \frac{1}{\eta}|\hat{\tau}_1 - \hat{\tau}_{T+1}| \overset{(a)}{\leq} \frac{1}{\eta}\left(1 - \alpha\eta + \frac{\epsilon\eta}{1 - \epsilon}\right) = \frac{1}{\eta} - \alpha + \frac{\epsilon}{1 - \epsilon}$$

where (a) follows from the assumption that $\hat{\tau}_1 \in [0, 1]$ and Lemma C.4. In addition, Lemma G.5 gives us that

$$\mathbb{P}\left\{\left|\sum_{t=1}^T \mathbb{E}_{\tilde{S}_t, S_{t,y}}\left[\nabla_{\hat{\tau}_t}\tilde{l}_{1-\alpha}(\hat{\tau}_t, \tilde{S}_t, \{S_{t,y}\}_{y=1}^K)\right] - \sum_{t=1}^T \nabla_{\hat{\tau}_t}\tilde{l}_{1-\alpha}(\hat{\tau}_t, \tilde{S}_t, \{S_{t,y}\}_{y=1}^K)\right| \leq \frac{\sqrt{2T\log(2/\delta)}}{1 - \epsilon}\right\} \geq 1 - \delta$$

Thus, with at least $1 - \delta$ probability, we have

$$\left|\sum_{t=1}^T \mathbb{E}_{\tilde{S}_t, S_{t,y}}\left[\nabla_{\hat{\tau}_t}\tilde{l}_{1-\alpha}(\hat{\tau}_t, \tilde{S}_t, \{S_{t,y}\}_{y=1}^K)\right]\right|$$

$$= \left|\sum_{t=1}^T \mathbb{E}_{\tilde{S}_t, S_{t,y}}\left[\nabla_{\hat{\tau}_t}\tilde{l}_{1-\alpha}(\hat{\tau}_t, \tilde{S}_t, \{S_{t,y}\}_{y=1}^K)\right] - \sum_{t=1}^T \nabla_{\hat{\tau}_t}\tilde{l}_{1-\alpha}(\hat{\tau}_t, \tilde{S}_t, \{S_{t,y}\}_{y=1}^K) + \sum_{t=1}^T \nabla_{\hat{\tau}_t}\tilde{l}_{1-\alpha}(\hat{\tau}_t, \tilde{S}_t, \{S_{t,y}\}_{y=1}^K)\right|$$

$$\leq \left|\sum_{t=1}^T \mathbb{E}_{\tilde{S}_t, S_{t,y}}\left[\nabla_{\hat{\tau}_t}\tilde{l}_{1-\alpha}(\hat{\tau}_t, \tilde{S}_t, \{S_{t,y}\}_{y=1}^K)\right] - \sum_{t=1}^T \nabla_{\hat{\tau}_t}\tilde{l}_{1-\alpha}(\hat{\tau}_t, \tilde{S}_t, \{S_{t,y}\}_{y=1}^K)\right| + \left|\sum_{t=1}^T \nabla_{\hat{\tau}_t}\tilde{l}_{1-\alpha}(\hat{\tau}_t, \tilde{S}_t, \{S_{t,y}\}_{y=1}^K)\right|$$

$$\leq \frac{\sqrt{2T\log(2/\delta)}}{1 - \epsilon} + \frac{1}{\eta} + \alpha + \frac{\epsilon}{1 - \epsilon}$$

Applying the second property in Proposition 3.2, we have

$$\sum_{t=1}^T \mathbb{E}_{\tilde{S}, S_y}\left[\nabla_{\hat{\tau}_t}\tilde{l}_{1-\alpha}(\hat{\tau}_t, \tilde{S}, \{S_y\}_{y=1}^K)\right] = \sum_{t=1}^T \mathbb{E}_{S_t}\left[\nabla_{\hat{\tau}_t}l_{1-\alpha}(\hat{\tau}_t, S_t)\right] = \sum_{t=1}^T [\mathbb{P}\{S_t \leq \hat{\tau}_t\} - (1 - \alpha)]$$

which implies that with at least $1 - \delta$ probability,

$$\left|\sum_{t=1}^T [\mathbb{P}\{Y_t \notin \mathcal{C}_t(X_t)\} - \alpha]\right| = \left|\sum_{t=1}^T [\mathbb{P}\{S_t \leq \hat{\tau}_t\} - (1 - \alpha)]\right| \leq \frac{\sqrt{2T\log(2/\delta)}}{1 - \epsilon} + \frac{1}{\eta} - \alpha + \frac{\epsilon}{1 - \epsilon} \tag{9}$$

Therefore, we can conclude that

$$\text{ExErr}(T) = \left|\frac{1}{T}\sum_{t=1}^T \mathbb{P}\{Y_t \notin \mathcal{C}_t(X_t)\} - \alpha\right| \leq \sqrt{\frac{\log(2/\delta)}{1 - \epsilon}} \cdot \frac{1}{\sqrt{T}} + \left(\frac{1}{\eta} - \alpha + \frac{\epsilon}{1 - \epsilon}\right) \cdot \frac{1}{T}$$

holds with at least $1 - \delta$ probability. $\square$

## C.4. Proof for Proposition 3.4

**Proposition C.6** (Restatement of Proposition 3.4). *Consider online conformal prediction under uniform label noise with noise rate $\epsilon \in (0, 1)$. Given Assumptions 2.1 and 2.2, when updating the threshold according to Eq. (4), for any $\delta \in (0, 1)$ and $T \in \mathbb{N}^+$, the following bound holds with probability at least $1 - \delta$:*

$$\mathrm{EmErr(T)} \leq \frac{2 - \epsilon}{1 - \epsilon}\sqrt{2\log\left(\frac{4}{\delta}\right)} \cdot \frac{1}{\sqrt{T}} + \left(\frac{1}{\eta} - \alpha + \frac{\epsilon}{1 - \epsilon}\right) \cdot \frac{1}{T}.$$

*Proof.* Lemma G.3 gives us that

$$\mathbb{P}\left\{\left|\sum_{t=1}^{T}\mathbb{E}_{S_t}\left[\nabla_{\hat{\tau}_t}l_{1-\alpha}(\hat{\tau}_t, S_t)\right] - \sum_{t=1}^{T}\nabla_{\hat{\tau}_t}l_{1-\alpha}(\hat{\tau}_t, S_t)\right| \leq \sqrt{2T\log\left(\frac{4}{\delta}\right)}\right\} \geq 1 - \frac{\delta}{2}$$

Follows from Proposition 3.3 (see Eq. (9)), we know

$$\left|\sum_{t=1}^{T}\mathbb{E}_{S_t}\left[\nabla_{\hat{\tau}_t}l_{1-\alpha}(\hat{\tau}_t, S_t)\right]\right| = \left|\sum_{t=1}^{T}\mathbb{E}_{\tilde{S}, S_y}[\nabla_{\hat{\tau}_t}\tilde{l}_{1-\alpha}(\hat{\tau}_t, \tilde{S}, \{S_y\}_{y=1}^{K})]\right| \leq \frac{\sqrt{2T\log(4/\delta)}}{1 - \epsilon} + \frac{1}{\eta} - \alpha + \frac{\epsilon}{1 - \epsilon}$$

holds with at least $1 - \delta/2$ probability. In addition, recall that

$$\sum_{t=1}^{T}\nabla_{\hat{\tau}_t}l_{1-\alpha}(\hat{\tau}_t, S_t) = \sum_{t=1}^{T}\left[\mathbb{1}\left\{S_t \leq \hat{\tau}_t\right\} - (1 - \alpha)\right]$$

Thus, by union bound, we can obtain that with at least probability $1 - \delta$,

$$\left|\sum_{t=1}^{T}\nabla_{\hat{\tau}_t}l_{1-\alpha}(\hat{\tau}_t, S_t)\right| = \left|\sum_{t=1}^{T}\nabla_{\hat{\tau}_t}l_{1-\alpha}(\hat{\tau}_t, S_t) - \sum_{t=1}^{T}\mathbb{E}_{S_t}\left[\nabla_{\hat{\tau}_t}l_{1-\alpha}(\hat{\tau}_t, S_t)\right] + \sum_{t=1}^{T}\mathbb{E}_{S_t}\left[\nabla_{\hat{\tau}_t}l_{1-\alpha}(\hat{\tau}_t, S_t)\right]\right|$$

$$\leq \left|\sum_{t=1}^{T}\nabla_{\hat{\tau}_t}l_{1-\alpha}(\hat{\tau}_t, S_t) - \sum_{t=1}^{T}\mathbb{E}_{S_t}\left[\nabla_{\hat{\tau}_t}l_{1-\alpha}(\hat{\tau}_t, S_t)\right]\right| + \left|\sum_{t=1}^{T}\mathbb{E}_{S_t}\left[\nabla_{\hat{\tau}_t}l_{1-\alpha}(\hat{\tau}_t, S_t)\right]\right|$$

$$\leq \sqrt{2T\log\left(\frac{4}{\delta}\right)} + \frac{\sqrt{2T\log(4/\delta)}}{1 - \epsilon} + \frac{1}{\eta} - \alpha + \frac{\epsilon}{1 - \epsilon}$$

$$= \frac{2 - \epsilon}{1 - \epsilon} \cdot \sqrt{2T\log\left(\frac{4}{\delta}\right)} + \frac{1}{\eta} - \alpha + \frac{\epsilon}{1 - \epsilon}$$

Recall that

$$\left|\sum_{t=1}^{T}\left[\mathbb{1}\left\{Y_t \notin \mathcal{C}_t(X_t)\right\} - \alpha\right]\right| = \left|\sum_{t=1}^{T}\left[\mathbb{1}\left\{S_t \leq \hat{\tau}_t\right\} - (1 - \alpha)\right]\right| = \left|\sum_{t=1}^{T}\nabla_{\hat{\tau}_t}l_{1-\alpha}(\hat{\tau}_t, S_t)\right|$$

Therefore, we can conclude that

$$\mathrm{EmErr(T)} = \left|\frac{1}{T}\sum_{t=1}^{T}\mathbb{1}\left\{Y_t \notin \mathcal{C}_t(X_t)\right\} - \alpha\right| = \left|\frac{1}{T}\sum_{t=1}^{T}\nabla_{\hat{\tau}_t}l_{1-\alpha}(\hat{\tau}_t, S_t)\right|$$

$$\leq \frac{2 - \epsilon}{1 - \epsilon}\sqrt{2\log\left(\frac{4}{\delta}\right)} \cdot \frac{1}{\sqrt{T}} + \left(\frac{1}{\eta} - \alpha + \frac{\epsilon}{1 - \epsilon}\right) \cdot \frac{1}{T}$$

holds with at least $1 - \delta$ probability. $\square$

# D. Proof for Section 4

## D.1. Proof for Proposition 4.1

**Lemma D.1** (Lemma 1 in Angelopoulos et al. (2024))**.** *Given Assumptions 2.1 and 2.2, when updating the threshold according to Eq. (5), for any $T \in \mathbb{N}^+$, we have*

$$-\alpha \cdot \max_{1 \le t \le T-1} \eta_t \le \hat{\tau}_T \le 1 + (1 - \alpha) \cdot \max_{1 \le t \le T-1} \eta_t. \tag{10}$$

**Proposition D.2** (Restatement of Proposition 4.1)**.** *Consider online conformal prediction under uniform label noise with noise rate $\epsilon \in (0, 1)$. Given Assumptions 2.1 and 2.2, when updating the threshold according to Eq. (5), for any $\delta \in (0, 1)$ and $T \in \mathbb{N}^+$, the following bound holds with probability at least $1 - \delta$:*

$$\alpha - \frac{1}{T} \sum_{t=1}^{T} \mathbb{1}\left\{Y_t \notin \mathcal{C}_t(X_t)\right\} \le \frac{2 - \epsilon}{1 - \epsilon} \sqrt{2 \log\left(\frac{4}{\delta}\right)} \cdot \frac{1}{\sqrt{T}} + \frac{1 + \max\limits_{1 \le t \le T-1} \eta_t}{1 - \epsilon} \sum_{t=1}^{T} \left|\eta_t^{-1} - \eta_{t-1}^{-1}\right| \cdot \frac{1}{T} +$$

$$\frac{\epsilon}{1 - \epsilon} \cdot \frac{1}{T} \sum_{t=1}^{T} \left((1 - \alpha) - \frac{1}{K} \mathbb{E}\left[\mathcal{C}_t(X_t)\right]\right).$$

*Proof.* Consider the gradient of clean pinball loss:

$$\sum_{t=1}^{T} \nabla_{\tau_t} l_{1-\alpha}(\tau_t, S_t) = \sum_{t=1}^{T} \nabla_{\tau_t} l_{1-\alpha}(\tau_t, S_t) - \sum_{t=1}^{T} \mathbb{E}_S\left[\nabla_{\tau_t} l_{1-\alpha}(\tau_t, S_t)\right] + \sum_{t=1}^{T} \mathbb{E}_S\left[\nabla_{\tau_t} l_{1-\alpha}(\tau_t, S_t)\right]$$

$$\le \underbrace{\left|\sum_{t=1}^{T} \nabla_{\tau_t} l_{1-\alpha}(\tau_t, S_t) - \sum_{t=1}^{T} \mathbb{E}_S\left[\nabla_{\tau_t} l_{1-\alpha}(\tau_t, S_t)\right]\right|}_{(a)} + \underbrace{\sum_{t=1}^{T} \mathbb{E}_S\left[\nabla_{\tau_t} l_{1-\alpha}(\tau_t, S_t)\right]}_{(b)}$$

Part (a): Lemma G.3 gives us that

$$\mathbb{P}\left\{\left|\sum_{t=1}^{T} \mathbb{E}_{S_t}\left[\nabla_{\hat{\tau}_t} l_{1-\alpha}(\hat{\tau}_t, S_t)\right] - \sum_{t=1}^{T} \nabla_{\hat{\tau}_t} l_{1-\alpha}(\hat{\tau}_t, S_t)\right| \le \sqrt{2T \log\left(\frac{4}{\delta}\right)}\right\} \ge 1 - \frac{\delta}{2}.$$

Part (b): Recall that the expected gradient of the clean pinball loss satisfies

$$\mathbb{E}_S\left[\nabla_{\hat{\tau}_t} l_{1-\alpha}(\hat{\tau}_t, S_t)\right] = \mathbb{P}\left\{S_t \le \hat{\tau}_t\right\} - (1 - \alpha)$$

$$\overset{(a)}{=} \left(\frac{1}{1 - \epsilon} \mathbb{P}\left\{\tilde{S}_t \le \hat{\tau}_t\right\} - \frac{\epsilon}{K(1 - \epsilon)} \sum_{y=1}^{K} \mathbb{P}\left\{S_{t,y} \le \hat{\tau}_t\right\}\right) - (1 - \alpha)$$

$$= \frac{1}{1 - \epsilon} \left(\mathbb{P}\left\{\tilde{S}_t \le \hat{\tau}_t\right\} - (1 - \alpha)\right) - \frac{\epsilon}{1 - \epsilon} \left(\frac{1}{K} \sum_{y=1}^{K} \mathbb{P}\left\{S_{t,y} \le \hat{\tau}_t\right\} - (1 - \alpha)\right)$$

$$= \frac{1}{1 - \epsilon} \mathbb{E}_{\tilde{S}_t}\left[\nabla_{\hat{\tau}_t} l_{1-\alpha}(\hat{\tau}_t, \tilde{S}_t)\right] - \frac{\epsilon}{1 - \epsilon} \left(\frac{1}{K} \mathbb{E}\left[\mathcal{C}_t(X_t)\right] - (1 - \alpha)\right)$$

where (a) comes from Lemma G.1. In addition, Lemma G.3 implies that

$$\mathbb{P}\left\{\left|\sum_{t=1}^{T} \mathbb{E}_{\tilde{S}_t}\left[\nabla_{\hat{\tau}_t} l_{1-\alpha}(\hat{\tau}_t, \tilde{S}_t)\right] - \nabla_{\hat{\tau}_t} l_{1-\alpha}(\hat{\tau}_t, \tilde{S}_t)\right| \le \sqrt{2T \log\left(\frac{4}{\delta}\right)}\right\} \ge 1 - \frac{\delta}{2}$$

Thus, we can derive an upper bound for (b) as follows

$$\sum_{t=1}^{T} \mathbb{E}_S \left[ \nabla_{\hat{\tau}_t} l_{1-\alpha}(\hat{\tau}_t, S) \right]$$

$$= \frac{1}{1-\epsilon} \sum_{t=1}^{T} \mathbb{E}_{\tilde{S}} \left[ \nabla_{\hat{\tau}_t} l_{1-\alpha}(\hat{\tau}_t, \tilde{S}) \right] + \frac{\epsilon}{1-\epsilon} \sum_{t=1}^{T} \left( (1-\alpha) - \frac{1}{K} \mathbb{E} \left[ \mathcal{C}_t(X_t) \right] \right)$$

$$= \frac{1}{1-\epsilon} \left( \sum_{t=1}^{T} \mathbb{E}_{\tilde{S}} \left[ \nabla_{\hat{\tau}_t} l_{1-\alpha}(\hat{\tau}_t, \tilde{S}) \right] - \sum_{t=1}^{T} \nabla_{\hat{\tau}_t} l_{1-\alpha}(\hat{\tau}_t, \tilde{S}) + \sum_{t=1}^{T} \nabla_{\hat{\tau}_t} l_{1-\alpha}(\hat{\tau}_t, \tilde{S}) \right) + \frac{\epsilon}{1-\epsilon} \sum_{t=1}^{T} \left( (1-\alpha) - \frac{1}{K} \mathbb{E} \left[ \mathcal{C}_t(X_t) \right] \right)$$

$$\leq \frac{1}{1-\epsilon} \left| \sum_{t=1}^{T} \mathbb{E}_{\tilde{S}} \left[ \nabla_{\hat{\tau}_t} l_{1-\alpha}(\hat{\tau}_t, \tilde{S}) \right] - \sum_{t=1}^{T} \nabla_{\hat{\tau}_t} l_{1-\alpha}(\hat{\tau}_t, \tilde{S}) \right| + \frac{1}{1-\epsilon} \left| \sum_{t=1}^{T} \nabla_{\hat{\tau}_t} l_{1-\alpha}(\hat{\tau}_t, \tilde{S}) \right| + \frac{\epsilon}{1-\epsilon} \sum_{t=1}^{T} \left( (1-\alpha) - \frac{1}{K} \mathbb{E} \left[ \mathcal{C}_t(X_t) \right] \right)$$

$$\leq \frac{1}{1-\epsilon} \sqrt{2T \log \left( \frac{4}{\delta} \right)} + \frac{1}{1-\epsilon} \underbrace{\left| \sum_{t=1}^{T} \nabla_{\hat{\tau}_t} l_{1-\alpha}(\hat{\tau}_t, \tilde{S}) \right|}_{(c)} + \frac{\epsilon}{1-\epsilon} \sum_{t=1}^{T} \left( (1-\alpha) - \frac{1}{K} \mathbb{E} \left[ \mathcal{C}_t(X_t) \right] \right)$$

Then, we derive an upper bound for (c). Define $\eta_0^{-1} = 0$:

$$\left| \sum_{t=1}^{T} \nabla_{\hat{\tau}_t} l_{1-\alpha}(\hat{\tau}_t, S_t) \right| = \left| \sum_{t=1}^{T} \eta_t^{-1} \cdot \left( \eta_t \cdot \nabla_{\hat{\tau}_t} l_{1-\alpha}(\hat{\tau}_t, S_t) \right) \right|$$

$$= \left| \sum_{t=1}^{T} (\eta_t^{-1} - \eta_{t-1}^{-1}) \cdot \left( \sum_{s=t}^{T} \eta_s \cdot \nabla_{\hat{\tau}_s} l_{1-\alpha}(\hat{\tau}_s, S_s) \right) \right|$$

$$\overset{(a)}{=} \left| \sum_{t=1}^{T} (\eta_t^{-1} - \eta_{t-1}^{-1}) \cdot (\hat{\tau}_T - \hat{\tau}_t) \right|$$

$$\overset{(b)}{\leq} \sum_{t=1}^{T} \left| \eta_t^{-1} - \eta_{t-1}^{-1} \right| \cdot \left| \hat{\tau}_T - \hat{\tau}_t \right|$$

$$\overset{(c)}{\leq} (1 + \max_{1 \leq t \leq T-1} \eta_t) \sum_{t=1}^{T} \left| \eta_t^{-1} - \eta_{t-1}^{-1} \right|$$

where (a) comes from the update rule (Eq. (2)), (b) is due to triangle inequality, and (c) follows from Lemma D.1. Thus, we have

$$\sum_{t=1}^{T} \mathbb{E}_S \left[ \nabla_{\hat{\tau}_t} l_{1-\alpha}(\hat{\tau}_t, S) \right] \leq \frac{1}{1-\epsilon} \sqrt{2T \log \left( \frac{4}{\delta} \right)} + \frac{1 + \max_{1 \leq t \leq T-1} \eta_t}{1-\epsilon} \sum_{t=1}^{T} \left| \eta_t^{-1} - \eta_{t-1}^{-1} \right| + \frac{\epsilon}{1-\epsilon} \sum_{t=1}^{T} \left( (1-\alpha) - \frac{1}{K} \mathbb{E} \left[ \mathcal{C}_t(X_t) \right] \right)$$

By taking union bound and combining two parts, we can obtain that

$$\sum_{t=1}^{T} \nabla_{\tau_t} l_{1-\alpha}(\tau_t, S_t) \leq \sqrt{2T \log \left( \frac{4}{\delta} \right)} + \frac{1}{1-\epsilon} \sqrt{2T \log \left( \frac{4}{\delta} \right)} + \frac{1 + \max_{1 \leq t \leq T-1} \eta_t}{1-\epsilon} \sum_{t=1}^{T} \left| \eta_t^{-1} - \eta_{t-1}^{-1} \right| +$$

$$\frac{\epsilon}{1-\epsilon} \sum_{t=1}^{T} \left( \frac{1}{K} \mathbb{E} \left[ \mathcal{C}_t(X_t) \right] - (1-\alpha) \right)$$

$$= \frac{2-\epsilon}{1-\epsilon} \sqrt{2T \log \left( \frac{4}{\delta} \right)} + \frac{1 + \max_{1 \leq t \leq T-1} \eta_t}{1-\epsilon} \sum_{t=1}^{T} \left| \eta_t^{-1} - \eta_{t-1}^{-1} \right| + \frac{\epsilon}{1-\epsilon} \sum_{t=1}^{T} \left( (1-\alpha) - \frac{1}{K} \mathbb{E} \left[ \mathcal{C}_t(X_t) \right] \right)$$

holds with at least $1 - \delta$ probability. Recall that

$$\sum_{t=1}^{T} \nabla_{\tau_t} l_{1-\alpha}(\tau_t, S_t) = \sum_{t=1}^{T} \left( \mathbb{1}\left\{ S_t \leq \tau_t \right\} - (1-\alpha) \right) = \sum_{t=1}^{T} \left( -\mathbb{1}\left\{ S_t \geq \tau_t \right\} + \alpha \right) = \sum_{t=1}^{T} \left( -\mathbb{1}\left\{ Y_t \notin \mathcal{C}_t(X_t) \right\} + \alpha \right)$$

We can conclude that

$$\alpha - \frac{1}{T}\sum_{t=1}^{T}\mathbb{1}\left\{Y_t \notin \mathcal{C}_t(X_t)\right\} \leq \frac{2-\epsilon}{1-\epsilon}\sqrt{2\log\left(\frac{4}{\delta}\right)} \cdot \frac{1}{\sqrt{T}} + \frac{1 + \max_{1\leq t\leq T-1}\eta_t}{1-\epsilon}\sum_{t=1}^{T}\left|\eta_t^{-1} - \eta_{t-1}^{-1}\right| \cdot \frac{1}{T} +$$

$$\frac{\epsilon}{1-\epsilon} \cdot \frac{1}{T}\sum_{t=1}^{T}\left((1-\alpha) - \frac{1}{K}\mathbb{E}\left[\mathcal{C}_t(X_t)\right]\right)$$

$\square$

### D.2. Proof for Proposition 4.3

**Proposition D.3** (Restatement of Proposition 4.3). *Consider online conformal prediction under uniform label noise with noise rate $\epsilon \in (0,1)$. Given Assumptions 2.1 and 2.2, when updating the threshold according to Eq. (6), for any $\delta \in (0,1)$ and $T \in \mathbb{N}^+$, the following bound holds with probability at least $1-\delta$:*

$$\mathrm{ExErr(T)} \leq \sqrt{\frac{\log(2/\delta)}{1-\epsilon}} \cdot \frac{1}{\sqrt{T}} + \left[\left(1 + \max_{1\leq t\leq T-1}\eta_t \cdot \frac{1+\epsilon}{1-\epsilon}\right)\sum_{t=1}^{T}\left|\eta_t^{-1} - \eta_{t-1}^{-1}\right|\right] \cdot \frac{1}{T}.$$

*Proof.* Define $\eta_0^{-1} = 0$:

$$\left|\sum_{t=1}^{T}\nabla_{\hat{\tau}_t}\tilde{l}_{1-\alpha}(\hat{\tau}_t, \tilde{S}_t, \{S_{t,y}\}_{y=1}^{K})\right| = \left|\sum_{t=1}^{T}\eta_t^{-1} \cdot \left(\eta_t \cdot \nabla_{\hat{\tau}_t}\tilde{l}_{1-\alpha}(\hat{\tau}_t, \tilde{S}_t, \{S_{t,y}\}_{y=1}^{K})\right)\right|$$

$$= \left|\sum_{t=1}^{T}(\eta_t^{-1} - \eta_{t-1}^{-1}) \cdot \left(\sum_{s=t}^{T}\eta_s \cdot \nabla_{\hat{\tau}_s}\tilde{l}_{1-\alpha}(\hat{\tau}_s, \tilde{S}_s, \{S_{y,s}\}_{y=1}^{K})\right)\right|$$

$$\overset{(a)}{=} \left|\sum_{t=1}^{T}(\eta_t^{-1} - \eta_{t-1}^{-1}) \cdot (\hat{\tau}_T - \hat{\tau}_t)\right|$$

$$\overset{(b)}{\leq} \sum_{t=1}^{T}\left|\eta_t^{-1} - \eta_{t-1}^{-1}\right| \cdot \left|\hat{\tau}_T - \hat{\tau}_t\right|$$

$$\overset{(c)}{\leq} \left(1 + \max_{1\leq t\leq T-1}\eta_t \cdot \frac{1+\epsilon}{1-\epsilon}\right)\sum_{t=1}^{T}\left|\eta_t^{-1} - \eta_{t-1}^{-1}\right|$$

where (a) comes from the update rule (Eq. (6)), (b) is due to triangle inequality, and (c) follows from Lemma D.1. In addition, Lemma G.5 gives us that

$$\mathbb{P}\left\{\left|\sum_{t=1}^{T}\mathbb{E}_{\tilde{S}_t, S_{t,y}}\left[\nabla_{\hat{\tau}_t}\tilde{l}_{1-\alpha}(\hat{\tau}_t, \tilde{S}_t, \{S_{t,y}\}_{y=1}^{K})\right] - \sum_{t=1}^{T}\nabla_{\hat{\tau}_t}\tilde{l}_{1-\alpha}(\hat{\tau}_t, \tilde{S}_t, \{S_{t,y}\}_{y=1}^{K})\right| \leq \frac{\sqrt{2T\log(2/\delta)}}{1-\epsilon}\right\} \geq 1-\delta$$

Thus, with at least $1-\delta$ probability, we have

$$\left|\sum_{t=1}^{T}\mathbb{E}_{\tilde{S}_t, S_{t,y}}\left[\nabla_{\hat{\tau}_t}\tilde{l}_{1-\alpha}(\hat{\tau}_t, \tilde{S}_t, \{S_{t,y}\}_{y=1}^{K})\right]\right|$$

$$= \left|\sum_{t=1}^{T}\mathbb{E}_{\tilde{S}_t, S_{t,y}}\left[\nabla_{\hat{\tau}_t}\tilde{l}_{1-\alpha}(\hat{\tau}_t, \tilde{S}_t, \{S_{t,y}\}_{y=1}^{K})\right] - \sum_{t=1}^{T}\nabla_{\hat{\tau}_t}\tilde{l}_{1-\alpha}(\hat{\tau}_t, \tilde{S}_t, \{S_{t,y}\}_{y=1}^{K}) + \sum_{t=1}^{T}\nabla_{\hat{\tau}_t}\tilde{l}_{1-\alpha}(\hat{\tau}_t, \tilde{S}_t, \{S_{t,y}\}_{y=1}^{K})\right|$$

$$\leq \left|\sum_{t=1}^{T}\mathbb{E}_{\tilde{S}_t, S_{t,y}}\left[\nabla_{\hat{\tau}_t}\tilde{l}_{1-\alpha}(\hat{\tau}_t, \tilde{S}_t, \{S_{t,y}\}_{y=1}^{K})\right] - \sum_{t=1}^{T}\nabla_{\hat{\tau}_t}\tilde{l}_{1-\alpha}(\hat{\tau}_t, \tilde{S}_t, \{S_{t,y}\}_{y=1}^{K})\right| + \left|\sum_{t=1}^{T}\nabla_{\hat{\tau}_t}\tilde{l}_{1-\alpha}(\hat{\tau}_t, \tilde{S}_t, \{S_{t,y}\}_{y=1}^{K})\right|$$

$$\leq \frac{\sqrt{2T\log(2/\delta)}}{1-\epsilon} + \left(1 + \max_{1\leq t\leq T-1}\eta_t \cdot \frac{1+\epsilon}{1-\epsilon}\right)\sum_{t=1}^{T}\left|\eta_t^{-1} - \eta_{t-1}^{-1}\right|$$

Applying the second property in Proposition 3.2, we have

$$\sum_{t=1}^{T} \mathbb{E}_{\tilde{S}, S_y} \left[ \nabla_{\hat{\tau}_t} \tilde{l}_{1-\alpha}(\hat{\tau}_t, \tilde{S}, \{S_y\}_{y=1}^{K}) \right] = \sum_{t=1}^{T} \mathbb{E}_{S_t} \left[ \nabla_{\hat{\tau}_t} l_{1-\alpha}(\hat{\tau}_t, S_t) \right] = \sum_{t=1}^{T} \left[ \mathbb{P}\{S_t \leq \hat{\tau}_t\} - (1-\alpha) \right]$$

which implies that with at least $1 - \delta$ probability,

$$\left| \sum_{t=1}^{T} \left[ \mathbb{P}\{Y_t \notin \mathcal{C}_t(X_t)\} - \alpha \right] \right| = \left| \sum_{t=1}^{T} \left[ \mathbb{P}\{S_t \leq \hat{\tau}_t\} - (1-\alpha) \right] \right|$$
$$\leq \frac{\sqrt{2T \log(2/\delta)}}{1-\epsilon} + \left( 1 + \max_{1 \leq t \leq T-1} \eta_t \cdot \frac{1+\epsilon}{1-\epsilon} \right) \sum_{t=1}^{T} |\eta_t^{-1} - \eta_{t-1}^{-1}| \tag{11}$$

Therefore, we can conclude that

$$\mathrm{ExErr(T)} = \left| \frac{1}{T} \sum_{t=1}^{T} \mathbb{P}\{Y_t \notin \mathcal{C}_t(X_t)\} - \alpha \right| \leq \sqrt{\frac{\log(2/\delta)}{1-\epsilon}} \cdot \frac{1}{\sqrt{T}} + \left[ \left( 1 + \max_{1 \leq t \leq T-1} \eta_t \cdot \frac{1+\epsilon}{1-\epsilon} \right) \sum_{t=1}^{T} |\eta_t^{-1} - \eta_{t-1}^{-1}| \right] \cdot \frac{1}{T}$$

holds with at least $1 - \delta$ probability. $\square$

### D.3. Proof for Proposition 4.4

**Proposition D.4** (Restatement of Proposition 4.4). *Consider online conformal prediction under uniform label noise with noise rate $\epsilon \in (0, 1)$. Given Assumptions 2.1 and 2.2, when updating the threshold according to Eq. (6), for any $\delta \in (0, 1)$ and $T \in \mathbb{N}^{+}$, the following bound holds with probability at least $1 - \delta$:*

$$\mathrm{EmErr(T)} \leq \frac{2 - \epsilon}{1 - \epsilon} \sqrt{2 \log\left(\frac{4}{\delta}\right)} \cdot \frac{1}{\sqrt{T}} + \left( 1 + \max_{1 \leq t \leq T-1} \eta_t \cdot \frac{1+\epsilon}{1-\epsilon} \right) \sum_{t=1}^{T} |\eta_t^{-1} - \eta_{t-1}^{-1}| \cdot \frac{1}{T}.$$

*Proof.* Lemma G.3 gives us that

$$\mathbb{P}\left\{ \left| \sum_{t=1}^{T} \mathbb{E}_{S_t} \left[ \nabla_{\hat{\tau}_t} l_{1-\alpha}(\hat{\tau}_t, S_t) \right] - \sum_{t=1}^{T} \nabla_{\hat{\tau}_t} l_{1-\alpha}(\hat{\tau}_t, S_t) \right| \leq \sqrt{2T \log\left(\frac{4}{\delta}\right)} \right\} \geq 1 - \frac{\delta}{2}$$

Follows from Proposition 4.3 (see Eq. (11)), we know

$$\left| \sum_{t=1}^{T} \mathbb{E}_{S_t} \left[ \nabla_{\hat{\tau}_t} l_{1-\alpha}(\hat{\tau}_t, S_t) \right] \right| = \left| \sum_{t=1}^{T} \mathbb{E}_{\tilde{S}, S_y} \left[ \nabla_{\hat{\tau}_t} \tilde{l}_{1-\alpha}(\hat{\tau}_t, \tilde{S}, \{S_y\}_{y=1}^{K}) \right] \right|$$
$$\leq \frac{\sqrt{2T \log(2/\delta)}}{1-\epsilon} + \left( 1 + \max_{1 \leq t \leq T-1} \eta_t \cdot \frac{1+\epsilon}{1-\epsilon} \right) \sum_{t=1}^{T} |\eta_t^{-1} - \eta_{t-1}^{-1}|$$

holds with at least $1 - \delta/2$ probability. In addition, recall that

$$\sum_{t=1}^{T} \nabla_{\hat{\tau}_t} l_{1-\alpha}(\hat{\tau}_t, S_t) = \sum_{t=1}^{T} \left[ \mathbb{1}\{S_t \leq \hat{\tau}_t\} - (1-\alpha) \right]$$

Thus, by union bound, we can obtain that with at least probability $1 - \delta$,

$$\left| \sum_{t=1}^{T} \nabla_{\hat{\tau}_t} l_{1-\alpha}(\hat{\tau}_t, S_t) \right| = \left| \sum_{t=1}^{T} \nabla_{\hat{\tau}_t} l_{1-\alpha}(\hat{\tau}_t, S_t) - \sum_{t=1}^{T} \mathbb{E}_{S_t} \left[ \nabla_{\hat{\tau}_t} l_{1-\alpha}(\hat{\tau}_t, S_t) \right] + \sum_{t=1}^{T} \mathbb{E}_{S_t} \left[ \nabla_{\hat{\tau}_t} l_{1-\alpha}(\hat{\tau}_t, S_t) \right] \right|$$

$$\leq \left| \sum_{t=1}^{T} \nabla_{\hat{\tau}_t} l_{1-\alpha}(\hat{\tau}_t, S_t) - \sum_{t=1}^{T} \mathbb{E}_{S_t} \left[ \nabla_{\hat{\tau}_t} l_{1-\alpha}(\hat{\tau}_t, S_t) \right] \right| + \left| \sum_{t=1}^{T} \mathbb{E}_{S_t} \left[ \nabla_{\hat{\tau}_t} l_{1-\alpha}(\hat{\tau}_t, S_t) \right] \right|$$

$$\leq \sqrt{2T \log \left( \frac{4}{\delta} \right)} + \frac{\sqrt{2T \log(2/\delta)}}{1 - \epsilon} + \left( 1 + \max_{1 \leq t \leq T-1} \eta_t \cdot \frac{1 + \epsilon}{1 - \epsilon} \right) \sum_{t=1}^{T} \left| \eta_t^{-1} - \eta_{t-1}^{-1} \right|$$

$$= \frac{2 - \epsilon}{1 - \epsilon} \cdot \sqrt{2T \log \left( \frac{4}{\delta} \right)} + \left( 1 + \max_{1 \leq t \leq T-1} \eta_t \cdot \frac{1 + \epsilon}{1 - \epsilon} \right) \sum_{t=1}^{T} \left| \eta_t^{-1} - \eta_{t-1}^{-1} \right|$$

Recall that

$$\left| \sum_{t=1}^{T} \left[ \mathbb{1} \left\{ Y_t \notin \mathcal{C}_t(X_t) \right\} - \alpha \right] \right| = \left| \sum_{t=1}^{T} \left[ \mathbb{1} \left\{ S_t \leq \hat{\tau}_t \right\} - (1 - \alpha) \right] \right| = \left| \sum_{t=1}^{T} \nabla_{\hat{\tau}_t} l_{1-\alpha}(\hat{\tau}_t, S_t) \right|$$

Therefore, we can conclude that

$$\text{EmErr(T)} = \left| \frac{1}{T} \sum_{t=1}^{T} \mathbb{1} \left\{ Y_t \notin \mathcal{C}_t(X_t) \right\} - \alpha \right| = \left| \frac{1}{T} \sum_{t=1}^{T} \nabla_{\hat{\tau}_t} l_{1-\alpha}(\hat{\tau}_t, S_t) \right|$$

$$\leq \frac{2 - \epsilon}{1 - \epsilon} \sqrt{2 \log \left( \frac{4}{\delta} \right)} \cdot \frac{1}{\sqrt{T}} + \left( 1 + \max_{1 \leq t \leq T-1} \eta_t \cdot \frac{1 + \epsilon}{1 - \epsilon} \right) \sum_{t=1}^{T} \left| \eta_t^{-1} - \eta_{t-1}^{-1} \right| \cdot \frac{1}{T}$$

holds with at least $1 - \delta$ probability. $\qquad\qquad\qquad\qquad\qquad\qquad\qquad\qquad\qquad\qquad\qquad\qquad\qquad$ $\square$

# E. Proof for Appendix B.1

**Proposition E.1** (Restatement of Proposition B.1). *Consider online conformal prediction under uniform label noise with noise rate $\epsilon \in (0,1)$. Given Assumptions 2.1 and 2.2, when updating the threshold according to Eq. (6), for any $T \in \mathbb{N}^+$, we have:*

$$\sum_{t=1}^{T} \left( \tilde{l}_{1-\alpha}(\hat{\tau}_t, \tilde{S}_t, \{S_{t,y}\}_{y=1}^{K}) - \tilde{l}_{1-\alpha}(\tau^*, \tilde{S}_t, \{S_{t,y}\}_{y=1}^{K}) \right) \le \frac{1}{2\eta_T} \left( 1 + \max_{1 \le t \le T-1} \eta_t \cdot \frac{1+\epsilon}{1-\epsilon} \right)^2 + \left( \frac{1+\epsilon}{1-\epsilon} \right)^2 \cdot \sum_{t=1}^{T} \frac{\eta_t}{2}.$$

*Proof.* The update rule (Eq. (1)) gives us that

$$(\hat{\tau}_{t+1} - \tau^*)^2 = (\hat{\tau}_t - \tau^*)^2 + 2\eta_t(\tau^* - \hat{\tau}_t) \cdot \nabla_{\hat{\tau}_t} \tilde{l}_{1-\alpha}(\hat{\tau}_t, \tilde{S}_t, \{S_{t,y}\}_{y=1}^{K}) + \eta_t^2 \cdot \left( \nabla_{\hat{\tau}_t} \tilde{l}_{1-\alpha}(\hat{\tau}_t, \tilde{S}_t, \{S_{t,y}\}_{y=1}^{K}) \right)^2.$$

Recall that the robust pinball loss is defined as

$$\tilde{l}_{1-\alpha}(\tau, \tilde{S}, \{S_y\}_{y=1}^{K}) = \frac{1}{1-\epsilon} l_{1-\alpha}(\tau, \tilde{S}) + \frac{\epsilon}{K(1-\epsilon)} \sum_{y=1}^{K} l_{1-\alpha}(\tau, S_y).$$

Since pinball loss is convex, robust pinball loss inherits the convexity property. Thus, we have

$$(\tau^* - \hat{\tau}_t) \cdot \nabla_{\hat{\tau}_t} \tilde{l}_{1-\alpha}(\hat{\tau}_t, \tilde{S}_t, \{S_{t,y}\}_{y=1}^{K}) \le \tilde{l}_{1-\alpha}(\tau^*, \tilde{S}_t, \{S_{t,y}\}_{y=1}^{K}) - \tilde{l}_{1-\alpha}(\hat{\tau}_t, \tilde{S}_t, \{S_{t,y}\}_{y=1}^{K}).$$

It follows that

$$(\hat{\tau}_{t+1} - \tau^*)^2 \le (\hat{\tau}_t - \tau^*)^2 + 2\eta_t \cdot \left( \tilde{l}_{1-\alpha}(\tau^*, \tilde{S}_t, \{S_{t,y}\}_{y=1}^{K}) - \tilde{l}_{1-\alpha}(\hat{\tau}_t, \tilde{S}_t, \{S_{t,y}\}_{y=1}^{K}) \right) +$$
$$\eta_t^2 \cdot \left( \nabla_{\hat{\tau}_t} \tilde{l}_{1-\alpha}(\hat{\tau}_t, \tilde{S}_t, \{S_{t,y}\}_{y=1}^{K}) \right)^2. \tag{12}$$

Following from Lemma G.2, we have

$$\left( \nabla_{\hat{\tau}} \tilde{l}_{1-\alpha}(\hat{\tau}_t, \tilde{S}_t, \{S_{t,y}\}_{y=1}^{K}) \right)^2 \le \left( \frac{1+\epsilon}{1-\epsilon} \right)^2.$$

Dividing this inequality by $\eta_t$ and summing over $t = 1, 2, \cdots, T$ provides

$$\sum_{t=1}^{T} \left( \tilde{l}_{1-\alpha}(\hat{\tau}_t, \tilde{S}_t, \{S_{t,y}\}_{y=1}^{K}) - \tilde{l}_{1-\alpha}(\tau^*, \tilde{S}_t, \{S_{t,y}\}_{y=1}^{K}) \right)$$
$$\le \sum_{t=1}^{T} \left( \frac{1}{2\eta_t}(\hat{\tau}_t - \tau^*)^2 - \frac{1}{2\eta_t}(\hat{\tau}_{t+1} - \tau^*)^2 \right) + \left( \frac{1+\epsilon}{1-\epsilon} \right)^2 \cdot \sum_{t=1}^{T} \frac{\eta_t}{2}$$
$$= \frac{1}{2\eta_1}(\hat{\tau}_1 - \tau^*)^2 - \frac{1}{2\eta_{T+1}}(\hat{\tau}_{T+1} - \tau^*)^2 + \sum_{t=1}^{T-1} \left( \frac{1}{2\eta_{t+1}} - \frac{1}{2\eta_t} \right) (\hat{\tau}_t - \tau^*)^2 + \left( \frac{1+\epsilon}{1-\epsilon} \right)^2 \cdot \sum_{t=1}^{T} \frac{\eta_t}{2}.$$

Lemma G.4 gives us that

$$(\hat{\tau}_t - \tau^*)^2 \le \left( 1 + \max_{1 \le t \le T-1} \eta_t \cdot \frac{1+\epsilon}{1-\epsilon} \right)^2,$$

which follows that

$$\sum_{t=1}^{T} \left( \tilde{l}_{1-\alpha}(\hat{\tau}_t, \tilde{S}_t, \{S_{t,y}\}_{y=1}^{K}) - \tilde{l}_{1-\alpha}(\tau^*, \tilde{S}_t, \{S_{t,y}\}_{y=1}^{K}) \right)$$
$$\le \frac{1}{2\eta_1} \left( 1 + \max_{1 \le t \le T-1} \eta_t \cdot \frac{1+\epsilon}{1-\epsilon} \right)^2 + \left( \frac{1}{2\eta_T} - \frac{1}{2\eta_1} \right) \cdot \left( 1 + \max_{1 \le t \le T-1} \eta_t \cdot \frac{1+\epsilon}{1-\epsilon} \right)^2 + \left( \frac{1+\epsilon}{1-\epsilon} \right)^2 \cdot \sum_{t=1}^{T} \frac{\eta_t}{2}$$
$$= \frac{1}{2\eta_T} \left( 1 + \max_{1 \le t \le T-1} \eta_t \cdot \frac{1+\epsilon}{1-\epsilon} \right)^2 + \left( \frac{1+\epsilon}{1-\epsilon} \right)^2 \cdot \sum_{t=1}^{T} \frac{\eta_t}{2}.$$

$\square$

## F. Proof for Appendix B.2

**Lemma F.1.** *Under Assumption B.2, the pinball loss satisfies*

$$|\hat{\tau} - \tau|^q \leq \frac{q(1-q)}{b} \left(\frac{1}{2\varepsilon_0}\right)^q \cdot E\left[l_{1-\alpha}(\hat{\tau}, S) - l_{1-\alpha}(\tau, S)\right]$$

*Proof.* We employ the proof technique from Lemma C.4 in Bhatnagar et al. (2023). The Assumption B.2 gives us that

$$\mathbb{P}\{R = \hat{\gamma}\} \geq b|\hat{\gamma} - \gamma|^{q-2} \quad \Longleftrightarrow \quad \mathbb{P}\{S = \hat{\tau}\} \geq 2b\,|2(\hat{\tau} - \tau)|^{q-2}, \quad \hat{\gamma} = 2\hat{\tau} - 1$$

By Theorem 2.7 of Steinwart & Christmann (2011), we have

$$|\hat{\gamma} - \gamma| \leq 2^{1-1/q} q^{1/q} \gamma^{-1/q} \cdot \left(E\left[l_{1-\alpha}(\hat{\gamma}, R) - l_{1-\alpha}(\gamma, R)\right]\right)^{1/q} = 2\left(\frac{q(1-q)}{b}\left(\frac{1}{2\varepsilon_0}\right)^q \cdot E\left[l_{1-\alpha}(\hat{\gamma}, R) - l_{1-\alpha}(\gamma, R)\right]\right)^{1/q}$$

Since $|\hat{\gamma} - \gamma| = 2|\hat{\tau} - \tau|$ and $l_{1-\alpha}(\hat{\gamma}, R) = l_{1-\alpha}(\hat{\tau}, S)$, we can obtain

$$|\hat{\tau} - \tau|^q \leq \frac{q(1-q)}{b}\left(\frac{1}{2\varepsilon_0}\right)^q \cdot E\left[l_{1-\alpha}(\hat{\tau}, S) - l_{1-\alpha}(\tau, S)\right]$$

$\square$

**Proposition F.2** (Restatement of Proposition B.3). *Consider online conformal prediction under uniform label noise with noise rate $\epsilon \in (0,1)$. Given Assumptions 2.1, 2.2 and B.2, when updating the threshold according to Eq. (6), for any $T \in \mathbb{N}^+$, we have:*

$$|\bar{\tau} - \tau|^q \leq \frac{q(1-q)}{b}\left(\frac{1}{2\varepsilon_0}\right)^q \cdot \left[\frac{(\hat{\tau}_1 - \tau^*)^2}{2\sum_{t=1}^T \eta_t} + \frac{\sum_{t=1}^T \eta_t^2}{\sum_{t=1}^T \eta_t}\cdot\left(\frac{1+\epsilon}{1-\epsilon}\right)^2\right]$$

*where $\bar{\tau} = \sum_{t=1}^T \eta_t\hat{\tau}_t / \sum_{t=1}^T \eta_t$.*

*Proof.* We begin our proof from Eq. (12):

$$(\hat{\tau}_{t+1} - \tau)^2 \leq (\hat{\tau}_t - \tau)^2 + 2\eta_t \cdot \left(\tilde{l}_{1-\alpha}(\tau, \tilde{S}_t, \{S_{t,y}\}_{y=1}^K) - \tilde{l}_{1-\alpha}(\hat{\tau}_t, \tilde{S}_t, \{S_{t,y}\}_{y=1}^K)\right) + \eta_t^2 \cdot \left(\nabla_{\hat{\tau}_t}\tilde{l}_{1-\alpha}(\hat{\tau}_t, \tilde{S}_t, \{S_{t,y}\}_{y=1}^K)\right)^2.$$

By taking expectation condition on $(X_t, Y_t)$ (or equivalently on $\tilde{S}_t$ and $\{S_{t,y}\}_{y=1}^K$), and applying Lemma G.2 and Proposition 3.2, we have

$$\mathbb{E}\left[(\hat{\tau}_{t+1} - \tau^*)^2\right] \leq (\hat{\tau}_t - \tau^*)^2 + 2\eta_t \cdot \left(\mathbb{E}\left[l_{1-\alpha}(\tau, S)\right] - \mathbb{E}\left[l_{1-\alpha}(\hat{\tau}_t, S)\right]\right) + \eta_t^2 \cdot \left(\frac{1+\epsilon}{1-\epsilon}\right)^2$$

By rearranging and taking expectation, we have

$$2\eta_t \cdot \left(\mathbb{E}\left[l_{1-\alpha}(\hat{\tau}_t, S) - l_{1-\alpha}(\tau, S)\right]\right) \leq \mathbb{E}\left[(\hat{\tau}_t - \tau^*)^2\right] - \mathbb{E}\left[(\hat{\tau}_{t+1} - \tau^*)^2\right] + \eta_t^2 \cdot \left(\frac{1+\epsilon}{1-\epsilon}\right)^2$$

Summing over $t = 1, 2, \cdots, T$ provides

$$2\sum_{t=1}^T \eta_t \cdot \left(\mathbb{E}\left[l_{1-\alpha}(\hat{\tau}_t, S) - l_{1-\alpha}(\tau, S)\right]\right) \leq (\hat{\tau}_1 - \tau^*)^2 - \mathbb{E}\left[(\hat{\tau}_{t+1} - \tau^*)^2\right] + \left(\frac{1+\epsilon}{1-\epsilon}\right)^2 \cdot \sum_{t=1}^T \eta_t^2$$

$$\leq (\hat{\tau}_1 - \tau^*)^2 + \left(\frac{1+\epsilon}{1-\epsilon}\right)^2 \cdot \sum_{t=1}^T \eta_t^2$$

Let us denote $\bar{\tau} := \sum_{t=1}^{T} \eta_t \hat{\tau}_t / \sum_{t=1}^{T} \eta_t$. Applying Jensen's inequality and dividing both sides by $2 \sum_{t=1}^{T} \eta_t$ gives

$$\mathbb{E}\left[l_{1-\alpha}\left(\bar{\tau}, S\right) - l_{1-\alpha}(\tau, S)\right] \leq \sum_{t=1}^{T} \frac{\eta_t}{\sum_{t=1}^{T} \eta_t} \cdot \left(\mathbb{E}\left[l_{1-\alpha}(\hat{\tau}_t, S) - l_{1-\alpha}(\tau, S)\right]\right)$$

$$\leq \frac{(\hat{\tau}_1 - \tau^*)^2}{2\sum_{t=1}^{T} \eta_t} + \frac{\sum_{t=1}^{T} \eta_t^2}{\sum_{t=1}^{T} \eta_t} \cdot \left(\frac{1+\epsilon}{1-\epsilon}\right)^2$$

Continuing from Lemma F.1, we can conclude that

$$|\bar{\tau} - \tau|^q \leq \frac{q(1-q)}{b} \left(\frac{1}{2\varepsilon_0}\right)^q \cdot \left[\frac{(\hat{\tau}_1 - \tau^*)^2}{2\sum_{t=1}^{T} \eta_t} + \frac{\sum_{t=1}^{T} \eta_t^2}{\sum_{t=1}^{T} \eta_t} \cdot \left(\frac{1+\epsilon}{1-\epsilon}\right)^2\right]$$

$\square$

## G. Helpful lemmas

**Lemma G.1.** *The distribution of the true non-conformity score, noise non-conformity score, and scores of all classes satisfy the following relationship:*

$$(1)\mathbb{P}\left\{S = s\right\} = \frac{1}{1-\epsilon}\mathbb{P}\left\{\tilde{S} = s\right\} - \frac{\epsilon}{K(1-\epsilon)}\sum_{y=1}^{K}\mathbb{P}\left\{S_y = s\right\};$$

$$(2)\mathbb{P}\left\{S \leq s\right\} = \frac{1}{1-\epsilon}\mathbb{P}\left\{\tilde{S} \leq s\right\} - \frac{\epsilon}{K(1-\epsilon)}\sum_{y=1}^{K}\mathbb{P}\left\{S_y \leq s\right\};$$

$$(3)\mathbb{P}\left\{S > s\right\} = \frac{1}{1-\epsilon}\mathbb{P}\left\{\tilde{S} > s\right\} - \frac{\epsilon}{K(1-\epsilon)}\sum_{y=1}^{K}\mathbb{P}\left\{S_y > s\right\},$$

*where $S$, $\tilde{S}$, and $S_y$ denote the true score, noisy score, and score for class $y$ respectively.*

*Proof.* **(1):**

$$\mathbb{P}\left\{\tilde{S} = s\right\} = \mathbb{P}\left\{\tilde{S} = s | \tilde{S} = S\right\} \cdot \mathbb{P}\left\{\tilde{S} = S\right\} + \mathbb{P}\left\{\tilde{S} = s | \tilde{S} = \bar{S}\right\} \cdot \mathbb{P}\left\{\tilde{S} = \bar{S}\right\}$$

$$= \mathbb{P}\left\{S = s\right\} \cdot \mathbb{P}\left\{\tilde{Y} = Y\right\} + \mathbb{P}\left\{\bar{S} = s\right\} \cdot \mathbb{P}\left\{\tilde{Y} = \bar{Y}\right\}$$

$$= (1-\epsilon)\mathbb{P}\left\{S = s\right\} + \epsilon\mathbb{P}\left\{\bar{S} = s\right\},$$

which follows that

$$\mathbb{P}\left\{S = s\right\} = \frac{1}{1-\epsilon}\mathbb{P}\left\{\tilde{S} = s\right\} - \frac{\epsilon}{1-\epsilon}\mathbb{P}\left\{\bar{S} = s\right\} = \frac{1}{1-\epsilon}\mathbb{P}\left\{\tilde{S} = s\right\} - \frac{\epsilon}{K(1-\epsilon)}\sum_{y=1}^{K}\mathbb{P}\left\{S_y = s\right\}.$$

**(2):**

$$\mathbb{P}\left\{\tilde{S} \leq s\right\} = \mathbb{P}\left\{\tilde{S} \leq s | \tilde{S} = S\right\} \cdot \mathbb{P}\left\{\tilde{S} = S\right\} + \mathbb{P}\left\{\tilde{S} \leq s | \tilde{S} = \bar{S}\right\} \cdot \mathbb{P}\left\{\tilde{S} = \bar{S}\right\}$$

$$= \mathbb{P}\left\{S \leq s\right\} \cdot \mathbb{P}\left\{\tilde{Y} = Y\right\} + \mathbb{P}\left\{\bar{S} \leq s\right\} \cdot \mathbb{P}\left\{\tilde{Y} = \bar{Y}\right\}$$

$$= (1-\epsilon)\mathbb{P}\left\{S \leq s\right\} + \epsilon\mathbb{P}\left\{\bar{S} \leq s\right\},$$

which follows that

$$\mathbb{P}\left\{S \leq s\right\} = \frac{1}{1-\epsilon}\mathbb{P}\left\{\tilde{S} \leq s\right\} - \frac{\epsilon}{1-\epsilon}\mathbb{P}\left\{\bar{S} \leq s\right\} = \frac{1}{1-\epsilon}\mathbb{P}\left\{\tilde{S} \leq s\right\} - \frac{\epsilon}{K(1-\epsilon)}\sum_{y=1}^{K}\mathbb{P}\left\{S_y \leq s\right\}.$$

**(3):**

$$\mathbb{P}\left\{\tilde{S} > s\right\} = \mathbb{P}\left\{\tilde{S} > s | \tilde{S} = S\right\} \cdot \mathbb{P}\left\{\tilde{S} = S\right\} + \mathbb{P}\left\{\tilde{S} > s | \tilde{S} = \bar{S}\right\} \cdot \mathbb{P}\left\{\tilde{S} = \bar{S}\right\}$$

$$= \mathbb{P}\left\{S > s\right\} \cdot \mathbb{P}\left\{\tilde{Y} = Y\right\} + \mathbb{P}\left\{\bar{S} > s\right\} \cdot \mathbb{P}\left\{\tilde{Y} = \bar{Y}\right\}$$

$$= (1-\epsilon)\mathbb{P}\left\{S > s\right\} + \epsilon\mathbb{P}\left\{\bar{S} > s\right\},$$

which follows that

$$\mathbb{P}\left\{S > s\right\} = \frac{1}{1-\epsilon}\mathbb{P}\left\{\tilde{S} > s\right\} - \frac{\epsilon}{1-\epsilon}\mathbb{P}\left\{\bar{S} > s\right\} = \frac{1}{1-\epsilon}\mathbb{P}\left\{\tilde{S} > s\right\} - \frac{\epsilon}{K(1-\epsilon)}\sum_{y=1}^{K}\mathbb{P}\left\{S_y > s\right\}.$$

$\square$

**Lemma G.2.** *The gradient of robust pinball loss can be bounded as follows*

$$\alpha - 1 - \frac{\epsilon}{1 - \epsilon} \leq \nabla_{\hat{\tau}} \tilde{l}_{1-\alpha}(\hat{\tau}, \tilde{S}, \{S_y\}_{y=1}^K) \leq \alpha + \frac{\epsilon}{1 - \epsilon}. \tag{13}$$

*Proof.* Consider the gradient of robust pinball loss:

$$\nabla_{\hat{\tau}} \tilde{l}_{1-\alpha}(\hat{\tau}, \tilde{S}, \{S_y\}_{y=1}^K) = \underbrace{\frac{\alpha}{1 - \epsilon} \mathbb{1}\left\{\hat{\tau} \geq \tilde{S}\right\}}_{(a)} - \underbrace{\sum_{y=1}^K \frac{\alpha\epsilon}{K(1 - \epsilon)} \mathbb{1}\left\{\hat{\tau} \geq S_y\right\}}_{(b)} - \underbrace{\frac{1 - \alpha}{1 - \epsilon} \mathbb{1}\left\{\hat{\tau} < \tilde{S}\right\}}_{(c)} + \underbrace{\sum_{y=1}^K \frac{(1 - \alpha)\epsilon}{K(1 - \epsilon)} \mathbb{1}\left\{\hat{\tau} < S_y\right\}}_{(d)}.$$

Due to fact that $\mathbb{1}\{\cdot\} \in [0, 1]$, we can bound each part as follows:

Part (a):

$$\frac{\alpha}{1 - \epsilon} \mathbb{1}\left\{\hat{\tau} \geq \tilde{S}\right\} \in \left[0, \frac{\alpha}{1 - \epsilon}\right].$$

Part (b):

$$\sum_{y=1}^K \frac{\alpha\epsilon}{K(1 - \epsilon)} \mathbb{1}\left\{\hat{\tau} \geq S_y\right\} \in \left[0, \frac{\alpha\epsilon}{1 - \epsilon}\right].$$

Part (c):

$$\frac{1 - \alpha}{1 - \epsilon} \mathbb{1}\left\{\hat{\tau} < \tilde{s}\right\} \in \left[0, \frac{1 - \alpha}{1 - \epsilon}\right].$$

Part (d):

$$\sum_{y=1}^K \frac{(1 - \alpha)\epsilon}{K(1 - \epsilon)} \mathbb{1}\left\{\hat{\tau} < S_y\right\} \in \left[0, \frac{(1 - \alpha)\epsilon}{1 - \epsilon}\right].$$

Combining four parts, we can conclude that

$$\nabla_{\hat{\tau}} \tilde{l}_{1-\alpha}(\hat{\tau}, \tilde{S}, \{S_y\}_{y=1}^K) \leq \frac{\alpha}{1 - \epsilon} - 0 - 0 + \frac{(1 - \alpha)\epsilon}{1 - \epsilon} = \alpha + \frac{\epsilon}{1 - \epsilon};$$

$$\nabla_{\hat{\tau}} \tilde{l}_{1-\alpha}(\hat{\tau}, \tilde{S}, \{S_y\}_{y=1}^K) \geq 0 - \frac{\alpha\epsilon}{1 - \epsilon} - \frac{1 - \alpha}{1 - \epsilon} + 0 = \alpha - 1 - \frac{\epsilon}{1 - \epsilon}.$$

□

**Lemma G.3.** *With at least probability $1 - \delta$, we have*

$$\left|\sum_{t=1}^T \mathbb{E}_{S_t}\left[\nabla_{\hat{\tau}_t} l_{1-\alpha}(\hat{\tau}_t, S_t)\right] - \sum_{t=1}^T \nabla_{\hat{\tau}_t} l_{1-\alpha}(\hat{\tau}_t, S_t)\right| \leq \sqrt{2T \log\left(\frac{4}{\delta}\right)}.$$

*Proof.* Define

$$Y_T = \sum_{t=1}^T \mathbb{E}_{S_t}\left[\nabla_{\hat{\tau}_t} l_{1-\alpha}(\hat{\tau}_t, S_t)\right] - \sum_{t=1}^T \nabla_{\hat{\tau}_t} l_{1-\alpha}(\hat{\tau}_t, S_t);$$

$$D_T = Y_T - Y_{T-1} = \mathbb{E}_S\left[\nabla_{\hat{\tau}_t} l_{1-\alpha}(\hat{\tau}_t, S_t)\right] - \nabla_{\hat{\tau}_t} l_{1-\alpha}(\hat{\tau}_t, S_t).$$

Now, we will verify that $\{Y_T\}$ is a martingale, and $\{D_T\}$ is a bounded martingale difference sequence. Due to the definition of $\{Y_T\}$, we have

$$\mathbb{E}_{S_t}[Y_T | Y_{T-1}, \cdots, Y_t] = \mathbb{E}_{S_t}[D_T + Y_{T-1} | Y_{T-1}, \cdots, Y_t] = \mathbb{E}_{S_t}[D_T | Y_{T-1}, \cdots, Y_t] + Y_{T-1} = Y_{T-1},$$

where the last equality follows from the definition of $\{D_T\}$. In addition, we have

$$\mathbb{E}_{S_t}\left[\nabla_{\hat{\tau}_t} l_{1-\alpha}(\hat{\tau}_t, S_t)\right] = \mathbb{P}\{S_t \leq \hat{\tau}_t\} - (1-\alpha) \in [\alpha-1, \alpha],$$

and Eq. (13) gives us that

$$D_T = \mathbb{E}_{S_t}\left[\nabla_{\hat{\tau}_t} l_{1-\alpha}(\hat{\tau}_t, S_t)\right] - \nabla_{\hat{\tau}_t} l_{1-\alpha}(\hat{\tau}_t, S_t) \in [-1, 1].$$

Therefore, by applying Azuma–Hoeffding's inequality, we can have

$$\mathbb{P}\left\{\left|\sum_{t=1}^T D_t\right| \leq t\right\} = \mathbb{P}\left\{\left|\sum_{t=1}^T \mathbb{E}_{S_t}\left[\nabla_{\hat{\tau}_t} l_{1-\alpha}(\hat{\tau}_t, S_t)\right] - \sum_{t=1}^T \nabla_{\hat{\tau}_t} l_{1-\alpha}(\hat{\tau}_t, S_t)\right| \leq r\right\} \geq 1 - 2\exp\left\{-\frac{r^2}{2T}\right\}.$$

Using $r = \sqrt{2T \log(4/\delta)}$, we have

$$\mathbb{P}\left\{\left|\sum_{t=1}^T \mathbb{E}_{S_t}\left[\nabla_{\hat{\tau}_t} l_{1-\alpha}(\hat{\tau}_t, S_t)\right] - \sum_{t=1}^T \nabla_{\hat{\tau}_t} l_{1-\alpha}(\hat{\tau}_t, S_t)\right| \leq \sqrt{2T \log\left(\frac{4}{\delta}\right)}\right\} \geq 1 - \frac{\delta}{2}.$$

$\square$

**Lemma G.4.** *Consider online conformal prediction under uniform label noise with noise rate $\epsilon \in (0, 1)$. Given Assumptions 2.1 and 2.2, when updating the threshold according to Eq. (6), for any $T \in \mathbb{N}^+$, we have*

$$-\max_{1 \leq t \leq T-1} \eta_t \cdot \left(\alpha + \frac{\epsilon}{1-\epsilon}\right) \leq \hat{\tau}_T \leq 1 + \max_{1 \leq t \leq T-1} \eta_t \cdot \left(\frac{1}{1-\epsilon} - \alpha\right) \tag{14}$$

*for all $T \in \mathbb{N}^+$.*

*Proof.* We prove this by induction. First, we know $\hat{\tau}_1 \in [0, 1]$ by assumption, which indicates that Eq. (14) is satisfied at $t = 1$. Then, we assume that Eq. (14) holds for $t = T$, and we will show that $\hat{\tau}_{T+1}$ lies in this range. Consider three cases:

**Case 1.** If $\hat{\tau}_T \in [0, 1]$, we have

$$\hat{\tau}_{T+1} = \hat{\tau}_T - \eta_T \cdot \nabla_{\hat{\tau}_T} \tilde{l}_{1-\alpha}(\hat{\tau}_T, \tilde{S}_T, \{S_{t,y}\}_{y=1}^K) \overset{(a)}{\in} \left[0 - \eta_T \cdot \left(\alpha + \frac{\epsilon}{1-\epsilon}\right), 1 - \eta_T \cdot \left(\alpha - 1 - \frac{\epsilon}{1-\epsilon}\right)\right]$$

$$\subseteq \left[-\max_{1 \leq t \leq T-1} \eta_t \cdot \left(\alpha + \frac{\epsilon}{1-\epsilon}\right), 1 + \max_{1 \leq t \leq T-1} \eta_t \cdot \left(\frac{1}{1-\epsilon} - \alpha\right)\right]$$

where (a) follows from Eq. (13).

**Case 2.** Consider the case where $\hat{\tau}_T \in [1, 1 + \max_{1 \leq t \leq T-1} \eta_t \cdot (1/(1-\epsilon) - \alpha)]$. The assumption that $\tilde{S}_T, S_{t,y} \in [0, 1]$ implies $\mathbb{1}\left\{\tilde{S}_T \leq \hat{\tau}_T\right\} = \mathbb{1}\{S_{t,y} \leq \hat{\tau}_T\} = 1$. Thus, we have

$$\nabla_{\hat{\tau}_T} \tilde{l}_{1-\alpha}(\hat{\tau}_T, \tilde{S}_T, \{S_{t,y}\}_{y=1}^K) = \frac{\alpha}{1-\epsilon} \mathbb{1}\left\{\hat{\tau}_T \geq \tilde{S}_T\right\} - \sum_{y=1}^K \frac{\alpha\epsilon}{K(1-\epsilon)} \mathbb{1}\{\hat{\tau}_T \geq S_{t,y}\} - \frac{1-\alpha}{1-\epsilon} \mathbb{1}\left\{\hat{\tau}_T \geq \tilde{S}_T\right\}$$

$$+ \sum_{y=1}^K \frac{(1-\alpha)\epsilon}{K(1-\epsilon)} \mathbb{1}\{\hat{\tau}_T \geq S_{t,y}\}$$

$$= \frac{\alpha}{1-\epsilon} - \sum_{y=1}^K \frac{\alpha\epsilon}{K(1-\epsilon)} = \alpha,$$

which follows that

$$\hat{\tau}_{T+1} = \hat{\tau}_T - \eta_T \cdot \nabla_{\hat{\tau}_T} \tilde{l}_{1-\alpha}(\hat{\tau}_T, \tilde{S}_T, \{S_{t,y}\}_{y=1}^K) = \hat{\tau}_T - \eta_T \alpha$$

$$\in \left[1 - \eta_T \alpha, 1 + \max_{1 \leq t \leq T-1} \eta_t \cdot \left(\frac{1}{1-\epsilon} - \alpha\right) - \frac{\eta_T}{1-\epsilon}\right]$$

$$\subseteq \left[-\max_{1 \leq t \leq T-1} \eta_t \cdot \left(\alpha + \frac{\epsilon}{1-\epsilon}\right), 1 + \max_{1 \leq t \leq T-1} \eta_t \cdot \left(\frac{1}{1-\epsilon} - \alpha\right)\right]$$

**Case 3.** Consider the case where $\hat{\tau}_T \in [-\max_{1 \leq t \leq T-1} \eta \cdot (\alpha + \epsilon/(1-\epsilon)), 0]$. The assumption that $\tilde{S}, S_y \in [0,1]$ implies $\mathbb{1}\left\{\tilde{S} \leq \hat{\tau}_T\right\} = \mathbb{1}\left\{S_y \leq \hat{\tau}_T\right\} = 0$. Thus, we have

$$\nabla_{\hat{\tau}_T} \tilde{l}_{1-\alpha}(\hat{\tau}_T, \tilde{S}_T, \{S_{t,y}\}_{y=1}^K) = \frac{\alpha}{1-\epsilon} \mathbb{1}\left\{\hat{\tau}_T \geq \tilde{S}_T\right\} - \sum_{y=1}^K \frac{\alpha\epsilon}{K(1-\epsilon)} \mathbb{1}\left\{\hat{\tau}_T \geq S_{t,y}\right\} - \frac{1-\alpha}{1-\epsilon} \mathbb{1}\left\{\hat{\tau}_T \geq \tilde{S}_T\right\}$$

$$+ \sum_{y=1}^K \frac{(1-\alpha)\epsilon}{K(1-\epsilon)} \mathbb{1}\left\{\hat{\tau}_T \geq S_{t,y}\right\}$$

$$= -\frac{1-\alpha}{1-\epsilon} \mathbb{1}\left\{\tau < \tilde{s}\right\} + \sum_{y=1}^K \frac{(1-\alpha)\epsilon}{K(1-\epsilon)} = \alpha - 1,$$

which follows that

$$\hat{\tau}_{T+1} = \hat{\tau}_T - \eta_T \cdot \nabla_{\hat{\tau}_T} \tilde{l}_{1-\alpha}(\hat{\tau}_T, \tilde{S}_T, \{S_{t,y}\}_{y=1}^K) = \hat{\tau}_T + \eta_T \cdot (1-\alpha)$$

$$\in \left[-\max_{1 \leq t \leq T-1} \eta_t \cdot \left(\alpha + \frac{\epsilon}{1-\epsilon}\right) + \eta_T \cdot (1-\alpha), 0 + \eta_T \cdot (1-\alpha)\right]$$

$$\subseteq \left[-\max_{1 \leq t \leq T-1} \eta_t \cdot \left(\alpha + \frac{\epsilon}{1-\epsilon}\right), 1 + \max_{1 \leq t \leq T-1} \eta_t \cdot \left(\frac{1}{1-\epsilon} - \alpha\right)\right]$$

Combining three cases, we can conclude that

$$-\max_{1 \leq t \leq T-1} \eta_t \cdot \left(\alpha + \frac{\epsilon}{1-\epsilon}\right) \leq \hat{\tau}_T \leq 1 + \max_{1 \leq t \leq T-1} \eta_t \cdot \left(\frac{1}{1-\epsilon} - \alpha\right)$$

$\square$

**Lemma G.5.** *With at least probability $1 - \delta$, we have*

$$\left|\sum_{t=1}^T \mathbb{E}_{\tilde{S}_t, S_{t,y}} \left[\nabla_{\hat{\tau}_t} \tilde{l}_{1-\alpha}(\hat{\tau}_t, \tilde{S}_t, \{S_{t,y}\}_{y=1}^K)\right] - \sum_{t=1}^T \nabla_{\hat{\tau}_t} \tilde{l}_{1-\alpha}(\hat{\tau}_t, \tilde{S}_t, \{S_{t,y}\}_{y=1}^K)\right| \leq \frac{\sqrt{2T \log(2/\delta)}}{1 - \epsilon}.$$

*Proof.* Define

$$Y_T = \sum_{t=1}^T \mathbb{E}_{\tilde{S}_t, S_{t,y}} \left[\nabla_{\hat{\tau}_t} \tilde{l}_{1-\alpha}(\hat{\tau}_t, \tilde{S}_t, \{S_{t,y}\}_{y=1}^K)\right] - \sum_{t=1}^T \nabla_{\hat{\tau}_t} \tilde{l}_{1-\alpha}(\hat{\tau}_t, \tilde{S}_t, \{S_{t,y}\}_{y=1}^K);$$

$$D_T = Y_T - Y_{T-1} = \mathbb{E}_{\tilde{S}_t, S_{t,y}} \left[\nabla_{\hat{\tau}_t} \tilde{l}_{1-\alpha}(\hat{\tau}_t, \tilde{S}_t, \{S_{t,y}\}_{y=1}^K)\right] - \nabla_{\hat{\tau}_t} \tilde{l}_{1-\alpha}(\hat{\tau}_t, \tilde{S}_t, \{S_{t,y}\}_{y=1}^K).$$

Now, we will verify that $\{Y_T\}$ is a martingale, and $\{D_T\}$ is a bounded martingale difference sequence. Due to the definition of $\{Y_T\}$, we have

$$\mathbb{E}_{\tilde{S}_t, S_{t,y}}[Y_T|Y_{T-1}, \cdots, Y_t] = \mathbb{E}_{\tilde{S}_t, S_{t,y}}[D_T + Y_{T-1}|Y_{T-1}, \cdots, Y_t] = \mathbb{E}_{\tilde{S}_t, S_{t,y}}[D_T|Y_{T-1}, \cdots, Y_t] + Y_{T-1} = Y_{T-1},$$

where the last equality follows from the definition of $\{D_T\}$. In addition, we have

$$\mathbb{E}_{\tilde{S}_t, S_{t,y}} \left[\nabla_{\hat{\tau}_t} \tilde{l}_{1-\alpha}(\hat{\tau}_t, \tilde{S}_t, \{S_{t,y}\}_{y=1}^K)\right] \in [\alpha - 1, \alpha],$$

and Lemma G.2 gives us that

$$D_T = \mathbb{E}_{\tilde{S}_t, S_{t,y}}[\nabla_{\hat{\tau}_t} \tilde{l}_{1-\alpha}(\hat{\tau}_t, \tilde{S}_t, \{S_{t,y}\}_{y=1}^K)] - \nabla_{\hat{\tau}_t} \tilde{l}_{1-\alpha}(\hat{\tau}_t, \tilde{S}_t, \{S_{t,y}\}_{y=1}^K) \in \left[-\frac{1}{1-\epsilon}, \frac{1}{1-\epsilon}\right].$$

Therefore, by applying Azuma–Hoeffding's inequality, we can have

$$\mathbb{P}\left\{\left|\sum_{t=1}^{T} D_t\right| \leq t\right\} = \mathbb{P}\left\{\left|\sum_{t=1}^{T}\mathbb{E}_{\tilde{S}_t, S_{t,y}}[\nabla_{\hat{\tau}_t}\tilde{l}_{1-\alpha}(\hat{\tau}_t, \tilde{S}_t, \{S_{t,y}\}_{y=1}^{K})] - \sum_{t=1}^{T}\nabla_{\hat{\tau}_t}\tilde{l}_{1-\alpha}(\hat{\tau}_t, \tilde{S}_t, \{S_{t,y}\}_{y=1}^{K})\right| \leq r\right\}$$

$$\geq 1 - 2\exp\left\{-\frac{[r(1-\epsilon)]^2}{2T}\right\}.$$

Using $r = [2T\log(2/\delta)]^{-1/2}/(1-\epsilon)$, we have

$$\mathbb{P}\left\{\left|\sum_{t=1}^{T}\mathbb{E}_{\tilde{S}_t, S_{t,y}}[\nabla_{\hat{\tau}_t}\tilde{l}_{1-\alpha}(\hat{\tau}_t, \tilde{S}_t, \{S_{t,y}\}_{y=1}^{K})] - \sum_{t=1}^{T}\nabla_{\hat{\tau}_t}\tilde{l}_{1-\alpha}(\hat{\tau}_t, \tilde{S}_t, \{S_{t,y}\}_{y=1}^{K})\right| \leq \frac{\sqrt{2T\log(2/\delta)}}{1-\epsilon}\right\} \geq 1 - \delta.$$

$\square$

