# OpenReview forum: "Robust Online Conformal Prediction under Uniform Label Noise"
_ICML.cc/2025/Conference — Submitted to ICML 2025_

### Official Review · Reviewer_3DHQ · 2025-03-13

**Overall Recommendation:** 3

**Summary:**

The authors consider online conformal prediction with uniform label noise, where the goal is to solve a sequential classification problem by providing prediction sets at each round, such that over time, the true label will lie in the prediction set with probability approximately $1-\alpha$ where $\alpha$ is a predetermined coverage parameter. The authors establish, both empirically and theoretically, that existing methods suffer from coverage gaps in the presence of label noise, with the most common phenomenon being over-coverage in which the prediction sets contain the true label with probability larger than $1-\alpha$. To address this, the authors design a robust algorithm using a variant of the pinball loss and show that their algorithm converges to have no coverage gap even in the presence of label noise.

## update after rebuttal:
My assessment remains after reading the other reviews and comments by the authors.

**Claims And Evidence:**

The theoretical claims made in this submission are supported by formal proofs, and the empirical evaluations are clear and convincing. I do not see any problematic claims.

**Essential References Not Discussed:**

I did not find any essential references not discussed.

**Experimental Designs Or Analyses:**

I did not check the soundness of experimental designs or analyses.

**Methods And Evaluation Criteria:**

It is a bit unclear to me why we care about exact coverage, that is, why over-coverage is an issue. I am far from an expert in this area of research, but my intuition tells me that if we cover the true label with probability larger than $1-\alpha$, then our algorithm is doing better, that is, its predictions are more accurate. Examining some of the previous works, it appears to me as coverage of at least $1-\alpha$ is sought after rather than exactly $1-\alpha$.

**Other Comments Or Suggestions:**

N/A

**Other Strengths And Weaknesses:**

Strengths:

* The paper is well-structured, easy to follow and to understand. The authors provide clear explanations for their methods, techniques and experiments.

* The theoretical results presented by the authors seem non-trivial, and they are supported by empirical evidence.

Weaknesses:

* I am unsure about the main motivation of this work - that is, why the fact that existing methods exhibit over-coverage in the presence of label noise is an issue. I could not find any reference to that in previous works, but my familiarity with this line of work is very limited.

**Questions For Authors:**

As I previously mentioned in the review, I don't think I fully understand why the phenomenon of over-coverage in the presence of label noise should be worrisome. On the face of it, it seems to me like higher coverage than the prespecified $(1-\alpha)$ only indicates more accurate predictions. Looking into a few of the previous works, I could not find a similar objective to that of precise coverage. As I am very unfamiliar with this area of research, I would appreciate it if the authors could provide an explanation and/or a more explicit motivation, and if I am convinced then I will increase my score.

**Relation To Broader Scientific Literature:**

It seems to me that for the most part the authors adequately relate the contributions of the paper to the broader literature on conformal prediction, however I am a bit unsure about the precise coverage objective, that is, whether or not this objective has been used in previous works and whether or not over-coverage has been recognized as an issue in previous works as well.

**Theoretical Claims:**

I did not check the correctness of the proofs of the theoretical claims.

---

> ### Author Rebuttal · Authors · 2025-03-31
>
> > 1. Justification on achieving precise $1-\alpha$ coverage
>
> Thank you for the insightful comment. Achieving precise $1-\alpha$ coverage is one of the common desiderata in conformal prediction [1,2,3,4], since over-coverage results in excessively large prediction sets, reducing their practical utility. In conformal prediction, smaller prediction sets are generally preferred to provide more informative outputs, aligning with the desideratum of size efficiency [5,6]. For instance, in medical diagnosis with $\alpha=0.1$, prediction sets exceeding the intended 90% coverage (e.g., reaching 95%) may include extraneous and irrelevant options, such as unlikely diseases, thereby reducing specificity. Conversely, users can select a smaller $\alpha$ to ensure prediction sets achieve the desired higher coverage. A more detailed explanation can be found in Chapter 3.6 of [1].
>
> Notably, over-coverage has been identified as a central challenge in conformal prediction under conditions of label noise [7]. In the literature, prior works [3,4] have been devoted to addressing this issue, as detailed in the Related Work section (Appendix A). In this work, we present the first attempt to address this issue within the framework of online conformal prediction. We believe this should clarify our focus, and we welcome any further discussion.
>
>
> ### References
> [1] Angelopoulos A N, et al. Theoretical foundations of conformal prediction. arXiv preprint 2024.
>
> [2] Angelopoulos A N, et al. A gentle introduction to conformal prediction and distribution-free uncertainty quantification. arXiv preprint 2021.
>
> [3] Sesia M, et al. Adaptive conformal classification with noisy labels. JRSSB 2024.
>
> [4] Penso C, et al. Estimating the Conformal Prediction Threshold from Noisy Labels. arXiv preprint 2025.
>
> [5] Angelopoulos A, et al. Uncertainty sets for image classifiers using conformal prediction. ICLR 2021.
>
> [6] Huang J, et al. Conformal prediction for deep classifier via label ranking. ICML 2024.
>
> [7] Einbinder B S, et al. Label noise robustness of conformal prediction. JMLR 2024.

---

> > ### Comment · Reviewer_3DHQ · 2025-04-02
> >
> > I thank the authors for their response.
> >
> > Having seen the evidence provided by the authors for the interest in exact coverage in the relevant literature, I will adjust my overall score accordingly.

---

> > > ### Author Response · Authors · 2025-04-02
> > >
> > > Thank you for raising the score. We sincerely appreciate your time and effort in reviewing our work.

---

### Official Review · Reviewer_VFDZ · 2025-03-14

**Overall Recommendation:** 3

**Summary:**

The paper "Robust Online Conformal Prediction under Uniform Label Noise" addresses the challenge of online conformal prediction (OCP) in the presence of uniform label noise. Conformal prediction is a widely used technique for uncertainty quantification, guaranteeing a predefined coverage probability for prediction sets. While recent advances have extended conformal prediction to online settings, existing methods typically assume perfectly accurate labels, which is often unrealistic in practical applications where label noise is prevalent.

To address this issue, the authors propose Noise-Robust Online Conformal Prediction (NR-OCP), which adapts the conformal threshold update process using a novel robust pinball loss function. This function provides an unbiased estimate of the clean pinball loss without requiring access to true labels, thereby mitigating the impact of label noise on the coverage guarantees. Theoretical analysis demonstrates that NR-OCP eliminates the coverage gap introduced by label noise and achieves a convergence rate of O(T⁻¹/²) for both empirical and expected coverage errors. The paper further validates its approach through experiments on CIFAR-100 and ImageNet, showing that NR-OCP achieves precise coverage while maintaining small prediction sets, outperforming standard online conformal prediction methods.

**Claims And Evidence:**

The paper provides a solid theoretical foundation to support its claims regarding the effect of label noise on coverage guarantees in OCP. The mathematical derivations are well-structured, and the proposed robust pinball loss is rigorously justified through expectation-based approximations.

The empirical evidence is also strong—experiments on CIFAR-100 and ImageNet with varying noise levels demonstrate the effectiveness of NR-OCP in reducing the coverage gap while maintaining compact prediction sets. However, one potential limitation is the reliance on synthetically generated uniform label noise, which may not fully capture real-world noise distributions that are often structured (e.g., class-dependent or instance-dependent noise). While the results convincingly show improvements over standard OCP, further validation on real-world noisy datasets could strengthen the paper’s contributions.

**Essential References Not Discussed:**

While the paper’s focus is on uniform noise, discussing extensions to other noise models would make the work more applicable to real-world scenarios.

**Experimental Designs Or Analyses:**

The results are statistically significant, as NR-OCP consistently achieves near-zero coverage gaps while maintaining smaller prediction sets compared to standard OCP. One minor concern is the lack of statistical significance testing in the reported results. Given the small coverage gaps, a confidence interval or hypothesis test would help ensure that the improvements are not due to random fluctuations.

**Methods And Evaluation Criteria:**

The methodology is well-aligned with the problem. The introduction of a robust pinball loss function directly addresses the issue of label noise by adjusting the threshold update mechanism in OCP. The choice of evaluation metrics—coverage gap and prediction set size—is appropriate, as these measure both the reliability and efficiency of the proposed approach.

That said, the experiments focus primarily on computer vision datasets (CIFAR-100, ImageNet). While these are standard benchmarks, additional experiments on other domains, such as natural language processing (e.g., text classification datasets with noisy labels) or tabular data, would help assess the generalizability of NR-OCP.

**Other Comments Or Suggestions:**

Equation (4): It would be helpful to clarify the role of learning rate decay in dynamic learning rates.
Notation: In some places, the notation for learning rates (ηt) and coverage errors could be better aligned for readability.
Figure 2 caption: The term "Baseline" should explicitly reference the specific online conformal prediction method used for comparison.

**Other Strengths And Weaknesses:**

Strengths:
Addresses a critical gap in conformal prediction by considering label noise.
Well-supported theoretical contributions—the results are rigorous and remove limiting assumptions in prior work.
Effective empirical validation—results demonstrate both strong coverage guarantees and efficiency improvements.
Clear and well-organized writing—the paper presents a complex topic in an accessible manner.
Weaknesses:
Relies on uniform label noise—real-world settings often involve more complex noise structures.
Lack of ablation studies—it would be useful to analyze how different components (e.g., robust pinball loss) contribute to performance gains.
Limited generalization across domains—the experiments focus on image classification, but additional evaluation on NLP or tabular data would strengthen the work.

**Questions For Authors:**

How does the variance of the robust pinball loss gradient updates compare to the standard pinball loss?

A high variance could impact stability—has this been analyzed empirically?
Would NR-OCP be effective under instance-dependent noise models?

Many real-world datasets exhibit structured noise—how might the method adapt to these scenarios?
Could NR-OCP be extended to adversarial label noise?

Have the authors considered robustness against worst-case noise perturbations rather than uniform noise?

**Relation To Broader Scientific Literature:**

The paper is well-situated in the broader literature on conformal prediction, online learning, and label noise robustness. It builds upon key prior works such as:

Online Conformal Prediction: (Gibbs & Candes, 2021; Angelopoulos et al., 2024)
Label Noise Robustness: (Einbinder et al., 2024; Penso & Goldberger, 2024)
Pinball Loss in Conformal Prediction: (Steinwart & Christmann, 2011)
A notable strength of the paper is that it removes a strong distributional assumption made by Einbinder et al. (2024), making the analysis more general. This is an important step forward in developing noise-robust conformal prediction methods.

**Theoretical Claims:**

The theoretical claims appear well-founded and consistent with prior work in online conformal prediction and robust learning. The proofs leverage concentration inequalities (e.g., Azuma–Hoeffding) and martingale-based arguments, which are standard tools for analyzing online learning algorithms.

One aspect that could benefit from additional clarification is the bias-variance tradeoff in gradient estimation. While the paper establishes that the robust pinball loss leads to an unbiased estimate in expectation, it does not discuss the variance of the estimate explicitly. A high variance in the gradient updates could impact the stability of NR-OCP, especially for small sample sizes.

---

> ### Author Rebuttal · Authors · 2025-03-31
>
> While AC suggests that the review is flagged as generated, we still provide detailed responses below:
>
> > 1. No ablation study on the proposed loss
>
> See response #2 to reviewer rF6c.
>
>
> > 2. Restriction to uniform label noise
>
> See response #3 to reviewer rF6c.
>
>
> > 3. Variance analysis for the gradient
>
> Here we provide a variance analysis for the gradient of robust pinball loss. Recall that the robust pinball loss is defined as
> $$
> \\nabla_ {\\hat{\\tau}_ {t}}\\tilde{l}_ {1-\\alpha} (\\hat{\\tau}_ {t},\\tilde{S}_ t,\\{S_ {t,y}\\}_ {y=1}^K)
> =\\nabla_ {\\hat{\\tau}_ {t}}l_ 1(\\hat{\\tau}_ {t},\\tilde{S}_ t)
> +\\nabla_ {\\hat{\\tau}_ {t}}l_ 2(\\hat{\\tau}_ {t},\\{S_ {t,y}\\}_ {y=1}^K)
> $$
> where
> \\begin{align}
> & \\nabla_ {\\hat{\\tau}_ {t}}l_ 1(\\hat{\\tau}_ {t},\\tilde{S}_ t)
> =\\frac{1}{1-\\epsilon}[1\\{\\tilde{S}_ t\\leq\\hat{\\tau}_ {t}\\}
> -(1-\\alpha)] \\
> & \nabla_ {\hat{\tau}_ {t}}l_ 2(\hat{\tau}_ {t},S_ {t,y})
> =\frac{\\epsilon}{K(1-\\epsilon)}\\sum_ {y=1}^K[1\\{S_ {t,y}\\leq\\hat{\\tau}_ {t}\\}-(1-\\alpha)]
> \\end{align}
> To proceed, we make the following simplifying assumptions:
>
> 1. The scores $S_{t,y}$ for different labels $y$ are identically distributed for a given instance $t$, with $p = \mathbb{P}(S_{t,y} < \tau)$. This assumes the score distribution is similar across labels, though in practice, it depends on the instance and model.
> 2. The scores $S_{t,y}$ are independent across different $y$. This is a simplification, as scores for the same instance may be correlated, but it provides a tractable starting point.
>
> Then, the gradient variance is given by
> $$
> Var(\nabla_ {\hat{\tau}_ {t}}\tilde{l}_ {1-\alpha})
> =Var(\nabla_ {\hat{\tau}_ {t}}l_ 1)
> +Var(\nabla_ {\hat{\tau}_ {t}}l_ 2)
> -2Cov(\nabla_ {\hat{\tau}_ {t}}l_ 1,\nabla_ {\hat{\tau}_ {t}}l_ 2)
> $$
> **Part 1.** $Var(\nabla_ {\hat{\tau}_ {t}}l_ 1)$:
> $$
> Var(\nabla_ {\hat{\tau}_ {t}}l_ 1)
> =Var(\frac{l_ {1-\alpha}(\hat{\tau}_ {t},\tilde{S}_ t)}{1-\epsilon})
> =\frac{p(1-p)}{(1-\epsilon)^2}
> $$
> **Part 2.** $Var(\nabla_ {\hat{\tau}_ {t}}l_ 2)$:
> $$
> Var(\nabla_ {\hat{\tau}_ {t}}l_ 2)
> =Var(\frac{\epsilon}{K(1-\epsilon)}\sum_ {y=1}^K[1\{S_ {t,y}\leq\hat{\tau}_ {t}\}-(1-\alpha)])
> =\frac{\epsilon^2p(1-p)}{K(1-\epsilon)^2}
> $$
> **Part 3.** $Cov(\nabla_ {\hat{\tau}_ {t}}l_ 1,\nabla_ {\hat{\tau}_ {t}}l_ 2)$:
> \begin{align}
> Cov(\nabla_ {\hat{\tau}_ {t}}l_ 1,\nabla_ {\hat{\tau}_ {t}}l_ 2)
> &=\mathbb{E}[\nabla_ {\hat{\tau}_ {t}}l_ 1\cdot\nabla_ {\hat{\tau}_ {t}}l_ 2]
> -\mathbb{E}[\nabla_ {\hat{\tau}_ {t}}l_ 1]\mathbb{E}[\nabla_ {\hat{\tau}_ {t}}l_ 2]
> \end{align}
> Since we assume $S_{t,y}$ are independent across different $y$, $Cov(\nabla_ {\hat{\tau}_ {t}}l_ 1,\nabla_ {\hat{\tau}_ {t}}l_ 2)$ can be reduced to
> \begin{align}
> Cov(\nabla_ {\hat{\tau}_ {t}}l_ 1,\nabla_ {\hat{\tau}_ {t}}l_ 2)
> &=\mathbb{E}[\frac{l_ {1-\alpha}(\hat{\tau}_ {t},\tilde{S}_ t)}{1-\epsilon}\cdot\frac{\epsilon l_ {1-\alpha}(\hat{\tau}_ {t},\tilde{S}_ t)}{K(1-\epsilon)}]
> -\mathbb{E}[\frac{l_ {1-\alpha}(\hat{\tau}_ {t},\tilde{S}_ t)}{1-\epsilon}]\mathbb{E}[\frac{\epsilon l_ {1-\alpha}(\hat{\tau}_ {t},\tilde{S}_ t)}{K(1-\epsilon)}] \\
> &=\frac{\epsilon p(1-p)}{K(1-\epsilon)^2}
> \end{align}
> In summary, the gradient variance is given by
> $$
> Var(\nabla_ {\hat{\tau}_ {t}}\tilde{l}_ {1-\alpha})
> =\frac{p(1-p)}{(1-\epsilon)^2}+\frac{\epsilon^2p(1-p)}{K(1-\epsilon)^2}-\frac{2\epsilon p(1-p)}{K(1-\epsilon)^2}=\frac{p(1-p)}{K}+\frac{p(1-p)(K-1)}{K(1-\epsilon)^2}
> $$
> Therefore, we can conclude that a larger noise rate $\epsilon$ will increase the gradient variance of robust pinball loss, reducing the stability of results.
>
>
> > 3. Other comments on presentation and notation
>
> Thank you for the suggestion. We will improve the clarity accordingly in the final version.

---

### Official Review · Reviewer_E8K1 · 2025-03-14

**Overall Recommendation:** 3

**Summary:**

This paper aims to develop an online conformal prediction method that can handle the case where the labels of data are noisy, which ensure the robustness of online conformal prediction. The novelty of method is mainly focus on adjusting the previous pinball loss to a robust pinball loss, which is a weighted sum of pinball loss with noisy scores and pinball loss with scores of all classes. This paper also theoretically proves the consistency between the loss under noisy data and clean data, and showing that the method can boost on the elimination of coverage gap caused by the noisy data. Sufficient experiements are also conducted to show the effectiveness of methods.

**Claims And Evidence:**

The claims are clear and the evidence is sufficient.

**Essential References Not Discussed:**

All the essential references are discussed.

**Experimental Designs Or Analyses:**

I am actually quite confused about the baselines part in the experimental details. Actually, the novelty of NR-OCP is the loss function that can be applied to the case of noisy data, so actually the loss function can be used on any of the online conformal prediction methods, but why only choose one standard online conformal prediction method (with pinball loss) with 4 different non-conformalty score algorithms? If I have any misunderstanding, please correct me.

**Methods And Evaluation Criteria:**

The method of the proposed loss and its efficiency are clearly stated and the banchmark datasets are promising.

**Other Comments Or Suggestions:**

1. In the whole paper, the notation $T$ is used before declaration. Given the definition of precise coverage guarantee, I think this should be the total number of sequences.
2. In the formula of empirical error and expected error, the input becomes a text version of $\text{T}$, which is a notaion inconsistency.

**Other Strengths And Weaknesses:**

I think there is no need to put that much theoretical analysis, for example, the Proposition 4.1, I know it represents the upper bound of coverage gap under the dynamic learning rate, but when considering the $T$ goes to inifinity, the conclusion from the this proposition has no differences with the Proposition 3.1, even though it is a more general case. I recommand to add some experiements to compare with other state-of-the-art online conformal prediction algorithms.

**Questions For Authors:**

1. In the experiements from Appendix, I found that sometimes when the noisy rate decreases, e.g., from 0.15 to 0.1, the coverage gap will increase, actually this is inconsistent with the theoretical analysis, can you explain why that happens?

**Relation To Broader Scientific Literature:**

This paper has a great motivation and contribution on online conformal prediction, which can solve the problem that the coverage gap will be larger, if noisy data is included.

**Theoretical Claims:**

I check the correctness of the proof for theorectical claims, it is quite solid.

---

> ### Author Rebuttal · Authors · 2025-03-31
>
> > 1. Lack of comparative experiments with other online conformal prediction algorithms
>
> Thank you for your insightful suggestion. We conduct new experiments by integrating our robust pinball loss into a new baseline - SAOCP[1]. In particular, we employ LAC score to generate prediction sets with error rate $\alpha=0.1$, using ResNet50 on CIFAR-100. In Figures 2 and 3 of [[link](https://anonymous.4open.science/r/Noise_Robust_Online_Conformal_Prediction-DD85/Supplementary_Materials_for__Robust_Online_Conformal_Prediction_under_Uniform_Label_Noise.pdf)], we present the new results of SAOCP and NR-SAOCP (Noise-robust SAOCP strengthened by our method).
>
> 1. Results in Figure 2a show that SAOCP cannot achieve the desired coverage in the presence of label noise, with higher noise rates resulting in a larger gap.
> 2. Results in Figures 2b, 3a, and 3b show that NR-SAOCP eliminates the long-run coverage gap in various noise rates.
>
> The new results highlight the effectiveness of our method against the updated SAOCP baseline. We will incorporate these findings into the final version and would greatly appreciate the reviewer’s suggestions for any additional baselines we may have overlooked.
>
> > 2. The theoretical analysis is excessive
>
> We sincerely thank you for the insightful comment. We'd like to clarify that the conclusion from Prop. 4.1 is equivalent to Prop. 3.1 only when $\sum_{t=1}^T|\eta_t^{-1}-\eta_{t-1}^{-1}|/T\to0$, considering the T goes to inifinity. Thus, we present Prop. 4.1 to extend beyond Prop. 3.1 by analyzing the coverage gap under a dynamic learning rate, a **widely adopted** technique in online learning [2,3,4,5]. In Prop. 4.3 and 4.4, we provide theoretical evidence for the effectiveness of our method under this general setting. We believe an extensive analysis can enrich the theoretical framework for understanding label noise in online conformal prediction.
>
>
> > 3. Inconsistent results in the experiments from the appendix
>
> Thank you for the careful review. We guess the "inconsistent" increase is found in the results of our method (e.g., ImageNet in Table 6). We'd like to clarify that our theoretical results in Prop. 3.1 and 4.1 demonstrate the coverage performance of the standard online CP method (Baseline) under label noise, where a systematic coverage gap is introduced by the label noise. In contrast, our robust pinball loss can reduce the gap to a very small level, rendering the outcomes of relative comparisons susceptible to stochastic noise. For example, in Table 6, the CovGap of baseline is larger with a higher noise rate, while our method achieves negligible CovGaps.
>
>
> > 4. Other comments on notation
>
> Thank you for pointing out the issues. We will fix these notations accordingly in the final version.
>
> ### Reference
> [1] Bhatnagar A, et al. Improved online conformal prediction via strongly adaptive online learning. ICML 2023.
>
> [2] Angelopoulos A N, et al. Online conformal prediction with decaying step sizes. ICML 2024.
>
> [3] Kim D, et al. Robust Bayesian Optimization via Localized Online Conformal Prediction. arXiv preprint 2024.
>
> [4] Hazan E. Introduction to online convex optimization. Foundations and Trends in Optimization 2016.
>
> [5] Orabona F. A modern introduction to online learning. arXiv preprint 2019.

---

> > ### Comment · Reviewer_E8K1 · 2025-04-05
> >
> > Thanks for your added experiments on one more baseline. I can also understand that Prop 4.1 is used for proving the following theoretical results in Section 4.2, but it seems not to provide extra help/information in understanding the Prop 4.3 and Prop 4.4, since in the interpretation, it also includes somewhat repeated contents, e.g. (line 316- 320 on the left). Also, for the experimental results in the Appendix, I can understand that the proposed method is much better than the baseline, but what I am confused is why the CovGap could increase more than twice than itself when the noisy rate decreases. For example, in Table 3, using the Dynamic LR schedule, the CovGaps of proposed method when $\alpha=0.05, \epsilon=0.1$ for both dataset are much larger than that of the proposed method when $\alpha=0.05, \epsilon=0.15$. The situation that CovGaps of proposed method in smaller error rate are larger is frequently happening among the other tabels, but hardly happened for the baseline method, can you explain that? Thanks.

---

> > > ### Author Response · Authors · 2025-04-05
> > >
> > > We sincerely thank the reviewer for the constructive feedback. We reply to the remaining concerns point by point.
> > >
> > > 1. **Writing of theoretical results**.
> > >
> > > Thank you for the detailed explanation and valuable suggestion. We agree that the interpretation of the coverage gap can be simplified due to its limited informativeness. We present Proposition 4.1 to validate the results of Proposition 3.1 under the learning rate condition: $\sum_{t=1}^T|\eta_t^{-1}-\eta_{t-1}^{-1}|/T\to0$. Following your advice, we will simplify the writing of Subsection 4.1 and relocate some theoretical explanations to the Appendix in the final version. This adjustment will allow us to include additional experimental results (e.g., results of SAOCP) in the revised manuscript.
> > >
> > > 2. **Results in Appendix**.
> > >
> > > Thank you for the thoughtful comment. As established in Propositions 3.4 and 4.4, our method’s empirical error converges to zero in probability at a rate of $\mathcal{O}(T^{-1/2})$, indicating that CovGap is primarily influenced by the number of iterations $T$ and random noise, rather than the noise rate $\epsilon$. Consequently, variations in CovGap across different noise rates are likely attributable to **random fluctuations** rather than a systematic dependency on $\epsilon$. In contrast, the empirical error (and CovGap) of the baseline method depends on the noise rate $\epsilon$, increasing as $\epsilon$ grows.
> > >
> > > To validate this, we reproduce the experiments on the specific example you highlighted — $\alpha=0.05, \epsilon\in\{0.1, 0.15\}$ with the Dynamic LR schedule. We repeat the experiment 30 times for each setting and perform a two-sample t-test to compare the CovGap values between the two noise rates. The null hypothesis $(H_0)$ is that the CovGap means between the two groups are equal. The corresponding results are presented below. The t-test yields a t-statistic of -0.1628 and a p-value of 0.8713, suggesting that we cannot reject $(H_0)$. This indicates that the observed differences in CovGap are **not statistically significant**, supporting the explanation of random variability rather than a direct effect of the noise rate. We hope this resolves your concerns. Thank you again for helping us improve the manuscript.
> > >
> > > | Id | ε=0.1  | ε=0.15 | Id | ε=0.1  | ε=0.15 | Id | ε=0.1  | ε=0.15 |
> > > |------|---------|---------|------|---------|---------|------|---------|---------|
> > > | 1    | 0.161%  | 0.040%  | 11   | 0.581%  | 0.450%  | 21   | 0.124%  | 0.120%  |
> > > | 2    | 0.240%  | 0.210%  | 12   | 0.280%  | 0.452%  | 22   | 0.040%  | 0.022%  |
> > > | 3    | 0.040%  | 0.072%  | 13   | 0.320%  | 0.030%  | 23   | 0.210%  | 0.440%  |
> > > | 4    | 0.300%  | 0.110%  | 14   | 0.300%  | 0.290%  | 24   | 0.103%  | 0.551%  |
> > > | 5    | 0.215%  | 0.300%  | 15   | 0.331%  | 0.440%  | 25   | 0.034%  | 0.151%  |
> > > | 6    | 0.250%  | 0.391%  | 16   | 0.090%  | 0.073%  | 26   | 0.070%  | 4.163e-15% |
> > > | 7    | 0.166%  | 0.520%  | 17   | 0.050%  | 4.163e-15% | 27   | 0.260%  | 0.090%  |
> > > | 8    | 0.160%  | 0.033%  | 18   | 0.341%  | 0.200%  | 28   | 0.674%  | 0.360%  |
> > > | 9    | 0.190%  | 0.160%  | 19   | 0.340%  | 0.030%  | 29   | 0.160%  | 0.430%  |
> > > | 10   | 0.400%  | 0.430%  | 20   | 0.210%  | 0.260%  | 30   | 0.010%  | 0.190%  |

---

### Official Review · Reviewer_rF6c · 2025-03-22

**Overall Recommendation:** 3

**Summary:**

This paper studies the robustness of online conformal prediction (OCP) under uniform label noise with a known noise rate. The authors demonstrate that label noise introduces a persistent gap between the actual and desired coverage rates, affecting the reliability of prediction sets. To address this, the paper proposes a novel method called Noise-Robust Online Conformal Prediction (NR-OCP), which updates the prediction threshold using a robust pinball loss designed to provide an unbiased estimate of the clean loss without requiring ground-truth labels. The authors provide theoretical guarantees showing that NR-OCP eliminates the coverage gap and achieves convergence rates of \(O(T^{-1/2})\) for both empirical and expected coverage errors under both constant and dynamic learning rate schedules. Extensive experiments on CIFAR-100 and ImageNet with various models and non-conformity scores confirm the effectiveness of NR-OCP in achieving accurate coverage and smaller prediction sets compared to standard OCP methods.

**Claims And Evidence:**

The paper makes several core claims: (1) that uniform label noise causes a systematic coverage gap in online conformal prediction; (2) that the proposed NR-OCP method, which utilizes a robust pinball loss, can eliminate this gap without access to clean labels; and (3) that NR-OCP achieves
O(T−1/2)O(T−1/2) convergence rates for both empirical and expected coverage errors under noisy conditions.
These claims are well supported by a combination of rigorous theoretical analysis and empirical validation. The authors provide clear mathematical derivations and propositions (e.g., Propositions 3.1–3.4, 4.1–4.4) to justify the existence of the coverage gap and the effectiveness of their method. The use of the robust pinball loss is motivated both intuitively and formally (Proposition 3.2), and the theoretical convergence guarantees are carefully proven under standard assumptions.
Empirical results on CIFAR-100 and ImageNet across various architectures, noise levels, and learning rate schedules consistently demonstrate that NR-OCP significantly reduces coverage gaps while producing smaller prediction sets compared to baseline methods. The experimental setup is sound, and the performance gains are consistent and statistically significant.

**Essential References Not Discussed:**

None

**Experimental Designs Or Analyses:**

The experimental evaluation in the paper is generally well-organized and aims to validate the theoretical claims regarding the robustness and efficiency of the proposed NR-OCP method under uniform label noise. the evaluation lacks are in Methods And Evaluation Criteria

**Methods And Evaluation Criteria:**

The proposed method—Noise-Robust Online Conformal Prediction (NR-OCP)—is well-motivated for addressing the robustness of online conformal prediction under uniform label noise. The idea of using a robust pinball loss to correct for the bias introduced by noisy labels is theoretically sound and justified with clear derivations. The authors further evaluate their method across multiple architectures (e.g., ResNet, DenseNet, VGG), datasets (CIFAR-100 and ImageNet), and learning rate schedules (constant and dynamic), which demonstrates the generality of their approach.

However, there are several limitations and areas where the methodology or evaluation could be improved. The baseline comparisons are exclusively standard online conformal prediction methods with noisy labels. It would be valuable to compare against existing noise-robust classification or calibration techniques that could be adapted to the conformal setting. The method introduces several components, but there is **no ablation study** to isolate the impact of each part or to verify whether the form of the robust loss is optimal. While CIFAR-100 and ImageNet with synthetic noise are common benchmarks

**Other Comments Or Suggestions:**

None

**Other Strengths And Weaknesses:**

It was mentioned in the previous content

**Questions For Authors:**

None

**Relation To Broader Scientific Literature:**

The paper contributes to the growing literature on robust uncertainty quantification by extending online conformal prediction (OCP) methods to settings with uniform label noise, a scenario often overlooked in prior works which generally assume clean supervision. While recent studies like Gibbs & Candès (2021) and Angelopoulos et al. (2024) have advanced OCP under distribution shifts, they typically assume label accuracy. The closest related work is by Einbinder et al. (2024), which analyzes OCP under uniform label noise but relies on strong assumptions and offers limited quantitative insight. The proposed NR-OCP method addresses this gap by using a robust pinball loss that allows theoretically sound updates without clean labels. However, the paper’s relevance to the broader literature is limited by several factors: it only addresses uniform label noise with a known noise rate, while more realistic settings often involve instance-dependent or asymmetric noise. Furthermore, the method’s reliance on full per-class score computation may reduce scalability, particularly in large-class scenarios such as language modeling or large-scale classification. The work also does not engage with the broader literature on noise-robust learning (e.g., co-teaching, reweighting, meta-learning) or label-noise-aware calibration, which could offer complementary or alternative approaches. Overall, while the paper fills an important niche within conformal prediction, its connection to the wider field of robust machine learning remains somewhat narrow, and its impact would be strengthened by broader methodological comparisons and extensions beyond the uniform-noise setting.

**Theoretical Claims:**

The core theoretical claims are sound and appear technically correct under their stated assumptions.

---

> ### Author Rebuttal · Authors · 2025-03-31
>
> > 1. Comparisons with existing noise-robust methods
>
> Thank you for raising this concern. We’d like to clarify that this work focuses on label noise in online conformal prediction, a domain where methods originally designed for classification and calibration cannot be easily adapted. In particular, prior noise-robust classification techniques typically aim to address the label noise issue in the training set. In the conformal prediction, however, **label noise occurs within the calibration set** [1,2,3,4], which is used to determine the threshold for generating prediction sets. Thus, the challenge addressed in this paper differs from traditional label-noise learning, rendering existing noise-robust methods unsuitable. Furthermore, although a few methods have been proposed for split conformal prediction [2,3,4], adapting them to the online conformal prediction framework is far from straightforward. In response #1 to reviewer E8K1, we provide a new comparison with an additional online conformal prediction method - SAOCP [5]. We would appreciate it if the reviewer could suggest any specific baselines for comparison.
>
>
> > 2. No ablation study on the proposed loss
>
> Thank you for the suggestion. First, we'd like to clarify that our robust pinball loss is a whole, rather than an assembly of multiple components. In Eq.(3), we present the robust pinball loss with $\ell_1$ and $\ell_2$ to simplify the expression of formula. Our loss function is not separable, otherwise, it will invalidate Prop. 3.2 and thus lose its effectiveness from the theoretical perspective (see the proof sketch of Prop. 3.2).
>
> To fully address this concern, we add an ablation study as you suggested. In particular, we compare the performance of online conformal prediction applied with part of robust pinball loss ($\ell_1$ and $\ell_2$ defined in Eq. (3)) and full loss, under uniform noisy labels with noise rates $\epsilon=0.05,0.1$. We employ LAC score to generate prediction sets with error rate $\alpha=0.1$, using ResNet50 on CIFAR-100. Figure 1 in [[link](https://anonymous.4open.science/r/Noise_Robust_Online_Conformal_Prediction-DD85/Supplementary_Materials_for__Robust_Online_Conformal_Prediction_under_Uniform_Label_Noise.pdf)] shows that employing part of robust pinball loss would violate the precise coverage guarantee, thereby demonstrating the optimality of our loss.
>
>
> > 3. Restriction to uniform label noise
>
> Thank you for your valuable feedback. We’d like to clarify that label noise is a novel challenge in the context of conformal prediction, distinct from the extensively studied field of noise-robust learning. As noted in our Related Work section (Appendix A), only a few recent studies have begun to explore label noise in conformal prediction. Our work stands out as **the first to address this issue specifically in online conformal prediction** - a particularly demanding setting due to the stringent theoretical guarantees required.
>
> Given the nascent state of this topic, existing efforts [1,2,3,4], including the most relevant prior work [1], have focused on simple noise models such as uniform label noise or noise transition matrix. We believe this focus provides a critical foundation, allowing us to establish rigorous theoretical guarantees under a controlled yet meaningful noise scenario. Extending this framework to more complex noise structures, while undoubtedly valuable, poses additional challenges that we view as an exciting direction for future research (see Limitation section in the original manuscript). We hope this clarifies the significance of our work as a starting point in the field of conformal prediction.
>
>
> > 4. The requirement of full per-class score computation
>
> Thank you for raising the concern. We'd like to clarify that the full per-class score computation in this work stems from standard online conformal prediction [6,7], rather than a unique limitation of our method. We agree that reducing this computational burden is a valuable consideration, and we see it as an interesting direction for future work. We appreciate your insight on this point.
>
> ### References
> [1] Einbinder B S, et al. Label noise robustness of conformal prediction. JMLR 2024.
>
> [2] Penso C, et al. A conformal prediction score that is robust to label noise. arXiv preprint 2024.
>
> [3] Sesia M, et al. Adaptive conformal classification with noisy labels. JRSSB 2024.
>
> [4] Bortolotti T, et al. Noise-Adaptive Conformal Classification with Marginal Coverage. arXiv preprint 2025.
>
> [5] Bhatnagar A, et al. Improved online conformal prediction via strongly adaptive online learning. ICML 2023.
>
> [6] Gibbs I, et al. Adaptive conformal inference under distribution shift. NIPS 2021.
>
> [7] Angelopoulos A N, et al. Online conformal prediction with decaying step sizes. ICML 2024.

---

> > ### Comment · Reviewer_rF6c · 2025-04-04
> >
> > I sincerely appreciate the authors’ detailed response and clarification. My concerns have been resolved, and I will be raising my score.

---

> > > ### Author Response · Authors · 2025-04-04
> > >
> > > Thank you for raising the score. We sincerely appreciate your time and effort in reviewing our work.

---

### Decision · Program_Chairs · 2025-05-01

**Decision:**

Reject

**Comment:**

This paper tracks conformal prediction problems in the case where label noise exists. Although the considered problem is new, its real-world impact and necessity are questionable. The proposed method relies on a very strong assumption—namely, that the label noise is uniform and the noise rate is known. However, in most practical scenarios, label noise is typically class-dependent or instance-dependent, and the true noise rate is usually unknown and difficult to estimate reliably. Although the authors mention that the noise rate can be calculated from historical data, this claim lacks empirical support or discussion of the potential impact of misestimated noise rates.

In the experiments, there are no real-world datasets (all hand-crafted), further increasing the concerns about the significance of this studied problem which is mismatched with what the authors want to claim. Although all reviewers have no negative opinions towards this paper, none of them has positive opinions. Thus, this is a borderline paper with less significance to the field.